# Emotional learning retroactively promotes memory integration through rapid neural reactivation and reorganization

Yannan Zhu[1,2,3], Yimeng Zeng[1,2], Jingyuan Ren[3], Lingke Zhang[1,2], Changming Chen[4], Guillen Fernandez[3], Shaozheng Qin[1,2,5]*

[1]State Key Laboratory of Cognitive Neuroscience and Learning & IDG/McGovern Institute for Brain Research, Beijing Normal University, Beijing, China; [2]Beijing Key Laboratory of Brain Imaging and Connectomics, Beijing Normal University, Beijing, China; [3]Donders Institute for Brain, Cognition and Behaviour, Radboud University Medical Center, Nijmegen, Netherlands; [4]School of Education, Chongqing Normal University, Chongqing, China; [5]Chinese Institute for Brain Research, Beijing, China

**Abstract** Neutral events preceding emotional experiences can be better remembered, likely by assigning them as significant to guide possible use in future. Yet, the neurobiological mechanisms of how emotional learning enhances memory for past mundane events remain unclear. By two behavioral studies and one functional magnetic resonance imaging study with an adapted sensory preconditioning paradigm, we show rapid neural reactivation and connectivity changes underlying emotion-charged retroactive memory enhancement. Behaviorally, emotional learning retroactively enhanced initial memory for neutral associations across the three studies. Neurally, emotional learning potentiated trial-specific reactivation of overlapping neural traces in the hippocampus and stimulus-relevant neocortex. It further induced rapid hippocampal-neocortical functional reorganization supporting such retroactive memory benefit, as characterized by enhanced hippocampal-neocortical coupling modulated by the amygdala during emotional learning, and a shift of hippocampal connectivity from stimulus-relevant neocortex to distributed transmodal prefrontal-parietal areas at post-learning rests. Together, emotional learning retroactively promotes memory integration for past neutral events through stimulating trial-specific reactivation of overlapping representations and reorganization of associated memories into an integrated network to foster its priority for future use.

*For correspondence: szqin@bnu.edu.cn

## Editor's evaluation

This manuscript presents valuable insights into neural mechanisms driving emotional memory enhancements for previously neutral information. The authors present compelling behavioral evidence that memory for items indirectly paired with an aversive event is enhanced, and that these enhancements are associated with increased fMRI signal interactions between the amygdala, hippocampus, and stimulus-relevant cortex. The work will be interesting to researchers interested in learning and memory.

## Introduction

Emotion shapes learning and memory for our episodic experiences. Experiencing an emotional event such as a psychological trauma, for instance, often not only strengthens our memory for the event itself, but also can benefit memories for other mundane events occurring separately in time (*LaBar*

*and Cabeza, 2006*; *Li et al., 2008*; *Wong et al., 2019*). There has been substantial progress in understanding the mechanisms underlying memory enhancement for emotional events themselves, owing to autonomic reactions to emotional arousal that stimulates the encoding and post-encoding processes of emotional memory (*Hamann, 2001*; *LaBar and Cabeza, 2006*). Yet, our understanding of the mechanisms of how emotional learning prompts memory-related brain systems in a way that retroactively enhances memory for past neutral events is still in its infancy. In many circumstances, the significance of our experiences such as reward or punishment often occurs after the event. Since we cannot determine which event will become significant later, human episodic memory is theorized to organize our experiences into a highly adaptive network of integrated representations, that can be prioritized in terms of the significance of preceding or following events (*Ritchey et al., 2016*; *Shohamy and Daw, 2015*). Such a mechanism allows seemingly mundane events to take on significance following a salient experience or emotional arousal, thereby generalizes their memories to future use in ever-changing environment (*Holmes et al., 2018*; *Wong et al., 2019*). However, this retroactive effect may also lead to maladaptive generalization recognized as a cognitive hallmark of core symptoms in some mental disorders, like posttraumatic stress disorder (PTSD) and phobia (*Lange et al., 2019*; *Mary et al., 2020*). Deciphering the neurobiological mechanisms of emotion-charged retroactive memory enhancement in humans is thus critical for further understanding of maladaptive generalization in these disorders.

In past decades, the emotion-charged retroactive memory effects are investigated by behavioral tagging and sensory preconditioning protocols in animals and humans. Studies of the behavioral tagging have provided prominent evidence for the delayed retroactive memory enhancement following a strongly salient or emotional experience, typically resulting in a generalized form of enhancement on initial weak memories encoded closely in time (*Ballarini et al., 2009*; *Dunsmoor et al., 2022*; *Takeuchi et al., 2016*). In our everyday memory, however, emotion-induced retroactive effect often occurs on some specific events through rapid trial-specific form of emotional learning. Such rapid effect reflects highly adaptative and flexible features of human episodic memory systems (*Howard et al., 2015*), which cannot readily be explained by the conventional behavioral tagging models. We thus focused on sensory preconditioning protocols with trial-specific associative learning tasks (*Brogden, 1939*). This paradigm typically consists of two stages: two neutral events (i.e. direct and indirect events) are paired together in an initial learning phase, and then the direct event is paired with a salient stimulus (i.e. threat or reward)—namely an emotional or reward learning phase (*Kurth-Nelson et al., 2015*; *Sharpe et al., 2017*; *Wong et al., 2019*). Animal work has shown that the salient stimulus can spread its value to the indirect neutral event through an integration mechanism (*Holmes et al., 2018*; *Sadacca et al., 2018*; *Sharpe et al., 2017*). Human studies have also suggested that a subsequent salient event could enhance perceptual discrimination and guide decision bias for its related neutral events (*Kuhl et al., 2010*; *Li et al., 2008*; *Wimmer and Shohamy, 2012*). These studies in the context of sensory preconditioning provide a probability that a retroactive memory benefit could emerge for specific neutral information that occurs before emotional learning. However, it remains elusive about the neurocognitive mechanisms of how emotional learning retroactively enhances memory for previously neutral events in a trial-specific manner.

Recent memory integration views of sensory preconditioning paradigm posit that, as new emotional experiences are encoded, related memories for past sensory (neutral) events can be reactivated and integrated with the emotional memory content, essentially resulting in a retroactive memory benefit (*Holmes et al., 2022*; *Schlichting and Frankland, 2017*; *Shohamy and Daw, 2015*; *Wong et al., 2019*). By this view, there appears an integrative encoding mechanism – that is, memories for related events are encoded into an integrated network across associations with overlapping representations (*Schlichting and Preston, 2014*; *Shohamy and Wagner, 2008*). Human neuroimaging studies provide compelling evidence supporting the integrative encoding mechanism, by which a newly encoded event can be updated and reorganized into relevant episodic memory through reactivation of overlapping neural ensembles engaged in both initial and new learning (*Schlichting and Preston, 2015*; *Shohamy and Wagner, 2008*; *van Kesteren et al., 2016*). The hippocampus serves as an integrative hub by binding disparate representations in stimulus-sensitive neocortical areas into episodic memories (*Kuhl et al., 2010*; *van Kesteren et al., 2016*). Reactivation of hippocampal and stimulus-sensitive neocortical representations has been well proposed to promote systems-level memory integration (*Schlichting and Preston, 2014*; *Sutherland and McNaughton, 2000*; *Zhuang et al., 2022*). The

reactivation of overlapping neural traces is considered as a scaffold for integrating newly learnt information into existing memories, making it possible to reorganize memories according to their future significance. However, it remains unknown how such reactivation occurs at a trial-specific level during emotional learning and then contributes to memory enhancement for initially mundane information. Based on above neurobiological models and empirical observations in neuroimaging studies, we hypothesized that this emotion-charged retroactive benefit would result from increased trial-specific reactivation of hippocampal and stimulus-sensitive neocortical representations, which enhances the memory association between initially learnt events and promotes their integration.

Emotion-charged memory enhancement is most likely based on autonomic reactions associated with emotional arousal, accompanying with (nor)adrenergic signaling that modulates neural ensembles in the amygdala as well as the hippocampus and related neural circuits (*Hamann, 2001*; *LaBar and Cabeza, 2006*). Many studies have suggested that emotional arousal can lead to better episodic memory through strengthening hippocampal connectivity with the amygdala and related neocortical regions during emotional memory encoding (*Dolcos et al., 2004*; *Hermans et al., 2014*; *Richardson et al., 2004*; *Ritchey et al., 2013*). Beyond online encoding, offline processes at awake rest and sleep also contribute to emotional memory benefit, most likely involving reconfiguration of hippocampal functional connectivity with distributed neocortical networks (*de Voogd et al., 2016*; *Schlichting and Frankland, 2017*; *Tambini et al., 2010*). Recent studies have demonstrated that neural engagement, for instance, excitable hippocampal-neocortical coordinated interactions prior to encoding could affect the allocation of new information into specific neural populations and form a preparatory network for igniting reactivation or replay cascades to support memory integration (*Josselyn and Frankland, 2018*; *Kaefer et al., 2022*; *Schlichting and Frankland, 2017*; *van Dongen et al., 2011*). Thus, it is possible that an excitable brain state at pre-emotional-encoding rest might modulate subsequent emotional memory integration and predict memory performance. Moreover, notable emotion-modulated changes in hippocampal-neocortical connectivity at post-emotional-encoding rest are also thought to associate with subsequent memory performance (*de Voogd et al., 2016*; *Hermans et al., 2017*; *Murty et al., 2017*). However, it remains unknown how emotion-related online and offline mechanisms support the retroactive benefit on episodic memory. Given these previous findings, we hypothesized that emotional learning would retroactively promote memory integration for neutral events, by acting on not only hippocampal dialogue with the amygdala and related neocortical regions during online encoding, but also hippocampal-neocortical reorganization during offline resting.

Here, we tested above hypotheses by two behavioral studies and one event-related functional magnetic resonance imaging (fMRI) study using an adapted sensory preconditioning paradigm. In each of the three studies, the experimental design consisted of three phases: an initial learning, a following emotional learning and a surprise associative memory test (*Figure 1A*). In Study 1 (N=30), participants learnt 72 neutral face-object pairs during the initial learning phase. Thereafter, each face from the initial learning phase was presented as a cue and paired with either an aversive screaming voice (i.e. aversive condition) or a neutral voice (i.e. neutral condition) during the emotional learning phase. A surprise associative memory test was administered 30 min later to assess memory performance for initial face-object associations. In Study 2 (N=28), the reproducibility of Study 1 was examined by another independent research group. In Study 3 (N=28), participants underwent fMRI scanning during the initial and emotional learning phases that were interleaved by three rest scans, with concurrent skin conductance recording to monitor autonomic arousal. A set of multi-voxel pattern similarity (*Zhuang et al., 2022*), task-dependent functional connectivity and machine learning-based prediction analyses, in conjunction with exploratory task-free functional connectivity and mediation analyses, were implemented to assess trial-specific reactivation of initial learning activity, functional coupling during online emotional learning and offline post-initial/emotional-learning rests, as well as their relationships with emotion-charged retroactive memory enhancement. Consistent with our hypotheses, we found that emotional learning retroactively enhanced memories for initially learnt neutral associations. This rapid retroactive enhancement was associated with prominent increases in trial-specific reactivation of initial learning activity in the hippocampus and stimulus-sensitive neocortex, as well as strengthening hippocampal coupling with the amygdala and related neocortical areas during emotional learning and hippocampal-neocortical offline reorganization during post-learning rests. Our findings suggest the neurobiological mechanisms by which memories for mundane neutral information can be enhanced

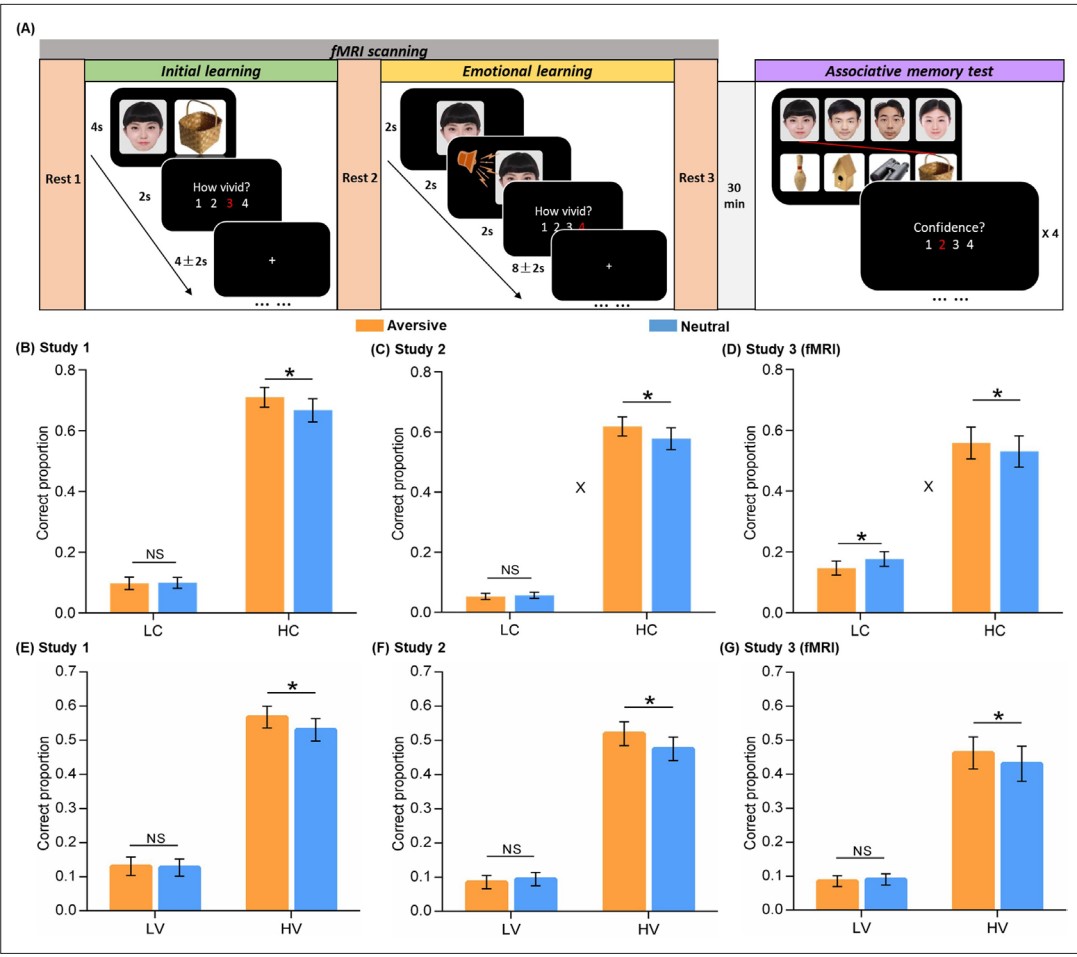

**Figure 1.** Experimental design and behavioral performance. (**A**) The experiment consisted of three phases. During the initial learning phase, participants were instructed to vividly imagine each face and its paired object interacting with each other. During the emotional learning phase, each face from initial learning was presented again and then paired with either an aversive scream or a neutral voice. Participants were also instructed to imagine each face and its paired voice interacting. After a 30-min delay, participants performed a recognition memory test for face-object associations by matching each face (*top*) with one of the four objects (*bottom*). (**B, C, D**) Associative memory performance as a function of confidence ratings and emotional conditions in Study 1 (n=30), 2 (n=28), and 3 (fMRI, n=28). Bar graphs depict averaged correct proportion for face-object associations remembered with low (LC) and high (HC) confidence in aversive and neutral conditions separately. (**E, F, G**) High-confidence memory performance as a function of vividness ratings and emotional conditions in Study 1, 2, and 3 (fMRI). Bar graphs depict averaged correct proportion for face-object associations remembered with high confidence, which were rated with low (LV) and high (HV) vividness during the initial learning phase in aversive and neutral conditions separately. Error bars represent standard error of mean. 'X' indicates significant interaction (p<0.05). Notes: NS, non-significant; *p<0.05; two-tailed tests. The sample sizes are 30, 28 and 28 participants separately in Study 1, 2 and 3, unless otherwise noted in the following figure legends.

The online version of this article includes the following figure supplement(s) for figure 1:

**Figure supplement 1.** Emotional ratings on the four voices.

**Figure supplement 2.** Face item memory performance.

**Figure supplement 3.** Associative memory performances on the four confidence levels.

**Figure supplement 4.** Relationships between vividness ratings and confidence ratings.

**Figure supplement 5.** Low-confidence memory performance as a function of vividness and emotion.

**Figure supplement 6.** Skin conductance levels (SCLs) in Study 3 (fMRI).

when subsequently associated with emotionally arousing experiences, through rapid trial-specific reactivation of overlapping neural traces and hippocampal-neocortical functional reorganization involved in memory integration.

## Results

### Emotional learning retroactively enhances its related memory for initial neutral association

We first examined the emotion-charged retroactive effect for associative memory performance in two behavioral studies and one fMRI study separately. One-sample t-tests on each confidence level in the memory test revealed that the proportion of correctly remembered face-object associations with a 3 or 4 confidence rating (i.e. a four-point rating scale: 4 = 'Very confident', 1 = 'Not confident at all') was significantly higher than chance [(1/4+1/3+1/2+1)/4*100%=52%] across three studies (all p<0.05; *see Figure 1—figure supplement 3 for statistics*). However, the proportion of remembered associations with a 1 or 2 rating was not reliably higher than chance (*see Figure 1—figure supplement 3 for statistics*). It indicates that remembering with high confidence may reflect more reliable and stronger memories as compared to low confident remembrance. Thus, all remembered associations were then sorted into a high confident bin with 3 and 4 ratings, and a low-confident bin with 1 and 2 ratings likely reflecting guessing or familiarity.

Separate 2 (Emotion: aversive vs. neutral) by 2 (Confidence: high vs. low) repeated-measures ANOVAs were conducted for the three studies. For Study 1, this analysis revealed main effects of Emotion ($F_{(1,29)}$ = 5.18, p=0.030, partial $\eta^2$=0.15) and Confidence ($F_{(1,29)}$ = 142.07, p<0.001, partial $\eta^2$=0.83; *Figure 1B*). Although we only observed a trending Emotion-by-Confidence interaction effect in Study 1 ($F_{(1,29)}$ = 2.57, p=0.120, partial $\eta^2$=0.08), it showed significant interaction effects in Studies 2 and 3 consistently (*see below for statistics*). Post-hoc comparisons revealed better memory for face-object associations only with high confidence in the aversive than neutral condition ($t_{(29)}$ = 2.18, p=0.037, $d_{av}$ = 0.22), but not with low confidence ($t_{(29)}$ = –0.15, p=0.883). Parallel analysis in Study 2 also revealed main effects of Emotion ($F_{(1,27)}$ = 5.53, p=0.026, partial $\eta^2$=0.17) and Confidence ($F_{(1,27)}$ = 206.14, p<0.001, partial $\eta^2$=0.88), and a significant Emotion-by-Confidence interaction ($F_{(1,27)}$ = 4.94, p=0.035, partial $\eta^2$=0.16; *Figure 1C*). Post-hoc comparisons also revealed better associative memory with high confidence in the aversive (vs. neutral) condition ($t_{(27)}$ = 2.47, p=0.020, $d_{av}$ = 0.23), but not with low confidence ($t_{(27)}$ = –0.55, p=0.588). In Study 3 (fMRI), we again observed a similar pattern of results, including the main effect of Confidence ($F_{(1,27)}$ = 29.35, p<0.001, partial $\eta^2$=0.52) and the significant Emotion-by-Confidence interaction ($F_{(1,27)}$ = 8.61, p=0.007, partial $\eta^2$=0.24; *Figure 1D*), specifically with a retroactive enhancement on associative memory with high confidence in the aversive (vs. neutral) condition ($t_{(27)}$ = 2.59, p=0.015, $d_{av}$ = 0.10), but an opposite effect for associations with low confidence ($t_{(27)}$ = –2.18, p=0.038, $d_{av}$ = 0.24). These results indicate a reliable emotion-induced retroactive enhancement on related memories for neutral associations remembered with high confidence.

Besides confidence level for final memory performance, we also investigated whether the initial encoding strength of neutral associations modulates our observed emotion-induced retroactive effect. In order to examine how much distinct effect the initial encoding strength, as indicated by vividness rating during initial learning, would contribute to subsequent emotional memory benefit, we tested the trial-level relationship between vividness ratings and confidence ratings with a linear mixed-effects model. The linear mixed-effects model included all rating trials across participants, with participant as a random effect to reduce the bias of inherent correlation among ratings on the same participant (*Gałecki and Burzykowski, 2013*; *Singh et al., 2021*). This analysis revealed significantly positive but moderate correlations between vividness ratings (i.e. a four-point scale: 4 = 'Very vivid', 1 = 'Not vivid at all') and confidence ratings across three studies (Study 1: $\beta$=0.15, p<0.001, 95% CI = [0.07, 0.22]; Study 2: $\beta$=0.13, p<0.001, 95% CI = [0.07, 0.19]; Study 3: $\beta$=0.24, p<0.001, 95% CI = [0.16, 0.32]; *Figure 1—figure supplement 4*). These results indicate that extending beyond confidence rating, the vividness rating as a proxy for initial encoding strength might have a potentially distinct effect on subsequent emotional learning.

Subsequently, we further sorted the neutral associations remembered with high confidence into a high-vividness bin with 3 and 4 ratings, and a low-vividness bin with 1 and 2 ratings. Separate paired

t-tests for the three studies revealed significantly better memory in the aversive (vs. neutral) condition, selectively for high-vividness encoded associations (Study 1: $t_{(29)}$ = 2.15, p=0.040, $d_{av}$ = 0.21; Study 2: $t_{(27)}$ = 2.16, p=0.040, $d_{av}$ = 0.25; Study 3: $t_{(27)}$ = 2.21, p=0.036, $d_{av}$ = 0.12), but not for low-vividness encoded associations (Study 1: $t_{(29)}$ = 0.27, p=0.793; Study 2: $t_{(27)}$ = –0.58, p=0.565; Study 3: $t_{(27)}$ = –0.40, p=0.696; *Figure 1E–G*). Parallel analyses were also conducted for those associations remembered with low confidence. However, we did not observe any reliable effect across three studies (*see Figure 1—figure supplement 5 for statistics*). Altogether, these results indicate that our observed emotion-induced retroactive benefit for related memory only occurs on relatively strong-encoded associations as indicated by high-vividness ratings.

In addition, to verify the elevation of autonomic arousal during emotional learning, we conducted a 2 (Emotion: aversive vs. neutral) by 2 (Phase: initial learning vs. emotional learning) repeated-measures ANOVA for skin conductance levels (SCLs) in the fMRI study (Study 3). This analysis revealed significant main effect of Phase ($F_{(1, 19)}$=12.23, p=0.002, partial $\eta^2$=0.39) as well as Emotion-by-Phase interaction ($F_{(1, 19)}$=4.73, p=0.043, partial $\eta^2$=0.20), but no main effect of Emotion ($F_{(1, 19)}$=0.40, p=0.533, partial $\eta^2$=0.02; *Figure 1—figure supplement 6*). Post-hoc comparisons revealed significantly higher SCL in the aversive than neutral condition during the emotional learning phase ($t_{(19)}$ = 2.23, p=0.038, $d_{av}$ = 0.50), but not during the initial learning phase ($t_{(19)}$ = –1.16, p=0.26). These results prove a higher level of automatic arousal successfully induced in the aversive (vs. neutral) condition during emotional learning, which potentially leads to a series of above-mentioned emotion-charged effects.

## Emotional learning potentiates trial-specific reactivation of initial learning activity in the hippocampus and neocortex

Next, we investigated how emotional learning affects reactivation of initial learning activity for face-object associations cued by overlapping faces (Study 3). To assess reactivation of neural activity, we estimated the similarity of stimulus-evoked multi-voxel activation patterns between the initial learning phase and the subsequent emotional learning phase. Given our priori hypothesis, we focused on the hippocampus and stimulus-sensitive cortical regions including the lateral occipital complex (LOC) for object processing (*Grill-Spector et al., 2001*; *Malach et al., 1995*) and the face fusiform area (FFA) for face processing (*Kanwisher et al., 1997*). The bilateral hippocampal, bilateral ventral LOC (vLOC) and bilateral FFA ROIs were functionally identified by the overlapping area of two activation contrasts of face-object initial encoding and face-voice emotional encoding relative to fixation (i.e. all encoding trials vs. fixation during each learning phase; *Figure 2—figure supplement 1*), in order to specify the most engaged regions in both initial and emotional learning (i.e. activated and/or reactivated). The hippocampal ROI was further constrained by an anatomical mask of the bilateral hippocampus from the WFU PickAtlas. The vLOC and FFA ROIs were further constrained by a respective LOC or FFA mask, which is the overlapping area of an anatomical mask from the WFU PickAtlas (i.e. a composite AAL template including the bilateral middle occipital cortex, bilateral middle temporal cortex and bilateral fusiform gyrus for LOC, and an AAL template of the bilateral fusiform gyrus for FFA) and a functionally defined mask from the Neurosynth platform (i.e. 'object recognition' or 'face recognition' as a searching term; *see Methods for details*).

We conducted a condition-level pattern similarity analysis to investigate the overall emotion-charged neural reactivation in above three ROIs. As shown in *Figure 2—figure supplement 2*, we computed similarities between phases during the presentation of face-voice associations for each condition, while taking the presentation of face cues as an illustration purpose only. These analyses revealed a general emotion-induced increase in reactivation of initial learning activity in the hippocampus and stimulus-sensitive vLOC and FFA (*see Figure 2—figure supplement 2 for statistics*). In addition, we implemented a whole-brain exploratory analysis on condition-level reactivation with searchlight algorithm. Beyond confirming emotion-induced increased reactivation in the hippocampal, vLOC and FFA ROIs, the whole-brain searchlight analysis identified a set of other brain regions that exhibited greater neural reactivation in the aversive than neutral condition. These regions included the superior medial frontal cortex, insula, precuneus, and angular gyrus (*Supplementary file 2*).

To further examine whether the emotion-charged increase in neural reactivation is due to the trial-specific reinstatement or a broad category representation pattern, we conducted a set of trial-level pattern similarity analyses. As shown in *Figure 2A* (*also see Figure 2—figure supplement 3*), we computed the pair-specific similarity for each pair (i.e. correlation between each face-voice

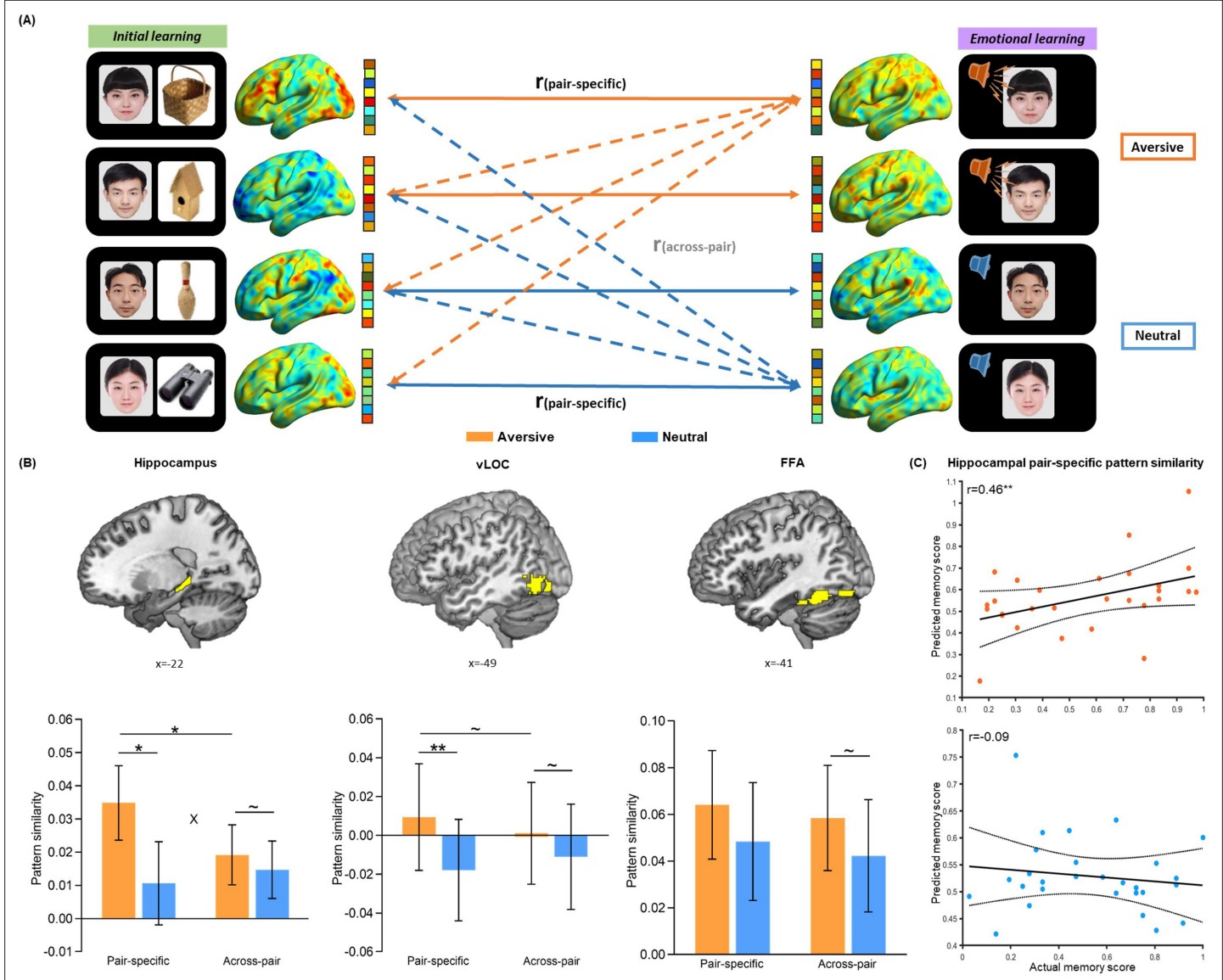

**Figure 2.** Trial-level neural reactivation of initial learning activity during emotional learning. (**A**) An illustration of trial-level reactivation analysis. Example data was from one subject. During initial learning (*left*), sagittal views of activation maps for four trials were shown. During emotional learning (*right*), sagittal views of activation maps for the corresponding trials with two in aversive condition and two in neutral condition were shown. Solid lines indicate correlations for pair-specific similarity measure and dash lines indicate correlations for across-pair similarity measure. These correlations from each similarity measure were then averaged across trials for each participant in aversive and neutral conditions separately. (**B**) Bar graphs depict the average pair-specific and across-pair pattern similarities in aversive and neutral conditions for the bilateral hippocampus (*left*), bilateral ventral LOC (vLOC, *middle*) and bilateral FFA (*right*) ROIs. 'X' indicates a significant interaction (p<0.05). Error bars represent standard error of mean (n=28). (**C**) Scatter plots depict correlations of observed associative memory performance (i.e. remembered with high confidence) with predicted memory outcome from machine learning prediction analysis based on hippocampal pair-specific pattern similarity in aversive and neutral conditions. Dashed lines indicate 95% confidence intervals, and solid lines indicate the best linear fit. Dots represent data from each participant (n=28). Notes: ~p < 0.10; *p<0.05; **p<0.01; two-tailed tests.

The online version of this article includes the following figure supplement(s) for figure 2:

**Figure supplement 1.** Brain systems involved in memory encoding effect during initial and emotional learning phases.

**Figure supplement 2.** Condition-level neural reactivation of initial learning activity during emotional learning.

**Figure supplement 3.** Trial-level pattern similarity analysis approach.

**Figure supplement 4.** Additional trial-level pattern similarity results.

**Figure supplement 5.** Prediction relationships between hippocampal between-phases pattern similarity and associative memory performance.

**Figure supplement 6.** Prediction relationships between hippocampal within-phase pattern similarity and associative memory performance.

pattern during emotional learning and its corresponding face-object pattern during initial learning) to measure trial-specific reinstatement, and the across-pair similarity among different pairs (i.e., averaging all correlations between each face-voice pattern during emotional learning and other different face-object patterns during initial learning) to measure category-level representation. These similarity measures were then entered into separate 2 (Emotion: aversive vs. neutral) by 2 (Measure: pair-specific vs. across-pair) repeated-measures ANCOVAs for each ROI, with individual's univariate activation differences (i.e. aversive vs. neutral) in both initial and emotional learning phases as covariates of no interest. These analyses revealed significant main effects of Emotion in the hippocampus ($F_{(1, 25)}$=9.48, p=0.005, partial $\eta^2$=0.28) and vLOC ($F_{(1, 25)}$=9.34, p=0.005, partial $\eta^2$=0.27), as well as a significant Emotion-by-Measure interaction effect in the hippocampus ($F_{(1, 25)}$=4.89, p=0.036, partial $\eta^2$=0.16) and a similar trend but non-significant interaction in the vLOC ($F_{(1, 25)}$=3.54, p=0.072, partial $\eta^2$=0.12), but no main effect of Measure in the two ROIs (both $F_{(1, 25)}$<0.60, p>0.440, partial $\eta^2$<0.03; *Figure 2B*). Post-hoc comparisons (with covariates controlled) revealed significantly higher pair-specific similarity in both hippocampus ($F_{(1, 25)}$=7.24, p=0.013, partial $\eta^2$=0.22) and vLOC ($F_{(1, 25)}$=8.28, p=0.008, partial $\eta^2$=0.25) in the aversive than neutral condition. But it only revealed non-significant trends of higher across-pair similarity in the hippocampus ($F_{(1, 25)}$=4.16, p=0.052, partial $\eta^2$=0.14) and vLOC ($F_{(1, 25)}$=3.37, p=0.078, partial $\eta^2$=0.12) in the aversive (vs. neutral) condition. We also observed significantly higher pair-specific than across-pair similarity in the hippocampus ($F_{(1, 25)}$=4.42, p=0.046, partial $\eta^2$=0.15) and a similar but non-significant trend in the vLOC ($F_{(1, 25)}$=2.96, p=0.097, partial $\eta^2$=0.11) in the aversive condition, but not in the neutral condition ($F_{(1, 25)}$<2.20, p>0.150, partial $\eta^2$<0.08 in both ROIs). Parallel analysis for FFA pattern similarity revealed neither main effect nor interaction effect (all $F_{(1, 25)}$<1.59, p>0.219, partial $\eta^2$<0.06; *Figure 2B*). These results indicate that emotional learning prompts greater trial-specific reinstatement relative to category-level representation in the hippocampus, and it also leads to a similar but non-significant trend in the vLOC.

Critically, we conducted a machine learning-based prediction analysis to examine the relationships of trial-specific reinstatement (i.e. pair-specific similarity) and category-level representation (i.e. across-pair similarity) with associative memory performance (i.e. remembered with high confidence). Individual's pair-specific or across-pair similarity for each of aversive and neutral conditions was entered as an independent variable, and memory performance in the corresponding condition was entered as a dependent variable into separate linear regression models in the prediction analysis. These analyses revealed that hippocampal pair-specific similarity positively predicted memory in the aversive ($r_{(predicted, observed)}$=0.46, p=0.008) but not neutral condition ($r_{(predicted, observed)}$=–0.09, p=0.480; *Figure 2C*). Further Steiger's test (*Steiger, 1980*) revealed a significant difference in correlation coefficients between aversive and neutral conditions (z=2.54, p=0.011). Hippocampal across-pair similarity positively predicted memory in both aversive ($r_{(predicted, observed)}$=0.43, p=0.014) and neutral ($r_{(predicted, observed)}$=0.50, p=0.001) conditions (*Figure 2—figure supplement 5A*), but with no significant difference between two conditions (z=–1.09, p=0.275). The prediction effects of hippocampal pair-specific (vs. across-pair) similarity on memory were non-significant in both conditions (both $r_{(predicted, observed)}$<0.10, p>0.201; *Figure 2—figure supplement 5B*). These results reveal an emotion-specific predictive effect of trial-specific reinstatement (as indicated by pair-specific similarity) for memory in the hippocampus, supporting the emotion-charged retroactive memory benefit. However, reliable category-level representation (as indicated by across-pair similarity) in the hippocampus predicts better memory regardless of emotion modulation, suggesting a general benefit on overall memory.

Additionally, to investigate whether the consistency of activity pattern during each phase accounts for the memory benefit as an alternative explanation, we computed the within-encoding similarity (i.e. averaging all correlations among face-object patterns during the initial learning phase), and the within-arousal similarity (i.e. averaging all correlations among face-voice patterns during the emotional learning phase) in aversive and neutral conditions separately (*Figure 2—figure supplement 3*). Machine learning-based prediction analyses revealed no any reliable relationship of within-encoding similarity (aversive: $r_{(predicted, observed)}$=–0.20, p=0.676; neutral: $r_{(predicted, observed)}$=–0.11, p=0.543; *Figure 2—figure supplement 6A*) or within-arousal similarity (aversive: $r_{(predicted, observed)}$=–0.28, p=0.960; neutral: $r_{(predicted, observed)}$=–0.28, p=0.960; *Figure 2—figure supplement 6B*) with associative memory performance (i.e. also remembered with high confidence) in both conditions. These results indicate that our observed emotion-charged retroactive memory enhancement is not related to the reliability or consistency of activity pattern within each phase.

# Emotional learning enhances hippocampal coupling with the amygdala and neocortex which predicts associative memory for initial neutral events

To investigate how emotional learning modulates functional interactions of the hippocampus and its related neural circuits involved in memory integration, we conducted a task-dependent psycho-physiological interaction (PPI) analysis for the emotional learning phase to assess functional connectivity of the hippocampal seed with every other voxel of the brain (*Figure 3A*). In line with our priori hypothesis, we mainly focused on significant clusters in the right amygdala (*Figure 3B*), left middle portion of FFA (mFFA; *Figure 3C*) and left superior portion of LOC (sLOC; *Figure 3D*), which showed greater functional coupling with the hippocampus in the aversive (vs. neutral) condition during emotional learning (*see Figure 3—figure supplement 1 for visualization; Supplementary file 3*). The amygdala ROI was further constrained by an anatomical mask of the bilateral amygdala from the WFU PickAtlas. The mFFA and sLOC ROIs were also constrained separately by the FFA or LOC mask, same with above pattern similarity analysis (*see Methods*). We also conducted a parallel control PPI analysis for the initial learning phase. This analysis revealed no any reliable emotional effect (i.e. aversive vs. neutral) during initial learning in the three ROIs identified above (*see Figure 3—figure supplement 1 for visualization*). These results indicate that emotional learning induces functional connectivity changes with prominent increases in hippocampal-amygdala and hippocampal-neocortical coupling.

We then investigated how emotion-charged hippocampal connectivity contributes to the retroactive benefit on associative memory. Data extracted from hippocampal connectivity with the amygdala and stimulus-sensitive neocortical regions (i.e. mFFA and sLOC) during the emotional learning phase were submitted to separate 2 (Emotion: aversive vs. neutral) by 2 (Memory: remembered with high confidence vs. forgotten) repeated-measures ANOVAs. The main effects of Emotion were expected, as these three ROIs were defined by the emotional contrast of aversive relative to neutral condition. Thus, we mainly examined whether such emotion-enhanced hippocampal connectivity is associated with the retroactive memory benefits, by focusing on the main effect of Memory and Emotion-by-Memory interaction. This analysis revealed significant main effects of Memory in hippocampal connectivity with the mFFA ($F_{(1, 26)}$=5.59, p=0.026, partial $\eta^2$=0.18; *Figure 3F*) and sLOC ($F_{(1, 26)}$=4.40, p=0.046, partial $\eta^2$=0.15; *Figure 3G*), but not in hippocampal-amygdala connectivity ($F_{(1, 26)}$=0.21, p=0.65, partial $\eta^2$=0.008; *Figure 3E*). There was no reliable Emotion-by-Memory interaction showing in the hippocampal connectivity with these three ROIs (all $F_{(1,26)}$ < 0.73, p>0.400, partial $\eta^2$<0.03; *Figure 3E–G*).

Given the modulatory effects of emotion for hippocampal connectivity reported in previous studies (*de Voogd et al., 2017*; *Richardson et al., 2004*) and emotion-charged neural reactivation in relation to associative memory obeserved in our study, we assumed that hippocampal connectivity, which works in concert with reactivation to promote memory integration (*Schlichting and Preston, 2014*; *Sutherland and McNaughton, 2000*), might also show some emotion-specific relationship with associative memory. Although we did not find reliable interaction effect in above ANOVA analyses, we employed the machine learning-based prediction analysis to investigate whether greater hippocampal coupling during emotional learning could predict better memory and whether this relationship would show emotional specificity. Interestingly, we found that greater hippocampal connectivity with mFFA and sLOC (i.e. remembered with high confidence relative to forgotten) were predictive of better associative memory (i.e. remembered with high confidence) in the aversive condition (mFFA: $r_{(predicted, observed)}$=0.57, p<0.001; sLOC: $r_{(predicted, observed)}$=0.65, p<0.001; *Figure 3H–J*), but not in the neutral condition (both $r_{(predicted, observed)}$<0.15, p>0.160; *Figure 3—figure supplement 2*). A similar trend (though not significant) was also shown in hippocampal-amygdala connectivity (i.e. remembered with high confidence relative to forgotten; aversive: $r_{(predicted, observed)}$=0.24, p=0.076; neutral: $r_{(predicted, observed)}$=–0.25, p=0.839). Further Steiger's tests revealed significant differences in correlation coefficients between two conditions for hippocampal coupling with mFFA (z=2.13, p=0.033) and sLOC (z=2.15, p=0.032), and a non-significant trend of difference for hippocampal-amygdala coupling (z=1.79, p=0.073). These results indicate that emotion-charged hippocampal connectivity with stimulus-sensitive neocortical regions positively predicts associative memory in the aversive but not neutral condition, implying an emotional specificity effect, though hippocampal-amygdala connectivity only shows a similar trend but non-significant effect.

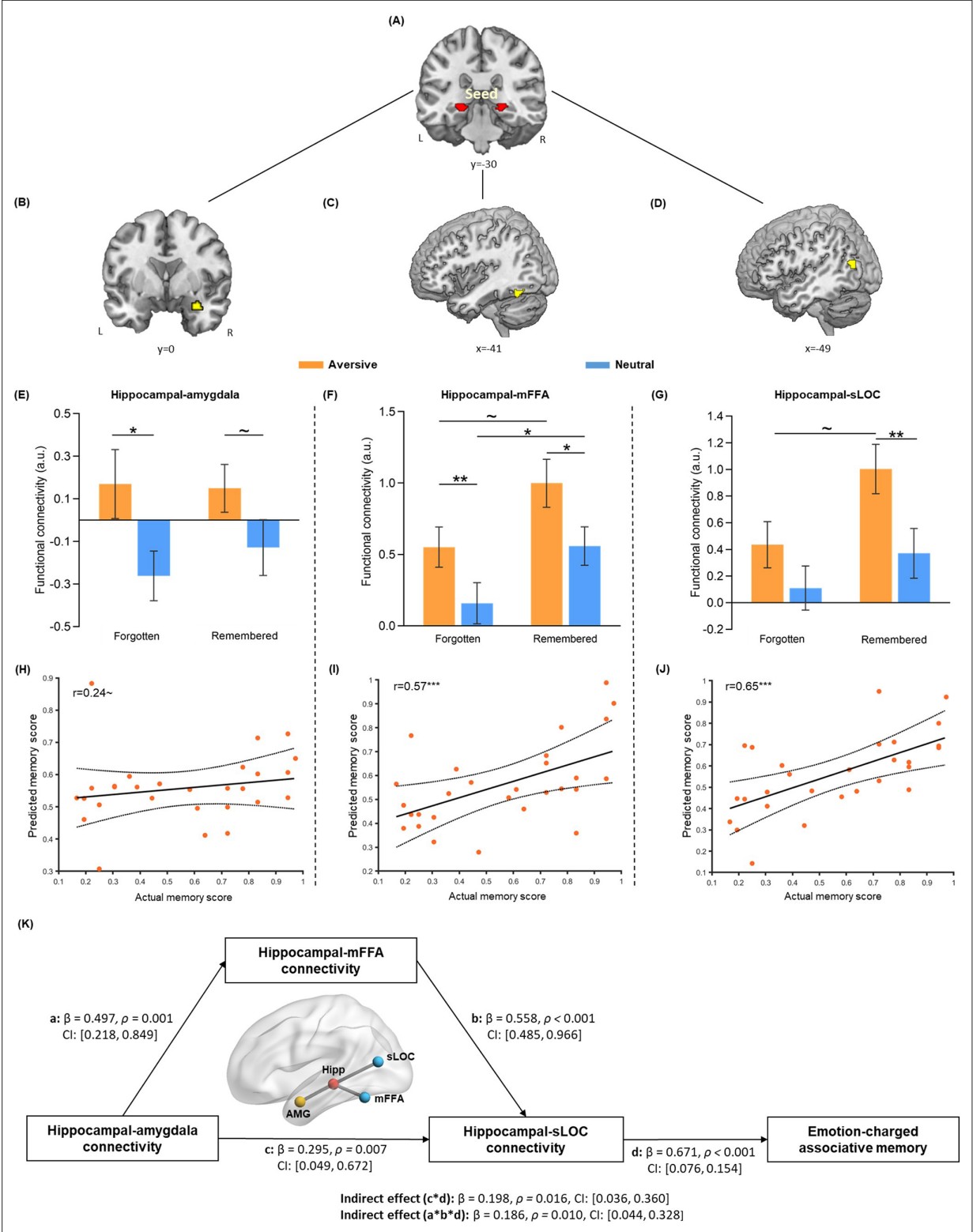

**Figure 3.** Hippocampal connectivity with the amygdala and neocortical regions during emotional learning phase accounting for emotion-charged retroactive memory enhancement. (**A**) The bilateral hippocampal seed used in task-dependent gPPI functional connectivity analysis (i.e. the same hippocampal ROI used in pattern similarity analysis). (**B–D**) Significant clusters in the right amygdala, left middle FFA (mFFA) and left superior LOC (sLOC) regions, showing greater functional coupling with the hippocampus in the aversive (vs. neutral) condition during emotional learning (n=28). (**E–G**) Bar graphs depict averaged hippocampal connectivity with the amygdala, mFFA and sLOC for face-object associations remembered with high

*Figure 3 continued on next page*

*Figure 3 continued*

confidence and forgotten in aversive and neutral conditions separately. Error bars represent standard error of mean (n=27, one participant had no forgotten trial in the neutral condition). (**H–J**) Scatter plots depict correlations of observed memory performance (i.e. remembered with high confidence) with predicted memory outcome from prediction analysis based on hippocampal-amygdala, hippocampal-mFFA and hippocampal-sLOC connectivity (i.e. remembered with high confidence vs. forgotten) in the aversive condition. Dashed lines indicate 95% confidence intervals, and solid lines indicate the best linear fit. Dots represent data from each participant (n=28). (**K**) The mediating effect of hippocampal-mFFA/sLOC connectivity on the positive association between hippocampal-amygdala connectivity and emotion-charged associative memory (n=28). Paths are marked with standardized coefficients. Solid lines indicate significant paths. Notes: ~p < 0.08; *p<0.05; **p<0.01; ***p<0.001; two-tailed tests.

The online version of this article includes the following figure supplement(s) for figure 3:

**Figure supplement 1.** Hippocampal connectivity in aversive and neutral conditions during initial and emotional learning phases.

**Figure supplement 2.** Prediction relationships between hippocampal connectivity and associative memory performance in the neutral condition.

**Figure supplement 3.** Alternative models for the relationships among hippocampal connectivity and associative memory in the aversive condition.

## Emotion-charged retroactive memory enhancement is associated with hippocampal connectivity with the amygdala and neocortex during emotional learning

The modulatory role of the amygdala on hippocampal dialogue with the neocortex is recognized to promote more efficient information transmission and communication between the hippocampus and related neocortical regions (*Alvarez et al., 2008*; *Hermans et al., 2017*; *Phelps and LeDoux, 2005*), which could further lead to emotion-charged memory enhancement. According to our empirical observations, memory performance in the aversive condition showed only a trending positive correlation with hippocampal-amygdala connectivity, but highly positive correlations with hippocampal-mFFA/sLOC connectivity. Therefore, we assumed a potential mediatory pathway among these variables, that is, hippocampal-amygdala connectivity would indirectly account for emotion-charged memory performance through the mediation of hippocampal-mFFA/sLOC connectivity. Based on above theoretical motivation as well as empirical observations, we further implemented an exploratory mediation analysis to investigate how the hippocampus, amygdala and neocortical systems during emotional learning work in concert with each other to support emotion-charged retroactive memory enhancement.

Specifically, we constructed a mediation model with hippocampal-amygdala connectivity (i.e. remembered with high confidence relative to forgotten) as input variable, hippocampal-mFFA and -sLOC connectivity (i.e. remembered with high confidence vs. forgotten) as two separate mediators and individual's associative memory performance (i.e. remembered with high confidence) as outcome variable in the aversive condition (*Figure 3K*). This exploratory mediation analysis revealed two significant mediating effects on the positive relationship between hippocampal-amygdala connectivity and memory outcome (hippocampal couplings with mFFA and sLOC: Indirect Est = 0.19, p=0.010, 95% CI = [0.044, 0.328], and hippocampal–sLOC coupling: Indirect Est = 0.20, p=0.016, 95% CI = [0.036, 0.360]). That is, hippocampal-amygdala connectivity affected hippocampal-sLOC connectivity through a partial mediating effect of hippocampal-mFFA connectivity, which could ultimately account for emotion-charged memory performance in the aversive condition. Notably, we also conducted additional control analyses with a set of alternative mediation and modulation models in the aversive condition (*Figure 3—figure supplement 3*). These analyses, however, did not reveal any reliable mediating or modulating effect, indicating that our data did not support such alternative models. Since there was no any significant relationship between hippocampal connectivity and associative memory in the neutral condition, we did not conduct mediation analysis in this condition. Altogether, these exploratory results indicate that increased hippocampal-amygdala coupling might indirectly account for emotion-charged associative memory, likely mediating by hippocampal-neocortical couplings during emotional learning.

## Emotion-charged retroactive memory enhancement is associated with the reorganization of hippocampal connectivity during post-learning rests

Furthermore, to explore how offline hippocampal connectivity changes at resting states prior to and after emotional learning contribute to emotion-charged retroactive memory benefit, we implemented seed-based correlational analyses of resting-state fMRI data separately for three rest scans

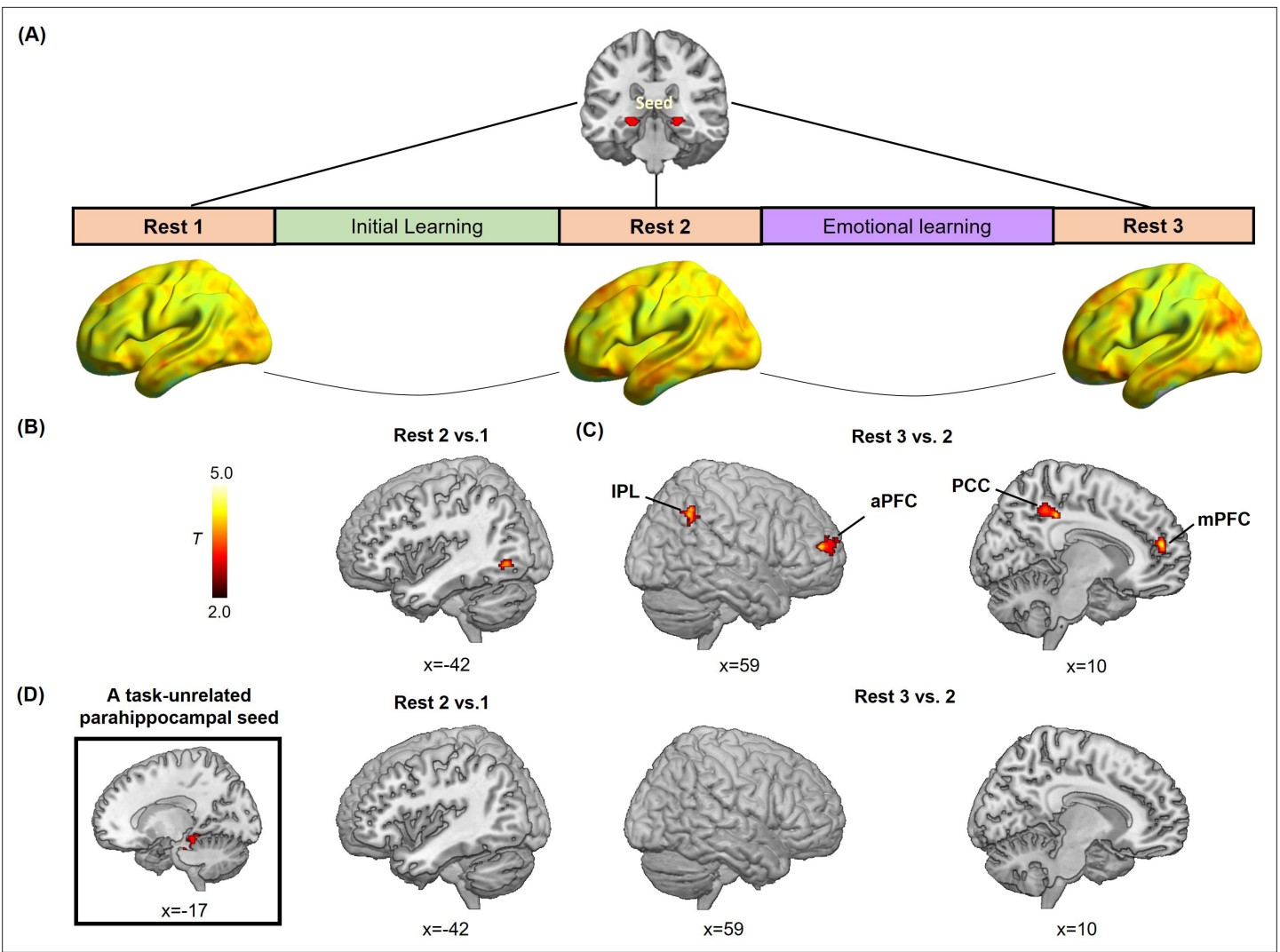

**Figure 4.** Hippocampal- and parahippocampal-neocortical connectivity at post-learning rests in relation to emotion-charged retroactive memory enhancement. (**A**) An illustration of hippocampal-seeded functional connectivity analyses, and sagittal views of hippocampal connectivity maps at the group level for three rest scans (n=27, data from one participant during Rest 2 was incomplete due to hardware malfunction). (**B**) Significant cluster in the left inferior LOC (iLOC), showing its greater connectivity with the hippocampus at Rest 2 (vs. Rest 1) in positive relation to memory in the aversive but not neutral condition. The iLOC ROI was constrained by the LOC mask, same with pattern similarity analysis. (**C**) Significant clusters in the bilateral inferior parietal lobule (IPL), the right anterior prefrontal cortex (aPFC), the right posterior cingulate cortex (PCC) and the right medial prefrontal cortex (mPFC), showing their greater connectivity with the hippocampus at Rest 3 (vs. Rest 2) in positive relation to memory in the aversive but not neutral condition. (**D**) A task-unrelated parahippocampal seed used in parallel control connectivity analyses (n=27), showing no reliable cluster under above-mentioned contrasts. Notes: Color bar represents T values.

The online version of this article includes the following figure supplement(s) for figure 4:

**Figure supplement 1.** Post-encoding hippocampal connectivity changes in relation to memory performances in aversive and neutral conditions.

that interleaved initial and emotional learning phases, with the hippocampal ROI as seed to assess its functional connectivity with the rest of the brain (*Figure 4A*). We computed the difference maps of hippocampal connectivity during Rest 2 relative to Rest 1 to reflect an initial-learning-related preparatory memory network primed to be engaged in subsequent emotional learning, and connectivity during Rest 3 relative to Rest 2 to reflect the following emotional-learning-related changes in the memory network. Thereafter, we conducted multiple regression analyses to compute these connectivity difference maps in relation to associative memory performance (i.e. remembered with high confidence) in aversive versus neutral conditions, which is analogous to the interaction effects between post-learning hippocampal connectivity changes (i.e. Rest 2 vs. 1 or Rest 3 vs. 2) and conditions

(i.e. aversive vs. neutral). For hippocampal connectivity changes after initial learning relative to baseline (i.e. Rest 2 vs. 1), we identified a significant cluster in the left inferior portion of LOC (iLOC) that was also constrained by the above-mentioned LOC mask with both anatomical and functional criteria (*Figure 4B*; *Supplementary file 4*). For connectivity changes after emotional learning relative to before (i.e. Rest 3 vs. 2), we identified significant clusters in a set of brain regions including the bilateral inferior parietal lobule (IPL) extending into the angular gyrus (AG), the right posterior cingulate cortex (PCC), the right anterior prefrontal cortex (aPFC) and the right medial prefrontal cortex (mPFC) (*Figure 4C*; *Supplementary file 4*). Specifically, the exploratory whole-brain analyses revealed that hippocampal-iLOC connectivity at post-initial-learning rest positively correlated with associative memory in the aversive condition, but negatively correlated with memory in the neutral condition (*see Figure 4—figure supplement 1A for visualization*). We also observed that hippocampal connectivity with transmodal prefrontal-parietal regions (i.e. IPL, PCC, aPFC and mPFC) at post-emotional-learning rest positively correlated with memory in the aversive condition, but negatively in the neutral condition (*see Figure 4—figure supplement 1B for visualization*). As a complement, the machine learning-based prediction analyses for post-learning hippocampal connectivity changes revealed very similar patterns of positive predictions for associative memory in the aversive condition (iLOC: $r_{(predicted,\ observed)}$=0.47; IPL: $r_{(predicted,\ observed)}$=0.54; PCC: $r_{(predicted,\ observed)}$=0.51; aPFC: $r_{(predicted,\ observed)}$=0.52; mPFC: $r_{(predicted,\ observed)}$=0.48; all p<0.005 while controlling for memory in the neutral condition), but negative predictions in the neutral condition (iLOC: $r_{(predicted,\ observed)}$=−0.45; IPL: $r_{(predicted,\ observed)}$=−0.55; PCC: $r_{(predicted,\ observed)}$=−0.50; aPFC: $r_{(predicted,\ observed)}$=−0.53; mPFC: $r_{(predicted,\ observed)}$=−0.51; all p<0.006 while controlling for memory in the aversive condition). These exploratory results indicate a potential shift of post-learning hippocampal connectivity from the object-sensitive lateral occipital complex to more distributed transmodal prefrontal and posterior parietal areas, which predicts emotion-charged retroactive memory benefit.

Additionally, we conducted parallel control analyses with a task-unrelated parahippocampal region as seed. This parahippocampal sensory region engaged in scene processing is not related to the stimuli used in our study. The control analyses revealed no reliable clusters survived for the task-unrelated parahippocampal connectivity during post-learning rests in relation with emotion-charged retroactive memory benefit (*Figure 4D*). It indicates that our observed offline hippocampal-neocortical functional reorganization is a special process involving with stimuli- and memory-related regions (i.e. iLOC, IPL, PCC, aPFC, and mPFC), but not with irrelevant brain regions (i.e. the task-unrelated parahippocampal region).

## Discussion

By three studies, we investigated the neurobiological mechanisms of how memories for associated mundane events are retroactively modulated by following learning of emotional events. As expected, emotional learning retroactively enhanced memory for initial neutral associations. This rapid retroactive enhancement was associated with increased trial-specific reactivation of initial learning activity in the hippocampus and stimulus-sensitive neocortex, as well as strengthened hippocampal coupling with the amygdala and neocortical regions during emotional learning. Complementally, hippocampal-amygdala coupling positively predicted the emotion-charged retroactive memory benefit, mediating by increased hippocampal-neocortical interactions. Moreover, we explored a potential shift of hippocampal-neocortical connectivity contributing to the emotion-charged retroactive memory enhancement during post-learning rests, from local stimulus-sensitive neocortex to more distributed transmodal prefrontal and posterior parietal regions. Our findings suggest that emotional learning retroactively promotes the integration for relevant neutral events into episodic memory to foster prediction of future events, through stimulating trial-specific reactivation of overlapping memory traces and reorganization of related memories with their updated values in an integrated network.

### Emotion-charged retroactive enhancement on memory integration for initial neutral events

Behaviorally, we observed that emotional learning retroactively enhanced associative memory for initial neutral events. This rapid memory enhancement occurred in the aversive rather than neutral condition, and appeared to be selective for face-object associations strongly encoded as indicated

by high-vividness ratings during initial learning. Notably, such enhancement effect was reproducible across two independent studies indicating its robustness and reliability. This is in line with findings from previous studies on sensory preconditioning, suggesting that a salient event such as fear could spread significant value to its associated mundane event and prioritize memory of this associated event for future use (*Li et al., 2008*; *Shohamy and Daw, 2015*; *Wimmer and Shohamy, 2012*). Beyond these studies focusing on item memory of indirectly associated event, our results provide novel evidence that the value of subsequent emotional event could also generalize into the initial associations of past neutral events, most likely through trial-specific reactivation of overlapping memory traces and reorganization of related memories into a more tightly integrated network.

Using an adapted sensory preconditioning paradigm, our observed emotion-charged retroactive memory benefit differs from the conventional behavioral tagging effects by three following aspects. First, such sensory preconditioning paradigm allows us to investigate trial-specific enhancement for each pair by associative learning task (*Wang and Kahnt, 2021*; *Wimmer and Shohamy, 2012*), in which the overlapping face cues were presented during both initial and emotional learning phases, rather than a general effect with non-overlapping stimuli in each phase (*Clewett et al., 2022*; *Dunsmoor et al., 2015*). Second, consistent with the sensory preconditioning models (*Kurth-Nelson et al., 2015*; *Sharpe et al., 2017*), we observed the retroactive memory enhancement in the immediate test, which differs from the delayed benefit pertaining to behavioral tagging literature (*Ballarini et al., 2009*; *Dunsmoor et al., 2012*; *Dunsmoor et al., 2011*). Third, our observed retroactive benefit was selective for strongly encoded associations as indicated by high-vividness ratings (presumably strong memories)(*Holmes et al., 2022*), which contradicts with the rescued effect for initial weak memories expected by behavioral tagging models (*Dunsmoor et al., 2015*; *Ritchey et al., 2016*; ). Neutral associations with high-vividness ratings were, at least temporarily, encoded into memory during the initial learning phase. Their memory traces could be subsequently reactivated by corresponding face cues, and further modulated by integrating with arousal during the emotional learning phase. However, associations with low-vividness ratings might be encoded relatively weaker. Since no specific memory traces were formed for these low-vividness associations, they could not be reactivated nor modulated later. It is thus conceivable that the emotion-charged trial-specific retroactive memory integration tends to occur on relatively strong associations. Together, our findings suggest that emotional learning with autonomic arousal occurring during reactivation of initial overlapping memory traces, may stimulate a more targeted and rapid memory reorganization through an integrative encoding mechanism.

## Emotion-charged trial-specific reactivation of overlapping neural traces in the hippocampus and neocortex

In parallel with above-described rapidly and selectively retroactive memory enhancement, our imaging results showed transient increases in trial-specific reactivation of initial learning activity in the aversive (vs. neutral) condition during the emotional learning phase. Firstly, our results from condition-level similarity analyses between initial and emotional learning phases demonstrate greater reactivation occurred after the onset of aversive (vs. neutral) voices. Such effect might result from emotion-induced autonomic arousal accompanying with elevated catecholamine release, that rapidly potentiates the excitability of overlapping neuronal ensembles and strengthens the reactivation of initial memory (*Wong et al., 2019*; *Zhou et al., 2009*). More interestingly, results from our trial-level similarity analyses between phases further suggest that such emotion-charged increased reactivation mostly reflect the trial-specific reinstatement rather than a board category representation. Indeed, compared with the general predictive effect of category-level representation on overall memory regardless of emotional conditions, such trial-specific reinstatement positively predict memory performance for initial neutral associations only in the aversive rather than neutral condition, suggesting a specificity of emotional modulation on trial-specific reinstatement but not on category-level representation. Based on memory allocation and integration models, reactivation of overlapping neuronal ensembles may serve as a mechanism by which memories for related prior and present experiences can be allocated into an integrated network of representations (*Schlichting and Frankland, 2017*; *Silva et al., 2009*). By this view, autonomic arousal triggered by emotional learning may stimulate trial-specific reactivation of initial encoding activity, which could strengthen the specific association between prior memories for neutral information in the integrated memory network. Moreover, we also computed

within-phase similarities to measure the consistency of activity pattern during initial and emotional learning phases separately. However, there was no significant relationship between within-phase similarities and memory performance. These results reveal that our observed emotion-charged retroactive memory enhancement does not result from the consistency of activity patterns.

Critically, we found the prominent emotional enhancement for trial-specific reactivation (i.e. pair-specific similarity) in the hippocampus and vLOC, but not in the FFA. The hippocampus is known to play a key role in the integration of information processed in distributed neocortical regions (i.e. LOC and FFA) into episodic memory through a pattern completion process (*Kuhl et al., 2010*; *Shohamy and Wagner, 2008*). Thus, hippocampal reactivation may strengthen the coherence of related representations throughout our brain (*Staudigl and Hanslmayr, 2018*; *Wimmer and Shohamy, 2012*). Consistently, we indeed observed a positive correlation between emotion-charged trial-specific reactivation in the hippocampus and memory for face-object associations. The object-sensitive vLOC showed an increase in trial-specific reactivation of object stimuli, which might reflect reinstatement of corresponding initial face-object associations during emotional learning (*Hofstetter et al., 2012*; *Tambini et al., 2010*). But the face-sensitive FFA could not provide pure evidence of reactivation since face information was presented during both phases. These results suggest task-related regional specificity of the observed emotion-charged trial-specific reactivation. In addition, the trending but not significant emotional effects for category representation (i.e. across-pair similarity) found in the hippocampus, vLOC and FFA might be due to more attention in the aversive (vs. neutral) condition generally attracted by the screaming stimuli, rather than trial-specific reactivation. This explanation is also supported by our condition-level similarity results, showing strengthened reactivation only occurred after the onset of aversive (vs. neutral) voices during emotional learning. Altogether, our findings suggest that emotional learning may induce transient increases in trial-specific reactivation of overlapping neural traces in the hippocampus and stimulus-sensitive neocortical regions (i.e. vLOC). This emotion-charged trial-specific reactivation shows emotional and regional specificity in relation to memory integration, which may promote a rapid memory reorganization of related events.

## Emotion-charged memory reorganization via hippocampal-amygdala-neocortical interactions

Coinciding with transient increases in trial-specific reactivation, our results further suggested an emotion-charged memory reorganization by stimulating functional connectivity among the hippocampus, amygdala, and neocortical circuits during emotional learning, as well as a potential shift of hippocampal-neocortical connectivity from local stimulus-sensitive occipital area to more distributed prefrontal and posterior parietal systems during post-learning resting state.

Four aspects of our data support this interpretation. First, we observed emotion-induced increases in hippocampal functional connectivity with the amygdala and face/object-sensitive neocortical regions (i.e. mFFA and sLOC) in the aversive (vs. neutral) condition during emotional learning, but not initial learning, phase. Second, although such hippocampal connectivity patterns did not show reliable Emotion-by-Memory interaction effect, results from our prediction analyses revealed that hippocampal connectivity with the neocortical regions during emotional learning positively predicted memory for face-object associations in the aversive rather than neutral condition. Consistent with emotion-induced increases in trial-specific reactivation, these results point toward the specificity of emotional modulation on the relationships between hippocampal connectivity and associative memory performance. Third, we explored that increased hippocampal-neocortical coupling could mediate the positive relationship between hippocampal-amygdala coupling and emotion-charged retroactive memory enhancement. This exploratory mediation effect suggests that emotional arousal directly paired with face cues could induce greater hippocampal-amygdala connectivity acting on hippocampal-FFA connectivity during emotional learning, which then stimulated the cued trial-specific reactivation in the hippocampus and LOC through increasing hippocampal-LOC interaction, and ultimately contributed to retroactive memory benefit for related events. These results are in line with previous studies that the modulatory role of amygdala could support more efficient information transmission and communication between the hippocampus and related neocortical regions (i.e. the FFA and LOC) (*Hamann, 2001*; *Hermans et al., 2014*), and hippocampal–neocortical functional coordination plays a critical role in reactivation and integration of episodic memories (*Kuhl et al., 2010*; *Schlichting and Preston, 2014*; *Sutherland and McNaughton, 2000*; *Wimmer and Shohamy, 2012*).

Thus, the observed emotion-charged hippocampal-amygdala-neocortical interactions are most likely to link with rapid trial-specific reactivation of past neutral experiences and reorganize their memories into a network with integrated representations. Our findings demonstrate a mechanism of emotion-induced memory reorganization via strengthened hippocampal–neocortical functional coupling, coinciding with the modulation of amygdala activity.

Last but not least, we explored a potential hippocampal-neocortical functional reorganization during post-learning rests predictive of emotion-charged retroactive memory benefit, with a shift away from local stimulus-sensitive LOC to more distributed prefrontal and posterior parietal regions including the aPFC, mPFC, PCC and IPL expanding into the AG. Initial learning elicited greater hippocampal-iLOC connectivity during post-initial-learning rest, which might reflect a possible increase in neural excitability of this circuit. Since neuronal excitability in the hippocampal-neocortical circuitry prior to encoding is recognized to modulate subsequent allocation and integration of newly acquired information into long-term memory (*Josselyn and Frankland, 2018*; *Kaefer et al., 2022*; *Schlichting and Frankland, 2017*; *van Dongen et al., 2011*; *Yoo et al., 2012*), we thus speculate that greater hippocampal-iLOC connectivity might provide a preparatory state to allocate new information into existing memory traces during emotional learning and thus contribute to emotion-charged retroactive memory benefit. Besides, neural circuits activated by emotional learning exhibit persistent activity and also alter a series of systems-level interactions during post-emotional-learning rest (*de Voogd et al., 2016*; *Hermans et al., 2017*; *Murty et al., 2017*), including hippocampal connectivity with aPFC, mPFC, PCC, and IPL observed in this study. These transmodal prefrontal-parietal regions are core nodes of the default mode network (DMN) that is recognized to play a crucial role in remembering past events and simulating possible future use (*Andrews-Hanna et al., 2010*; *Schacter et al., 2011*; *Spreng and Schacter, 2012*). Thus, our observed offline hippocampal-neocortical connectivity changes may contribute to emotion-charged retroactive memory benefit, likely through functional reorganization mechanisms of memory-related brain systems at both pre- and post-emotional-learning phases. Interestingly, hippocampal-neocortical connectivity changes involved in this reorganization positively correlated with memory in the aversive condition, but negatively correlated in the neutral condition (i.e. partial correlations controlling for memory in the other condition). It is possible to speculate that the emotion-charged retroactive memory benefit might not only reflect an enhancement of emotion-related information but also a suppression of other neutral information. These findings point toward a potential trade-off between prioritization of emotion-related memory and neglection of mundane neutral memory, aligning with the activity of locus coeruleus-norepinephrine system during post-encoding periods (*Clewett and Murty, 2019*).

Taken together, our findings suggest that emotional learning not only rapidly strengthens online task-dependent hippocampal-neocortical coupling through amygdala modulation, but it also potentiates offline hippocampal-neocortical reorganization integrating memory traces into more distributed networks and prioritizing them for future use. In line with emotion-charged retroactive memory benefit and increased trial-specific reactivation for associated events, our observed hippocampal connectivity changes during emotional learning and post-learning rests most likely reflect a relational process underlying memory integration, but not an indiscriminate generalization or persistence of emotional arousal (*Hermans et al., 2017*; *Tambini and Davachi, 2013*; *Tambini et al., 2010*). It points toward a rapid emotion-modulated reorganization mechanism, by which memories for neutral events can be updated according to the significance of subsequent emotional events. This emotion-modulated reorganization not only contributes to the integration of past and current experiences into episodic memory, but also supports future simulations.

## Limitations

Although our study provides converging evidence of rapid neural reactivation and connectivity reorganization underlying emotion-charged retroactive benefit on memory integration, several limitations should be considered. First, given our experimental design including an initial learning followed by an emotional learning and a surprise memory test, it is thus possible that subsequent emotional learning might overwrite the effect of initial learning on final memory performance. This could account for no reliable correlation of post-initial-learning hippocampal connectivity changes (i.e. Rest 2 vs. Rest 1) with either neutral or average memory performance. Future studies are required to test memory before emotional learning to disentangle post-learning signatures linked to pure memory effect for

neutral events and emotion-charged memory effect separately. Second, our moderate sample size in fMRI Study 3 would be underpowered to detect individual differences in across-subject correlations and mediation analyses. A larger sample size would reinforce the reproducibility of brain-behavior correlations in future studies. Third, our functionally defined ROIs in the hippocampal connectivity analysis during the emotional learning phase may bring potential selection bias. Future studies with anatomical ROIs would help mitigate this issue. Forth, it is challenging to reliably separate neural signals associated with 'face cues' and 'face-voice associations' due to only an interval of 2 s. Future design with longer and jittered intervals may resolve this issue.

## Conclusion

Our study demonstrates that emotional learning can retroactively promote memory integration for preceding neutral events through an emotion-modulated rapid reorganization mechanism, characterized by transient increases in trial-specific reactivation of overlapping neural traces, strengthened hippocampal-neocortical coupling modulated by the amygdala during emotional learning, and a shift of hippocampal-neocortical connectivity from local stimulus-sensitive neocortex to distributed prefrontal and posterior parietal areas during post-learning rests. Our findings across three independent studies advance the understanding of neurobiological mechanisms by which emotion can reshape our episodic memory of previous neutral events to foster its priority for future use, and also provide novel insights into maladaptive generalization in mental disorders like PTSD.

## Methods

### Participants

A total of 89 young, healthy college students participated in three separate studies. In Study 1, 30 participants (16 females; mean age ±s.d., 22.23±2.05 years old, ranged from 18 to 26 years) were recruited from Beijing area to participate in a behavioral experiment. In Study 2 by an independent research staff, 28 participants (14 females; mean age ±s.d., 21.83±1.93 years old, ranged from 18 to 26 years) were recruited from Xinyang city in Henan province for a replication experiment to ensure the reliability of our behavioral findings from Study 1. In Study 3, another independent cohort of 31 participants (17 females; mean age ±s.d., 22.55±2.25 years old, ranged from 18 to 27 years) was recruited from Beijing area to participate in an event-related fMRI experiment. Data from three participants were excluded from further analyses due to either falling asleep during fMRI scanning (n=2) or poor memory performance (i.e. the overall memory accuracy across confidence ratings and conditions was almost 0; n=1). The sample sizes across three studies were estimated by a power analysis using G*Power 3.1, which yielded the power of around 85–90% (i.e. from 26 to 30 participants) for a moderate repeated-measures ANOVA, consistent with many memory studies (*Gruber et al., 2016*; *Liu et al., 2016*; *Meyer and Benoit, 2022*; *Schlichting and Preston, 2014*; *Wimmer and Shohamy, 2012*). In Study 3, the sample size of 28 valid participants could give us power more than 70% for a moderate correlation (i.e. *r*>0.45) (*Cohen, 1992*; *Cohen, 2013*).

All participants were right-handed with normal hearing and normal or corrected-to-normal vision, reporting no history of any neurobiological diseases or psychiatric disorders. Informed written consent was obtained from each participant before the experiment. The Institutional Review Board for Human Subjects at Beijing Normal University (ICBIR_A_0098_002), Xinyang Normal University (same as above) and Peking University (IRB#2015-09-04) approved the procedures for Study 1, 2, and 3, respectively.

### Materials

One hundred and forty-four face images (72 males and 72 females) were carefully selected from a database with color photographs of Chinese individuals unknown to participants (*Chen et al., 2012*), under following criteria suggested by previous studies: direct gaze contact, no headdress, no glasses, no beard, etc (*Qin et al., 2007*). There was also no strong emotional facial expression in these faces, and no significant difference in terms of arousal, valence, attractiveness, and trustworthiness between male and female faces according to rating results in a previous study (*Liu et al., 2016*). Seventy-two object images were obtained from a website (http://www.lifeonwhite.com) or publicly available resources on the internet (*Dunsmoor et al., 2015*). All objects were common in life with neutral valence. Four short clips of female voices were carefully selected from an audio source

website (https://www.smzy.com/), with two aversive screams serving as emotional arousal manipulation and two neutral voices (i.e. 'Ah' and 'Eh') as control. Acoustic characteristics of the four voice clips were measured and controlled using Praat ( http://www.praat.org/), including duration (2 s), frequency (in Hertz) and power (in decibel) (*Supplementary file 1*). An independent cohort of 19 participants (10 females; mean age ±s.d., 21.90±2.35 years old, ranged from 18 to 26 years) was recruited from local area to participate in a pilot experiment, which was performed to rate the four voices before the formal memory studies. Participants were instructed to listen to each voice, and then rate valence and arousal separately for the voice on a 9-point self-rating manikin scale (i.e. 9 = 'Extremely pleasant' or 'Extremely arousing', 1 = 'Not pleasant at all' or 'Not arousing at all'). Two aversive screams had significantly higher arousal and lower valence than neutral voices (all p<0.001; *Figure 1—figure supplement 1*). There was no significant difference in arousal or valence between two aversive (or neutral) voices (all p>0.05; *Figure 1—figure supplement 1*).

Faces were randomly split into two sets of 72 images with half male and half female faces for each participant: one list was paired with objects to create 72 face-object pairs for the initial learning phase, and the other was served as foils for face recognition memory test [2 (Emotion: aversive vs. neutral) by 2 (Confidence: high vs. low)] repeated-measures ANOVAs on face item memory revealed neither main effect of Emotion (Study 1: $F_{(1, 29)}$=3.53, p=0.070; Study 2: $F_{(1, 26)}$=0.001, p=0.970; Study 3: $F_{(1, 27)}$=0.09, p=0.765) nor Emotion-by-Confidence interaction effect (Study 1: $F_{(1, 29)}$=1.75, p=0.196; Study 2: $F_{(1, 26)}$=0.62, p=0.439; Study 3: $F_{(1, 27)}$=1.08, p=0.307), but only significant main effect of Confidence (Study 1: $F_{(1, 29)}$=141.11, p<0.001, partial $\eta^2$=0.83; Study 2: $F_{(1, 26)}$=133.29, p<0.001, partial $\eta^2$=0.84; Study 3: $F_{(1, 27)}$=40.82, p<0.001, partial $\eta^2$=0.60). These results indicate that the modulatory effect of emotional learning did not present on face item memory, but might directly enhance face-object associative memory through increasing reactivation (*see Figure 1—figure supplement 2 for details*). Then, each face in face-object pairs was randomly paired with one of four voices to create 72 face-voice pairs and assigned into either an aversive condition (i.e. 36 faces paired with aversive screams) or a neutral condition (i.e. 36 faces paired with neutral voices) during the emotional learning phase.

## Experimental procedures

In each of the three studies, the experimental design consisted of three consecutive phases: an initial learning, a followed-up emotional learning, and a surprise recognition memory test (*Figure 1A*). During the initial learning phase, participants were instructed to view 72 face-object pairs in an incidental encoding task. During the emotional learning phase, faces from the initial learning phase were presented again as cues and paired with either an aversive or a neutral voice. After a 30-min delay, participants performed a surprise recognition memory test for face-object associations. In Study 3, participants underwent fMRI scanning with concurrent recording of skin conductance while they were performing initial learning and emotional learning phases interleaved by three rest scans. *Specifically*, the fMRI experiment began with an 8 min baseline rest scan (i.e. Rest 1), followed by the initial and emotional learning phases. Each of the two learning phases was followed by another 8 min rest scan (i.e. Rest 2 and 3). During rest scans, participants were shown a black screen and instructed to keep awake with their eyes open. Finally, the surprise associative memory test was performed outside the scanner.

### Initial learning task

During initial learning, 72 faces were randomly paired with 72 objects to create 72 face-object associations for each participant. Each association was centrally presented on the screen for 4 s, and followed by a vividness rating scale for 2 s. To ensure incidental memory encoding, participants were instructed to vividly imagine each face interacting with its paired object and give a vividness rating on their imagined scenario on a 4-point Likert scale (i.e. 4 = 'Very vivid', 1 = 'Not vivid at all'). Trials were interleaved by a fixation with an inter-trial interval jittered from 2 to 6 s (i.e. 4 s on average with 2 s step). The total of 72 face-object pairs were viewed twice, which were split into two runs with 12 min each.

### Emotional learning task

During emotional learning, each face from initial learning was presented at the center of screen for 2 s, and then followed by concurrent presentation of the same face paired with either an aversive or a neutral voice for another 2 s. To ensure the consistency with initial incidental learning task, participants

were again instructed to imagine the face interacting with its paired voice and give their vividness rating on a 4-point Likert scale for 2 s. After that, each trial was followed by a relatively long inter-trial interval jittered from 6 to 10 s (i.e. 8 s on average with 2 s step), to reduce potential contamination of voice-induced emotional arousal among neighboring trials. Totally 72 face-voice pairs were presented only once in a pseudo-randomized order that no more than 2 voices from the same condition (i.e. aversive or neutral condition) appeared in a row. The emotional learning phase lasted 16.8 min.

### Surprise recognition memory test

After a 30-min delay, participants were instructed to perform a self-paced recognition memory test for face-object associations. Each trial consisted of four faces from initial learning on the top of screen and their corresponding objects randomly located on the bottom of screen. A total of 72 learned face-object pairs were randomly sorted into 18 slides with 4 pairs each. Participants were asked to pair each face with one of the four objects according to their remembrance of face-object associations, and then gave their confidence rating for each pair separately on a 4-point scale (i.e. 4 = 'Very confident', 1 = 'Not confident at all'). Participants were required to make choice for each face in an order from left to right, and carefully recall before they made the choice to avoid errors.

## Behavioral data analysis

Participants' behavioral performances on vividness rating in the initial learning phase (i.e. the second run/viewing), memory accuracy and confidence rating in the memory test were analyzed using Statistical Product and Service Solutions (SPSS, version 22.0, IBM) and R (version 4.2.1). Two conditions were created according to emotional learning manipulations. During initial learning, trials for face-object associations subsequently paired with aversive screams were assigned into the aversive condition, and remaining trials with neutral voices were assigned into the neutral condition. During emotional learning, trials paired with aversive screams were assigned into the aversive condition, and trials with neutral voices were assigned into the neutral condition. One sample t-tests were conducted to examine the reliability of face-object associative memory performance as compared to the chance level for each confidence level (i.e. from 1 to 4). Given that participants were required to match 4 sets of face-object pairings within one screen at a time in the memory test, the chance level of associative memory performance was 52% calculated using the mean accuracy of 4 pairings [(1/4+1/3+1/2+1)/4*100%]. Then, all remembered associations were sorted into a high-confident bin with levels 3 and 4 reflecting reliable memory, and a low-confident bin with levels 1 and 2 reflecting guessing or familiarity (*Squire et al., 2007*). Separate 2 (Emotion: aversive vs. neutral) by 2 (Confidence: high vs. low) repeated-measures ANOVAs were conducted in the three studies to examine the emotion-charged effect on reliable memory performance rather than guessing or familiarity. Moreover, trial-level relationship between vividness ratings during initial learning and confidence ratings for final memory performance was tested by a linear mixed-effects model, in order to examine how much distinct effect the initial encoding strength of face-object associations (as indicated by vividness rating) would contribute to subsequent emotional memory benefit (*Gałecki and Burzykowski, 2013*; *Singh et al., 2021*). These remembered associations were further sorted into a high-vividness bin with levels 3 and 4, and a low-vividness bin with levels 1 and 2. We conducted paired t-tests between aversive and neutral conditions on memory accuracy with high-vividness and low-vividness ratings separately, to investigate how initial memory strength modulates our observed emotion-charged effect. Given the robust results were all found in high-confidence memory performance across the three studies, we thus used the high-confidence memory as a reliable measure of memory performance in the following reactivation and connectivity analyses.

## Skin conductance recording and analysis

Skin conductance was collected to assess autonomic arousal induced by aversive screaming (vs. neutral) voices during the emotional learning. It was recorded simultaneously with fMRI scanning using a Biopac MP 150 System (Biopac, Inc, Goleta, CA). Two Ag/AgCl electrodes filled with isotonic electrolyte medium were attached to the center phalanges of the index and middle fingers of each participant's left hand. The gain set to 5, the low-pass filter set to 1.0 Hz, and the high-pass filters set to DC (*Indovina et al., 2011*). Data were acquired at 1000 samples per second and transformed into microsiemens (µS) before further analyses. Given the temporal course of skin conductance in response

to certain event, mean skin conductance levels (SCLs) were calculated for a period of 6 s after each stimulus onset.

## Imaging acquisition

Whole-brain imaging data were collected on a 3T Siemens Prisma MR scanner (Siemens Medical, Erlangen, Germany) with a 20-channel head coil system at Peking University in Beijing, China. Functional images were collected using a multi-band echo-planar imaging (mb-EPI) sequence (slices, 64; slice thickness, 2 mm; TR, 2000 ms; TE, 30ms; flip angle, 90°; multiband accelerate factor, 2; voxel size, 2×2×2 mm; FOV, 224×224 mm; 240 volumes for each of the three rest scans, 365 and 508 volumes for the initial and emotional learning scans separately). To correct for distortions, field-map images (i.e. magnitude and phase images) were acquired (slices, 64; slice thickness, 2 mm; TR, 635ms; TE1, 4.92ms; TE2, 7.38ms; flip angle, 60°; voxel size, 2×2×2 mm; FOV, 224×224 mm). Structural images were acquired through three-dimensional sagittal T1-weighted magnetization-prepared rapid gradient echo (MPRAGE) sequence (slices, 192; slice thickness, 1 mm; TR, 2530ms; TE, 2.98ms; flip angle, 7°; inversion time, 1100ms; voxel size, 1×1×1 mm; FOV, 256×256 mm).

## Imaging preprocessing

Brain imaging data was preprocessed using Statistical Parametric Mapping (SPM12; http://www.fil.ion.ucl.ac.uk/spm). The first 4 volumes of functional images were discarded for signal equilibrium. Remaining images were firstly corrected for distortions related to magnetic field inhomogeneity. Subsequently, these functional images were realigned for head-motion correction and corrected for slice acquisition timing. Each participant's images were then co-registered to their own T1-weighted anatomical image, spatially normalized into a standard stereotactic Montreal Neurological Institute (MNI) space and resampled into 2 mm isotropic voxels. Finally, images were smoothed with a 6 mm FWHM Gaussian kernel. A high-pass filter (1/128 Hz cutoff) was also applied to remove low-frequency signal drifts.

## Regions of interest (ROIs) definition

To investigate the emotion-charged reactivation in both condition- and trial-level pattern similarity analyses, we identified the bilateral hippocampal, bilateral ventral LOC (vLOC) and bilateral FFA ROIs by the overlapping area of two group-level univariate activation contrasts of face-object association encoding (i.e. initial learning) and face-voice association encoding (i.e. emotional learning) separately relative to fixation (i.e. all encoding trials vs. fixation during each learning phase), using a stringent height threshold of p<0.0001 and an extent threshold of p<0.05 with family-wise error correction for multiple comparisons based on nonstationary suprathreshold cluster-size distributions computed by Monte Carlo simulations (*Miller et al., 2022*; *Nichols and Hayasaka, 2003*; *Figure 2—figure supplement 1*). The vLOC was further constrained by a LOC mask, which is the overlapping area of the combined anatomical automatic labeling (AAL) template including the 'bilateral middle occipital cortex', 'bilateral middle temporal cortex', and 'bilateral fusiform gyrus' from the WFU PickAtlas toolbox (*Grill-Spector et al., 2001*; *Kourtzi and Kanwisher, 2001*; *Malach et al., 1995*), and the mask derived from the Neurosynth platform for large-scale, automated synthesis of fMRI data (http://neurosynth.org/) with 'object recognition' as a searching term (p<0.01 with FDR correction). The FFA ROI was further constrained by a FFA mask, which is the overlapping area of the AAL template of 'bilateral fusiform gyrus' from the WFU PickAtlas toolbox (*Kanwisher et al., 1997*) and the mask from the Neurosynth platform with 'face recognition' as a searching term (p<0.01 with FDR correction).

The above-defined hippocampal ROI was also used as a seed to further investigate emotion-induced changes in hippocampal functional connectivity. For task-dependent hippocampal connectivity analysis during initial and emotional learning phases, the right amygdala, left middle portion of FFA (mFFA) (*Visconti di Oleggio Castello et al., 2021*) and left superior portion of LOC (sLOC) (*Barbieri et al., 2019*; *Olivo et al., 2019*) ROIs were defined using a group-level connectivity contrast of the aversive relative to neutral condition during the emotional learning phase, by a height threshold of p<0.005 and an extent threshold of p<0.05 with family-wise error correction for multiple comparisons based on nonstationary suprathreshold cluster-size distributions computed by Monte Carlo simulations (*Nichols and Hayasaka, 2003*). For post-encoding hippocampal connectivity analysis during three resting phases, significant clusters in the left inferior portion of LOC (iLOC) (*Barbieri et al.,*

2019; *Olivo et al., 2019*) as well as the bilateral IPL, right PCC, right aPFC and right mPFC were derived from the group-level multiple regression analyses on connectivity contrast maps (i.e. Rest 2 vs. 1, and Rest 3 vs. 2) with interaction effects between aversive and neutral conditions, by the same threshold criterion as task-dependent connectivity analysis above. Same with pattern similarity analysis, the mFFA, sLOC and iLOC ROIs were further constrained separately by the above-mentioned FFA or LOC mask.

A task-unrelated parahippocampal region, as a seed in parallel control post-encoding connectivity analyses, was defined by the bilateral posterior parahippocampal gyrus using the WFU PickAtlas toolbox. However, the posterior parahippocampal area is not only a key sensory region for scene processing, but also involved in the encoding of associations into an integrated representation (*Qin et al., 2007*). To avoid the interference of its association-encoded function, we further restricted the posterior parahippocampal area by removing its overlapping area with two group activation contrasts of encoding during initial and emotional learning separately relative to fixation (i.e. all face-object/ face-voice encoding trials vs. fixation during each learning phase), by a relatively less stringent height threshold of $p<0.01$ and an extent threshold of $p<0.05$ with family-wise error correction for multiple comparisons based on nonstationary suprathreshold cluster-size distributions (*Nichols and Hayasaka, 2003*).

## ROI-based pattern similarity analysis

To measure the neural reactivation of initial learning activity for face-object associations cued by the overlapping faces during emotional learning, we computed multi-voxel pattern similarity of stimulus-evoked activation between initial and emotional learning phases in each ROI. We analyzed the pattern similarity on condition level to determine an overall emotional effect of reactivation by comparing aversive and neutral conditions. Two separate GLMs were conducted for the initial learning phase and the emotional learning phase. In each GLM, two regressors of interest were modeled for trials with 2 s from onset of each stimulus (i.e. face-object associations during initial learning or face-voice associations during emotional learning) in aversive and neutral conditions, and convolved with the canonical hemodynamic response function (HRF) in SPM12. In the GLM of emotional learning phase, the 2 s presentations of face cues prior to the onset of face-voice associations in aversive and neutral conditions were also included as two separate regressors of no interest. Additionally, each participant's motion parameters from the realignment procedure were included in each GLM to regress out effects of head movement on brain response. Then, *t* values of spatial activation maps in each ROI for aversive and neutral conditions in the initial learning phase and the emotional learning phase (i.e. during the presentation of face cues and the following presentation of face-voice associations separately) were extracted into separate vectors. We eliminated the mean activation from each pattern by z-scoring across voxels of each ROI. Thus, the resulting mean-centered pattern has relative voxel amplitudes (i.e. voxel-level variability) preserved without any difference in the overall amplitudes, ensuring that the subsequent similarity results are fully attributable to the pattern itself (*Coutanche, 2013*; *Tompary and Davachi, 2017*). Similarity between vectors of initial and emotional learning phases was computed for each condition and presentation using Pearson's correlation, and then Fisher-transformed. The resultant similarity values for each ROI were entered into a 2 (Emotion: aversive vs. neutral) by 2 (Presentation: face cue vs. face-voice association) repeated-measures ANOVA.

To further examine whether the emotional effect of reactivation is contributed by the trial-specific reinstatement or a broad category representation, we conducted a set of trial-level multi-voxel pattern similarity analyses. We modelled each trial with 2 s from onset of the stimulus (i.e. face-object association collapsing across two repetitions during initial learning or face-voice association during emotional learning) as a separate regressor, convolved with the canonical hemodynamic response function (HRF) in SPM12. In the GLM of emotional learning phase, the presentation of all face cues with 2 s duration was also included as a regressor of no interest. Other procedures were the same with GLMs in the condition-level analysis above. This resulted in 72 regressors in the GLM of initial learning phase and 73 regressors in the GLM of emotional learning phase. *T* values of spatial activation pattern in each ROI for each trial were extracted into a separate vector and z-scored. Thereafter, similarity between vectors of initial and emotional learning phases was also computed for each condition and measure in each ROI using Pearson's correlation, and then Fisher-transformed.

We computed three measures of trial-level pattern similarity in aversive and neutral conditions separately (*Figure 2—figure supplement 3*): **(1) pair-specific similarity** (i.e. correlation between each face-voice pattern and its corresponding face-object pattern), **(2) across-pair within-condition similarity** (i.e. average correlation between each face-voice pattern and all other different face-object patterns within the same aversive/neutral condition) and **(3) across-pair between-condition similarity** (i.e. average correlation between each face-voice pattern and all other different face-object patterns from the different condition). We then conducted separate 2 (Emotion: aversive vs. neutral) by 3 (Measure: pair-specific vs. across-pair within-condition vs. across-pair between-condition) repeated-measures ANCOVAs in each ROI, with individual's univariate activation differences (i.e. aversive vs. neutral) in both initial and emotional learning phases as covariates of no interest (*see Figure 2—figure supplement 4 for statistics*).

To directly examine whether across-pair pattern similarity measures for within- and between-condition pairs show different effects, we further conducted separate 2 (Emotion: aversive vs. neutral) by 2 (Measure: across-pair within-condition vs. across-pair between-condition) repeated-measures ANCOVAs for each ROI, with above-mentioned covariates of no interest. This analysis revealed only a trending but non-significant main effect of Emotion (hippocampus: $F_{(1, 25)}=4.16$, $p=0.052$, partial $\eta^2=0.14$; vLOC: $F_{(1, 25)}=3.37$, $p=0.078$, partial $\eta^2=0.12$; FFA: $F_{(1, 25)}=3.96$, $p=0.058$, partial $\eta^2=0.14$), and neither main effect of Measure nor Emotion-by-Measure interaction effect for each ROI (all $F_{(1, 25)}<2.00$, $p>0.170$; *Figure 2—figure supplement 4*). These results indicate no reliable emotional modulation effect on both across-pair similarity measures, and no significant difference between values of the two across-pair similarities. We thus combined these two across-pair similarities (i.e. averaging all correlations in both across-pair within- and between-condition) to quantify a generally category-level representation pattern. To better characterize the relationships between trial-specific reinstatement (i.e. pair-specific similarity) and category-level representation (i.e. across-pair similarity), we conducted separate 2 (Emotion: aversive vs. neutral) by 2 (Measure: pair-specific vs. across-pair) repeated-measures ANCOVAs in each ROI, also with individual's univariate activation differences (i.e. aversive vs. neutral) in both initial and emotional learning phases as covariates of no interest to further mitigate the potential interference of overall activation from pattern results.

To investigate whether the consistency of activity pattern within each phase also accounts for the emotion-induced memory benefit, we computed two other trial-level pattern similarity measures in aversive and neutral conditions separately (*Figure 2—figure supplement 3*): **(1) within-encoding similarity** (i.e. average correlation among face-object patterns within the initial learning phase), and **(2) within-arousal similarity** (i.e. average correlation among face-voice patterns within the emotional learning phase). These two within-phase similarity measures were computed in a same approach as between-phase similarity measures above. Thereafter, we conducted machine learning-based prediction analyses of these two similarity measures with associative memory performance.

## Whole-brain pattern similarity analysis

A searchlight mapping method was implemented to assess the reactivation of initial learning activity during emotional learning on the whole-brain level. Similar to above ROI-based analysis on condition level, we computed multi-voxel pattern similarity between initial and emotional learning phases for aversive and neutral conditions separately in each searchlight, using a 6 mm spherical ROI centered on each voxel across the whole brain. The resultant Fisher-transformed searchlight maps for two conditions were then entered into a paired-t test (i.e. aversive vs. neutral) on the group-level analysis to determine other brain regions involved in emotion-induced increased reactivation. Significant clusters were identified from the group analysis using a height threshold of $p<0.005$ and an extent threshold of $p<0.05$ with family-wise error correction for multiple comparisons based on nonstationary suprathreshold cluster-size distributions computed by Monte Carlo simulations (*Nichols and Hayasaka, 2003*).

## Task-dependent functional connectivity analysis

To assess hippocampus-based functional connectivity associated with emotional learning, we conducted a generalized form of task-dependent psychophysiological interaction (gPPI) analysis during initial and emotional learning phases separately (*Friston et al., 1997*; *Xiong et al., 2021*). This analysis examined condition-specific modulation on functional connectivity of a specific seed (i.e. the

hippocampal ROI here) with the rest of the brain, after removing potentially confounding influences of overall task activation and common driving inputs. The physiological activity of given seed region was computed as the mean time series of all voxels. They were then deconvolved to estimate neural activity (i.e. physiological variable), and multiplied with the task design vector by contrasting aversive and neutral conditions (i.e. psychological variable) to form a psycho-physiological interaction vector. This interaction vector was convolved with a canonical HRF to form the PPI regressor of interest. The psychological variable representing the task conditions (i.e. aversive and neutral conditions) was also included in the GLM to remove out the effects of common driving inputs on brain connectivity. Contrast images corresponding to PPI effect (i.e. aversive vs. neutral) at the individual-subject level were then entered into the group-level analysis. Significant clusters were identified from the group analysis using a height threshold of $p<0.005$ and an extent threshold of $p<0.05$ with family-wise error correction for multiple comparisons based on nonstationary suprathreshold cluster-size distributions computed by Monte Carlo simulations (*Nichols and Hayasaka, 2003*).

To further investigate the effect of emotion-charged hippocampal functional connectivity on associative memory, we conducted an additional hippocampal-seeded gPPI analysis only during the emotional learning phase by taking Memory Status (i.e. forgotten vs. remembered with high confidence) into account. All procedures were same as the above gPPI analysis, except that we included four PPI regressors of interest (i.e. forgotten in aversive condition, forgotten in neutral condition, remembered with high confidence in aversive condition, remembered with high confidence in neutral condition) into the model. To ensure the reliability of associative memory performance, trials later remembered with low confidence regardless of Emotion conditions were not included in the model. Mean *t* values extracted from the resultant contrast images in each ROI were then submitted to a 2 (Emotion: aversive vs. neutral) by 2 (Memory Status: forgotten vs. remembered with high confidence) repeated-measures ANOVA.

## Task-free functional connectivity analysis

To assess emotion-induced changes in post-encoding hippocampal functional connectivity, we conducted seeded correlational analyses of resting-state fMRI data for three rest scans separately (i.e. Rest 1, 2, and 3) . Regional time series within the hippocampal seed were extracted from bandpass-filtered images with a temporal filter (0.008–0.10 Hz), and then submitted into the individual level fixed-effects analyses. Six motion parameters as well as cerebrospinal fluid and white matter of each participant that account for potential physiological noise and movement-related artifacts were regarded as covariates of no interest. To mitigate potential individual differences in baseline connectivity, hippocampal-seeded connectivity map at Rest 1 was subtracted from Rest 2 (i.e. Rest 2 vs. 1), and connectivity map at Rest 2 was subtracted from Rest 3 (i.e. Rest 3 vs. 2) for each participant. The resultant connectivity maps were then submitted separately into the second-level multiple regression models, with memory accuracy scores in the aversive and neutral conditions (i.e. memory with high confidence) as two separate covariates of interest. We conducted a contrast to test the difference in regression coefficients between aversive and neutral conditions – that is the interaction effect of post-encoding hippocampal connectivity changes (i.e. Rest 2 vs. 1 or Rest 3 vs. 2) in relation to memory performance and emotional conditions (i.e. aversive and neutral). Significant clusters were determined using the same criterion with above gPPI analyses. Additionally, we also conducted control analyses with a task-unrelated parahippocampal region as seed of interest. All procedures were same as the above hippocampal-seeded analysis.

To visualize the relationships between post-encoding hippocampal connectivity changes and memory performance in aversive and neutral conditions, mean *t* values extracted from the resultant contrast images in each ROI and their corresponding memory accuracies (i.e. memory with high confidence) were plotted into partial correlations without further statistical inferences.

## Prediction analysis

We used a machine learning-based prediction algorithm with balanced fourfold cross-validation to confirm the robustness of relationships of reactivation and functional connectivity with memory performance. This prediction analysis complements conventional correlation models which are sensitive to outliers and have no predictive value (*Cohen et al., 2010*; *Geisser, 1993*; *Qin et al., 2014*). Individual's reactivation (or functional connectivity) index was entered as an independent variable,

and their corresponding memory performance was entered as a dependent variable. Data for these two variables were divided into fourfolds. A linear regression model was built using data from three out of the four folds and used to predict the remaining data in the left-out fold. This procedure was repeated four times to compute a final $r_{(predicted, observed)}$. Such $r_{(predicted, observed)}$, representing the correlation between the observed values of the dependent variable and the predicted values generated by the linear regression model, was estimated as a measure of how well the independent variable predicted the dependent variable. Finally, a nonparametric approach was used to test the statistical significance of the model, by generating 1000 surrogate data sets with randomly shuffled participant labels, under the null hypothesis of $r_{(predicted, observed)}$ (*Cohen et al., 2010*). The statistical significance (i.e. p value) was determined by measuring the percentage of generated surrogate data greater than the true correlation.

## Mediation analysis

We conducted a mediation analysis to further explore how emotional learning affects initial associative memory through functional amygdala-hippocampal-neocortical pathways during emotional learning using Mplus 7.0 software (https://www.statmodel.com/index.shtml) (*Hayes et al., 2011*). Mediation models were constructed to investigate how hippocampal-neocortical (i.e. mFFA and sLOC) connectivity mediated the influence of hippocampal-amygdala connectivity on associative memory performance in the aversive condition. We used hippocampal-amygdala connectivity as the predictor, associative memory performance as the outcome, and hippocampal-mFFA and -sLOC connectivity as two separate mediators. In this model, individual's hippocampal connectivity with each targeted ROI (i.e. amygdala, mFFA and sLOC) was measured with mean $t$ values extracted from the corresponding contrast images of gPPI analysis (i.e. remembered with high confidence vs. forgotten). Individual's associative memory performance was measured with correctness proportion for face-object associations remembered with high confidence. The mediating effect of hippocampal-neocortical connectivity was tested by a bias-corrected bootstrap with 1,000 samples, which could improve the sensitivity and robustness of statistical estimates in small-to-moderate samples (*Preacher and Hayes, 2008*; *Shrout and Bolger, 2002*; *Tian et al., 2021*). Both direct and indirect effects of hippocampal-amygdala connectivity on associative memory were estimated, which generated percentile based on confidence intervals (CI).

## Estimates of effect size

Effect sizes reported for ANOVAs are partial eta squared, referred to in the text as $\eta^2$. For paired t-tests, we calculated Cohen's *d* using the mean difference score as the numerator and the average standard deviation of both repeated measures as the denominator (*Lakens, 2013*). This effect size is referred to in the text as $d_{av}$, where 'av' refers to the use of average standard deviation in the calculation.

## Acknowledgements

This work was supported by the National Natural Science Foundation of China (32130045, 82021004, 31871110, 31522028, 81571056), the Open Research Fund of the State Key Laboratory of Cognitive Neuroscience and Learning (CNLZD1503), and the PhD scholarship (201806040186) of the Chinese Scholarship Council. We thank Liping Zhuang and Siya Peng for their assistance in data analysis. We also thank Nils Kohn and three reviewers for their valuable suggestions and comments for the manuscript.

## Additional information

### Funding

| Funder | Grant reference number | Author |
| --- | --- | --- |
| National Natural Science Foundation of China | 32130045 | Shaozheng Qin |

| Funder | Grant reference number | Author |
|---|---|---|
| National Natural Science Foundation of China | 82021004 | Shaozheng Qin |
| National Natural Science Foundation of China | 31522028 | Shaozheng Qin |
| Open Research Fund of the State Key Laboratory of Cognitive Neuroscience and Learning | CNLZD1503 | Shaozheng Qin |
| Chinese Scholarship Council | 201806040186 | Yannan Zhu |
| National Natural Science Foundation of China | 31871110 | Changming Chen |
| National Natural Science Foundation of China | 81571056 | Shaozheng Qin |

The funders had no role in study design, data collection and interpretation, or the decision to submit the work for publication.

## Author contributions

Yannan Zhu, Conceptualization, Data curation, Formal analysis, Funding acquisition, Investigation, Visualization, Writing – original draft, Project administration, Writing – review and editing; Yimeng Zeng, Formal analysis, Visualization, Writing – review and editing; Jingyuan Ren, Formal analysis, Writing – review and editing; Lingke Zhang, Changming Chen, Investigation; Guillen Fernandez, Supervision, Writing – review and editing; Shaozheng Qin, Conceptualization, Data curation, Supervision, Funding acquisition, Writing – original draft, Project administration, Writing – review and editing

## Author ORCIDs

Yannan Zhu http://orcid.org/0000-0001-5935-4282
Jingyuan Ren http://orcid.org/0000-0002-7089-6397
Changming Chen http://orcid.org/0000-0002-3501-4643
Shaozheng Qin http://orcid.org/0000-0002-1859-2150

## Ethics

Informed written consent was obtained from each participant before the experiment.Consent authorisation for publication was also obtained by a written consent form from each individualwho provided his/her identifying image for illustration purpose in the article. The Institutional Review-Board for Human Subjects at Beijing Normal University (ICBIR_A_0098_002), Xinyang Normal University(same as above) and Peking University (IRB#2015-09-04) approved the procedures for Study 1, 2 and 3respectively.

## Decision letter and Author response

Decision letter https://doi.org/10.7554/eLife.60190.sa1
Author response https://doi.org/10.7554/eLife.60190.sa2

# Additional files

## Supplementary files

- Supplementary file 1. Acoustic characteristics of the four voice clips.
- Supplementary file 2. Brain regions involved in emotional effect on condition-level reactivation.
- Supplementary file 3. Brain regions involved in emotional effect on hippocampal connectivity during emotional learning.
- Supplementary file 4. Post-encoding hippocampal connectivity changes in positive relation to memory in the aversive condition but negative in the neutral condition.
- MDAR checklist

## Data availability

All fMRI data collected in this study are available on OpenNeuro under the accession number ds004109 (https://doi.org/10.18112/openneuro.ds004109.v1.0.0). All code used for analysis are available on GitHub (https://github.com/QinBrainLab/2017_EmotionLearning.git, copy archived at swh:1:rev:da02cb17f27adb21652f3fc878f4bd39e5b88e38).

The following dataset was generated:

| Author(s) | Year | Dataset title | Dataset URL | Database and Identifier |
|---|---|---|---|---|
| Zhu YN, Zhang L, Zeng YM, Chen C, Fernández G, Qin S | 2022 | Emotional learning retroactively promotes memory integration through rapid neural reactivation and reorganization | https://doi.org/10.18112/openneuro.ds004109.v1.0.0 | OpenNeuro, 10.18112/openneuro.ds004109.v1.0.0 |

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
