## [Editor Report]

This manuscript presents valuable insights into neural mechanisms driving emotional memory enhancements for previously neutral information. The authors present compelling behavioral evidence that memory for items indirectly paired with an aversive event is enhanced, and that these enhancements are associated with increased fMRI signal interactions between the amygdala, hippocampus, and stimulus-relevant cortex. The work will be interesting to researchers interested in learning and memory.

---

## [Decision Letter]

**Decision letter after peer review:**

Thank you for sending your article entitled "Emotional tagging retroactively promotes memory integration through rapid neural reactivation and reorganization" for peer review at *eLife*. Your article is being evaluated by 3 peer reviewers, and the evaluation is being overseen by a Reviewing Editor and Laura Colgin as the Senior Editor.

As you can see, although reviewers felt that your study was well-designed and the results novel and interesting, they had a number of concerns. Most notably, they requested additional analyses that (a) demonstrate that item rather than category information was reactivated (R1's point 3, R2's point 1), (b) show evidence for the regional specificity of these effects (R3's point 4), and (c) test the interaction in the connectivity analysis to show that coupling predicts memory for aversive but not neutral items (R1's point 5; also see R2's points 2 for different but related issues). Given that your study is potentially underpowered to detect individual differences (R3's point 1), reviewers were concerned that these analyses are not possible in your data. Please let us know how you plan to respond to these and the other comments raised by reviewers.

*Reviewer #1:*

Zhu and colleagues present a manuscript detailing neural mechanisms driving associative memory enhancements for previously neutral information using an acquired equivelance paradigm. The authors show greater associative memory for face-word object pairs, when the face is later paired with an aversive event. Further, these memory enhancements were associated with increased interactions between the amygdala, hippocampus, relevant-sensory cortex, and higher-order associative cortex both during encoding and post-encoding rest. The question is quite timely, and the results provide a nice extension of rodent work. The analysis plan was well designed, and the results were highly interesting. However, there were some weaknesses in the methodological approach as well and the interpretation of the data that derail from the overall impact. These concerns are detailed below.

1. While the authors provide a great mechanistic understanding of retroactive memory effects, the framing around behavioral tagging and studies related to Dunsmoor et al., (2015) seem inaccurate. In the behavioral tagging literature, the focus is on retroactive memory effects for information that is un-related to the arousing events. By using a sensory pre-conditioning paradigm, the same stimuli are shown during both the pre-arousal phase and the arousal phase. The authors tend to conflate sensory preconditioning with behavioral tagging. This is especially relevant as the authors show immediate memory benefits which are more consistent with sensory pre-conditioning rather than behavioral tagging. With that in mind, I think the introduction needs to be significantly re-worked and more work needs to be done integrating the findings with human sensory pre-conditioning literature (i.e., Wimmer & Shohamy 2012).

2. Related to the above point, I think two additional factors need to be addressed when thinking about interpreting the data through the lens of behavioral tagging versus sensory preconditioning. The first is that effects were occurring during an immediate memory test, as behavioral tagging is thought to support delayed effects. The second would be a moderating role for memory strength during phase 1, perhaps by using the vividness ratings. Behavioral tagging should produce larger effects on weak items (i.e., low vividness), while sensory preconditioning should produce larger effects on strong items (i.e., high vividness).

3. While I found the RSA results to be quite interesting, I had a few methodological concerns that made me question their validity. First, in general affective information can increase activation in many of the ROIs of interest in a non-specific way, including the FFA and LO. The authors should re-run their analyses controlling for univariate activation. Second, I think the RSA comparisons could be more specific and better controlled. Rather than collapsing across the entire condition, the RSA analysis would be more powerful run on an individual-trial level. Thus, the comparisons would be RSA between a pair of trials in phase 1 and 2, and then that trial in comparisons to all other trials within the same condition and all other trials in the other condition. This analysis would show whether a specific item is being reinstated or a more broad category representation.

4. There is a growing body of work characterizing post-encoding mechanisms contributing to memory benefits, in particular emotional memory benefits, that should be included in the discussion (see DeVoogd, Tambini, Murty, Hermans).

5. The post-encoding analyses need to have additional controls. Ideally, the authors would run an interaction analysis to characterize regions in which post-encoding coupling predicts Emotional but not Neutral memory. Further, control analyses in sensory regions that are not directly related to the stimuli (such as PPA) would be beneficial.

*Reviewer #2:*

The manuscript by Zhu and colleagues examines the retroactive influence of emotional 'tags' on associative memory for tagged items, providing new behavioral evidence for a retroactive memory benefit associated with emotional arousal. This is replicated across a few behavioral cohorts and is importantly distinct from prior demonstrations that emotion can enhance item memory for material that is conceptually related to 'tagged' information. fMRI is used to examine possible neural mechanisms for the retroactive benefit: greater reactivation of initial learning patterns is present for the emotionally tagged condition, and hippocampal interactions are enhanced in this condition as well. There is also evidence for post-tagging resting hippocampal interactions that predict the emotional-tagging memory benefit, suggesting that both mechanisms at the time of 'tagging', and subsequent mechanisms may mediate this behavioral effect. I think the main results in the manuscript are sound and advance the literature. I think there a few points that need to be considered, however.

1. The reinstatement effect (greater for emotional vs. neutral trials) is a compelling demonstration that is consistent with the retroactive memory benefit. A couple of points to consider regarding this analysis:

a. The current approach creates an average pattern for 'aversive' and 'neutral' stimulus pairs, both during tagging and initial learning. Thus, greater reinstatement for the aversive condition likely reflects category-level reinstatement of face/object information present during learning, rather than the specific reinstatement of a particular pair (which is implied in the discussion, e.g. "stimulating reactivation of overlapping memory traces" on line 414). Did the authors also examine evidence for pair-specific reinstatement (i.e. greater similarity between the tagging of Face A and encoding trial of Face A, vs. the tagging of Face A and encoding of Face B)? Evidence for trial-specific reinstatement would provide a tighter mechanistic link between reactivation and enhanced associative memory associated with emotional tagging.

b. At face value, the relationship between reinstatement evidence and associative memory (in Figure 2D) helps to link reinstatement with subsequent memory. However, given that the relationship is not specific (present for both emotional and neutral conditions) it makes me wonder whether there are alternative explanations beyond reactivation explaining better memory. Since reinstatement is a simple relationship between across-trial encoding and tagging phases, it is possible that participants with less reliable or consistent activity patterns during either the encoding or tagging phase alone (due to fluctuations in attention or other factors) may have worse memory, which may drive the correlation. This can be addressed by (1): showing that the similarity within encoding, i.e. across trials or conditions is not predictive of memory, (2): the same for the tagging phase, and/or (3): by examining evidence for pair-specific reinstatement as described above, which is a more specific measure.

2. The differences in hippocampal interactions based on emotional tagging (Figure 3D,E) reveal greater coupling with the amygdala and FFA/LOC during emotional vs. neutral 'tagging', consistent with the retroactive emotional memory enhancement. The further analyses in Figure 3F,G indicate, however, that these differences are not specifically linked with the emotion-related memory enhancement and instead show main effects of memory and emotion (not an interaction). Despite this lack of differences due to the emotional memory enhancement (across trials) the authors examine whether individual differences in these measures are related to emotionally-tagged memory.

a. Please provide logic linking these analyses – why was it expected that these measures would predict emotionally-tagged memory across subjects when trial-level effects relating to emotional memory (specifically) were not found?

b. Please also show the correlations that are described in the text (lines 327-335). It is a bit surprising to see the come up as boxes in the SEM analysis (Figure 3H) although they are not shown individually first. Do the correlations between memory and hippocampal connectivity (w/ amygdala and FFA/LOC) significantly differ between the emotionally-tagged and neutral conditions? The correlations are significant for the emotional condition, but not the neutral condition, implying specificity although this is not shown/tested. Or, the analysis can be done with the difference in memory for emotional vs. neutral conditions to gain this specificity, which was used for the rest analyses in Figure 4.

c. Lastly, logic should also be provided for constructing the SEM in the specific manner it was setup (hippocampal-cortical connectivity serving as the mediating variable between hipp-amygdala connectivity and memory).

3. The primary rest analysis – which examines changes in hippocampal connectivity (rest-3 minus rest-2) that are related to individual differences in emotionally-tagged memory enhancement – is clear and well-motivated. However, the logic underlying the first analysis, which relates changes in connectivity from rest-1 to rest-2 (baseline to post-initial-learning rest) to the emotionally-tagged memory enhancement, is not clear. Given that emotional tagging occurs after rest-2, there should be no meaningful signature associated with biased consolidation of this material prior to this time point. I understand that the authors are trying to show how emotional tagging, per se, alters post-encoding consolidation signatures during rest above and beyond those that are present after initial learning alone. I infer that the goal is to compare the post-tagging rest to an analysis that illustrates simple relationships between immediate post-encoding connectivity and later memory (not related to tagging). To achieve this goal with the current design, I would suggest a more straightforward analysis of examining changes in hippocampal connectivity from rest-1 to rest-2 that are related to individual differences in associative memory that are not related to tagging, such as the average across neutral and emotionally-tagged stims (or perhaps neutral stimuli that are not emotionally tagged). Average associative memory would likely be the best measure, considering that emotional tagging may enhance memory for related material at the cost of impaired memory for non-tagged material. There is clear logic in this analysis, in that it would reveal initial post-encoding resting connectivity that is related to memory and NOT the emotional tagging procedure. The contrast of this kind of analysis with the rest-3 minus rest-2 analysis would then be clear. Note that the main analysis (examining changes from rest-2 to rest-3) on its own somewhat controls for initial post-encoding activity (captured in rest-2), so perhaps this other analysis is not needed (although the claims made in the Discussion would have to be altered).

*Reviewer #3:*

In this paper, Zhu and colleagues report a series of studies (2 behavioral and 1 fMRI) designed to test the retroactive effects of aversive associations-- which they call "emotional tagging"-- on memory for previously-learned, overlapping neutral associations. Across the 3 studies, they found that neutral face-object associations were remembered better if the face was later paired with an aversive sound. In the fMRI study, the authors additionally showed that the tagging phase was associated with greater pattern reactivation when new associations were aversive, compared to neutral. They also found changes in functional connectivity with the hippocampus that were associated with the retroactive effects of tagging, both during the tagging phase itself and during a post-tagging resting-state scan.

This paper had several strengths. The behavioral findings were compelling, showing the same general pattern across the 3 studies. The topic is timely, as there is accumulating evidence for the retroactive effects of emotion on memory, but little neural data explaining such effects in humans. The analysis approach included multiple sophisticated methods.

However, there were also significant weaknesses that limit the impact of the paper.

1. The first major weakness is the sample size of the fMRI study (effective N=28), coupled with the fact that many of the conclusions were based on correlations with individual differences in memory-- including the findings that hippocampal reactivation and changes in functional connectivity predicted retroactive memory benefits. This is a concern because correlations across small sample sizes are less likely to replicate in future work. The mediation model is subject to these same concerns.

2. The second major weakness is a lack of theoretical clarity regarding the mechanisms supporting the retroactive memory benefit. Two main ideas are introduced: the idea of behavioral tagging (i.e., as related to synaptic tag-and-capture models) and the idea that reactivation plays an important role in memory allocation and integration. I see these as two separate theoretical perspectives that could make different predictions here, which may have been the authors' intention. However, they are not clearly set up as competing hypotheses, nor are they clearly integrated into a unified account. This made it difficult to interpret the results in light of either account.

3. The terms "reorganization" and "reconfiguration" are used to describe the functional connectivity results. I have seen these terms used to refer to network-level changes, but they overstate the functional connectivity differences here, which are simply changes in the maps resulting from a seed-based functional connectivity analysis.

4. It would be more informative to examine pattern reactivation in face-selective and object-selective areas separately (here, they appear to be lumped together), since only reactivation of object-selective areas should be taken as "pure" evidence of learning-phase reactivation, due to the overlap in faces shown during the learning and tagging phases. If there's an increase in pattern similarity in face-selective areas, this could be explained by changes in attention to the visual stimuli associated with the aversive sound rather than reactivation per se.

5. The prediction approach was a nice addition here, in that it can help to determine which effects are robust across subjects. However, it appears that they were based on clusters that were already deemed significant through conventional statistical analyses (e.g., see lines 794-795), which suggests that they are biased by the group effects.

[Editors' note: further revisions were suggested prior to acceptance, as described below.]

Thank you for resubmitting your work entitled "Emotional learning retroactively promotes memory integration through rapid neural reactivation and reorganization" for further consideration by *eLife*. Your revised article has been evaluated by Laura Colgin (Senior Editor) and a Reviewing Editor.

The manuscript has been improved but there are some remaining issues that need to be addressed, as outlined below:

Reviewers felt that the primary results reported in your manuscript are sound but that some of the details of the later analyses were not fully transparent which made those harder to evaluate. Please respond to the points raised by the individual reviewers.

*Reviewer #1:*

The authors did a truly outstanding job responding to the concerns that I raised in the first review. I sincerely believe that the new introduction and discussion provide a much more emperically-motivated depiction of the prior literature, and does a stand-out job delineating these findings from other related concepts such as behavioral tagging.

I was also excited to see such clear results from the additional analyses run. The trial-specific reactivation results give a much more clear picture of the types of reactivation supporting their retroactive memory effects, and the extent of the specificity is quite compelling.

Similarly, the additional controls of the post-encoding analyses were quite strong. One point did emerge that was interesting, which was that there were negative correlations with neutral memory. These findings, albeit with some speculation, suggest that there is a prioritization of related emotional information rather than just an enhancement, meaning there may be a trade-off with neutral information. This would be very aligned with the circuits of interest and the role of noreadrenaline (see Clewett & Murty, 2019). While not completely necessary for this manuscript, additional discussion of this feature of the data that emerged from the new analyses could be included.

*Reviewer #2:*

The manuscript by Zhu et al. has improved with revision; I appreciate the addition of the more specific trial-level reactivation analysis which more directly tests the notion that reactivation is a mechanism supporting retroactive emotional memory enhancements. The authors have responded to the prior concerns thoroughly. New details were added regarding ROI definition, in response to prior comments. These new details currently limit my enthusiasm for the manuscript due to the lack of straightforward presentation of some of the analyses/results in the paper: how regions in Figures 4-6 were identified (all other results/analyses are clearly described; see below). Otherwise, the paper makes an important contribution to the field.

A primary concern is the way regions were isolated/identified in the analyses performed after the pattern similarity (those in Figures 4-6). I am not sure whether the FFA/LOC was defined once (i.e. those used in Figures 2-3) and then voxels within those regions were isolated showing connectivity effects for analyses in Figures 4-6, or whether separate regions were isolated across each analysis and are referred to with a common label (implying they are the same). This stems from the second paragraph of the ROI definition section (Methods) which I found to be unclear. As written, it is *implied* that FFA and LOC are re-defined in subsequent analyses "FFA and LOC were derived from a group contrast map of the aversive relative to neutral condition" and "significant clusters (i.e. LOC…) … were derived from the group-level multiple regression analysis on connectivity…". If the ROIs are re-defined in different analysis, it is misleading to use the same label to refer to them across analyses (which implies that they are the same regions or would at least be restricted to a common definition). Please both (1) clarify the approach because it is still not clear to me how the ROIs were defined for connectivity analysis (i.e. did you search for specific contrast within the original FFA/LOC definitions, or were these ROIs re-defined for each analysis within the broader anatomical mask from WFU pickatlas) and (2) the description of the ROIs/results in the main manuscript text should not be misleading as to how the analysis is conducted. If separate regions are isolated for different analyses, then this should be reflected in the labels used for these regions (they should not all have the same label which imply face and object processing regions if they are isolated by other means).

The ROI definition issue raises possible problems of non-independence and multiple comparisons corrections which are currently obfuscated given that the analysis steps are not spelled out. If the 'FFA' and 'LOC' regions shown in Figure 4 were defined from showing greater functional connectivity w/ the hippocampus during aversive vs. neutral trials (not how these regions are typically defined), then the results reported in Figure 4C are non-independent and inferential tests (ANOVAs) should not be performed on this data as it is circular. It is also stated that the gPPI was performed separately for initial encoding and emotional learning phases and clusters were isolated from each – presumably the clusters shown in Figure 4 were isolated from the emotional learning analysis since they show a strong effect in that phase? Please state which analysis they were defined from. The problem of non-independence seems to be remedied in Figure 6 since no statistics are reported and just brain regions are shown. Another possible issue is multiple comparisons corrections, especially *if the FFA/LOC definitions are not carried forward into subsequent analyses*. The original ROI definition does not include an explicit correction but uses a stringent threshold of P<.0001, 30 voxels (fine for ROI definition such that ROIs are interrogated in later analyses). But this thresholding procedure is referenced in subsequent analyses, so presumably all regions isolated in Figures 4-6 are using this criterion. Thus it needs to be specified how this criterion satisfies multiple comparison/family wise error correction / was chosen. Moreover, please clarify if the FFA/LOC regions in Figures 4-6 were also constrained anatomically by the wfu pickatlas regions and neurovault contrasts.

*Reviewer #3:*

In this version of the manuscript, Zhu and colleagues have revised their introduction and discussion to situate their experiment within the framework of sensory preconditioning, rather than behavioral tagging. I found the theoretical framework to be much better developed in this revision, and the results link more clearly to the existing conditioning and memory integration literature. Another major revision was the inclusion of a trial-specific reactivation analysis, in which they showed that trial-specific hippocampal patterns were reinstated during the emotional learning phase. This could provide a mechanism by which initially-neutral associations are strengthened through emotional learning. This is an interesting and potentially important result.

In reading the manuscript again, I was impressed with the robust behavioral findings, and I thought the pattern similarity analyses were largely convincing (see one comment below).

I remain concerned that there is too much emphasis placed on the results based on across-subject correlations, given the relatively small sample size (N=28). This includes the structural equation model as well as the results of the resting-state functional connectivity analyses. I am more convinced by the prediction analyses that follow up on results from the pattern similarity and task-related connectivity analyses. The resting-state analyses are particularly susceptible to the problems of underpowered correlations because they were computed across all voxels in the brain (see Marek et al. 2022 Nature for further discussion). In their response to the previous reviews, the authors described the across-subject correlation analyses as "exploratory" and complementary to the main lines of evidence. Yet this is not how they are presented in the paper itself, where the SEM and resting-state analyses are highlighted as key findings (e.g., see lines 558-563 in the results summary on p. 29). At minimum, more caution should be expressed throughout, with analyses clearly marked as exploratory.

For the structural equation model, the authors used a bootstrapping technique that may improve the ability to estimate direct and indirect effects even in small sample sizes. The model fit indices, however, seem to have been computed in the standard way. Given that the analysis may be underpowered to detect model misspecification, the authors should tone down their description of the model fit (both for the primary model as well as the alternative tested models).

In the pattern similarity analyses, reinstatement was observed at the onset of the voice but not in response to the face cue alone. Yet it appears that the voice onset occurred only 2s after the onset of the face cue, which is the equivalent of one TR. With that temporal resolution, it shouldn't be possible to reliably separate these two signals in time, and including them in the same model may lead to unstable parameter estimates. How have the authors addressed this?

There are a couple of places in the Results where non-significant results are described as significant, in cases when one ROI shows a significant effect but the other doesn't: the LOC interaction in line 295, the amygdala correlation in line 433. Conclusions should also be updated to refer only to those findings that are significant.

---

## [Author Response]

Reviewer #1:Zhu and colleagues present a manuscript detailing neural mechanisms driving associative memory enhancements for previously neutral information using an acquired equivelance paradigm. The authors show greater associative memory for face-word object pairs, when the face is later paired with an aversive event. Further, these memory enhancements were associated with increased interactions between the amygdala, hippocampus, relevant-sensory cortex, and higher-order associative cortex both during encoding and post-encoding rest. The question is quite timely, and the results provide a nice extension of rodent work. The analysis plan was well designed, and the results were highly interesting. However, there were some weaknesses in the methodological approach as well and the interpretation of the data that derail from the overall impact. These concerns are detailed below.

We appreciate the positive evaluation and insightful comments raised by the Reviewer. In response to the Reviewer’s comments, we have conducted several additional analyses (see below) to address the methodological issues and data interpretation, and have revised the manuscript accordingly. We feel that with our additional analyses and revisions, our findings and conclusions about emotion-charged retroactive memory enhancement for previously neutral event are now more clearly supported.

1. While the authors provide a great mechanistic understanding of retroactive memory effects, the framing around behavioral tagging and studies related to Dunsmoor et al., (2015) seem inaccurate. In the behavioral tagging literature, the focus is on retroactive memory effects for information that is un-related to the arousing events. By using a sensory pre-conditioning paradigm, the same stimuli are shown during both the pre-arousal phase and the arousal phase. The authors tend to conflate sensory preconditioning with behavioral tagging. This is especially relevant as the authors show immediate memory benefits which are more consistent with sensory pre-conditioning rather than behavioral tagging. With that in mind, I think the introduction needs to be significantly re-worked and more work needs to be done integrating the findings with human sensory pre-conditioning literature (i.e., Wimmer & Shohamy 2012).

We appreciate the Reviewer for thoughtful comments and suggestions. We agree that our observed retroactive memory effect is more consistent with sensory preconditioning protocols. We have now undertaken several steps to emphasize sensory preconditioning literature and avoid conflating that with behavioral tagging in our revised manuscript. First of all, we would like to clarify that we introduced behavioral tagging literature as one of potential neurobiological mechanisms underlying the retroactive memory enhancement. As we had already discussed in our original manuscript, the retroactive memory benefit in our study differed from the conventional behavioral tagging effects, by at least three following aspects: (1) we observed the trial-specific retroactive memory benefit with associative learning task in which overlapping face cues were presented during both initial and emotional learning phases, rather than a category-level effect with non-overlapping stimuli proposed in behavioral tagging literature; (2) we observed the retroactive memory benefit in the immediate test, which differs from the delayed benefit pertaining to behavioral tagging literature; (3) we observed the retroactive benefit selective for strongly encoded associations with high-vividness ratings (presumably strong memories; also see our response to point 2 below), which differs from the rescued effect for weak memories expected by behavioral tagging literature. Hence, we fully agree that our observed retroactive memory benefit is more consistent with sensory preconditioning paradigm, which also accommodates with memory integration models.

Second, according to the Reviewer’s suggestion, we have now rewritten the first three paragraphs of the Introduction section to emphasize the sensory preconditioning literature in our revised manuscript *(*Page 1-3). Specifically, in the first paragraph, we introduced the retroactive enhancement of emotional experience on related memory for previously mundane events in our integrated episodic memory system (Holmes et al., 2022; Shohamy and Daw, 2015; Wong et al., 2019), and raised an open question on its underlying neurocognitive mechanisms in humans. Then in the second paragraph, we focused on the emotion-charged retroactive enhancement for specific episodic memory, referred to in the text as trial-specific effect, which more closely fits our everyday memories but goes beyond explanations by conventional behavioral tagging models (Ballarini et al., 2009; Clewett et al., 2022; Dunsmoor et al., 2015; Takeuchi et al., 2016). We thus introduced the sensory preconditioning paradigm which allows us to investigate trial-specific effect by associative learning tasks, and also introduced previous studies of sensory preconditioning in animals and humans which provide evidence for trial-specific retroactive effects on associated events (Brogden, 1939; Kurth-Nelson et al., 2015; Li et al., 2008; Sadacca et al., 2018; Sharpe et al., 2017; Wimmer and Shohamy, 2012). We further raised an open question on the neurobiological mechanisms underlying such emotion-charged trial-specific effect. Moreover, in the third paragraph, we introduced memory allocation and integration mechanisms which have been proposed to accommodate sensory preconditioning (Holmes et al., 2021; Schlichting and Frankland, 2017; Schlichting and Preston, 2015; Shohamy and Daw, 2015; Shohamy and Wagner, 2008; Wong et al., 2019), to understand the potential neurocognitive processes (i.e., reactivation of hippocampal and stimulus-sensitive neocortical representations, as well as hippocampal–neocortical coordinated interactions) underlying the emotion-charged trial-specific retroactive memory benefit.

Third, we have substantially adapted the Discussion section of our revised manuscript (Page 14). We decided to mainly interpret our results in the context of sensory preconditioning literature and memory integration models (Arcediano et al., 2003; Holmes et al., 2021; Li et al., 2008; Schlichting and Preston, 2015; Shohamy and Daw, 2015; Shohamy and Wagner, 2008; Wimmer and Shohamy, 2012; Wong et al., 2019). We have also elaborated the above-mentioned three major differences between our behavioral data and previous findings reported in behavioral tagging literature. We further concluded that our findings of immediate, trial-specific effects for relatively strong neutral memories provide new insights into the possible mechanisms underlying emotion-charged retroactive memory benefits. It may extend the existing views of sensory preconditioning and memory integration models in the literature.

Thank the Reviewer again for raising this important point. The integrity of the Introduction section and data interpretation in the Discussion section has now been improved in our revised manuscript.

2. Related to the above point, I think two additional factors need to be addressed when thinking about interpreting the data through the lens of behavioral tagging versus sensory preconditioning. The first is that effects were occurring during an immediate memory test, as behavioral tagging is thought to support delayed effects. The second would be a moderating role for memory strength during phase 1, perhaps by using the vividness ratings. Behavioral tagging should produce larger effects on weak items (i.e., low vividness), while sensory preconditioning should produce larger effects on strong items (i.e., high vividness).

We appreciate the Reviewer for this thoughtful suggestion. For the first factor regarding to immediate versus delayed memory enhancement effects, we have now decided to interpret our findings of emotion-charged retroactive memory benefit mainly through a lens of sensory preconditioning literature (Arcediano et al., 2003; Holmes et al., 2021; Li et al., 2008; Shohamy and Daw, 2015; Wimmer and Shohamy, 2012). We have further clarified the major differences between the retroactive memory benefit observed in our current study and those reported in previous behavioral tagging studies (Ballarini et al., 2009; Braun et al., 2018; Dunsmoor et al., 2015). Besides the immediate effect (vs. delayed effect) pointed by the Reviewer, we also discussed other potential differences pertaining to trial-specific versus category-level effect, and selective effect on relatively strong memories for initial neutral events versus weak ones. We have now clarified this issue in the Discussion section of our revised manuscript (Page 14).

In light of the Reviewer’s comment on the second factor, we conducted separate paired t-tests for associative memory performance with low and high vividness ratings across three independent studies. In line with our originally reported results, we observed the reliable emotion-induced retroactive benefit only occurring on associations remembered with high confidence. These analyses on high-confidence memory performances (see Figure 1E-G) revealed significantly better memory in the aversive than neutral condition for high-vividness encoded associations (Study 1: t_(29)_ = 2.15, *p* = 0.040, d_av_ = 0.21; Study 2: t_(27)_ = 2.16, *p* = 0.040, d_av_ = 0.25; Study 3: t_(27)_ = 2.21, *p* = 0.036, d_av_ = 0.12), but not for low-vividness encoded associations (Study 1: t_(29)_ = 0.27, *p* = 0.793; Study 2: t_(27)_ = -0.58, *p* = 0.565; Study 3: t_(27)_ = -0.40, *p* = 0.696). These results indicated a selectively emotion-induced memory enhancement only for initially high-vividness encoded neutral associations (potentially strong items).

Parallel analyses for those associations remembered with low confidence (see Figure 1—figure supplement 5), however, did not show consistent results across three studies. These analyses revealed a marginally significant emotion effect (i.e., aversive vs. neutral) for low-vividness encoded associations in Study 1 (t_(29)_ = 1.97, *p* = 0.058), which was not replicated by Study 2 (t_(27)_ = 0.27, *p* = 0.787) nor Study 3 (t_(27)_ = -1.74, *p* = 0.094). No significant emotion effect for high-vividness encoded associations was observed in three studies (Study 1: t_(29)_ = -0.98, *p* = 0.334; Study 2: t_(27)_ = -0.15, *p* = 0.886; Study 3: t_(27)_ = -1.10, *p* = 0.283). Given above inconsistent and insignificant results across three studies, the emotional effects for low-confidence remembered associations would not be discussed in the revised manuscript.

In our revised manuscript, we have noted above new results in the Results section (Page 6), and it now reads as follows: “To further investigate whether initial encoding strength of neutral associations (i.e., vividness rating during initial learning phase on a four-point scale: 4 = “Very vivid”, 1 = “Not vivid at all”) modulates our observed emotion-induced retroactive effect above, we sorted these associations remembered with high confidence into a high-vividness bin with 3 and 4 ratings, and a low-vividness bin with 1 and 2 ratings. Separate paired t-tests for the three independent studies revealed significantly better memory in the aversive (vs. neutral) condition, selectively for high-vividness encoded associations (Study 1: t_(29)_ = 2.15, *p* = 0.040, d_av_ = 0.21; Study 2: t_(27)_ = 2.16, *p* = 0.040, d_av_ = 0.25; Study 3: t_(27)_ = 2.21, *p* = 0.036, d_av_ = 0.12), but not for low-vividness encoded associations (Study 1: t_(29)_ = 0.27, *p* = 0.793; Study 2: t_(27)_ = -0.58, *p* = 0.565; Study 3: t_(27)_ = -0.40, *p* = 0.696; Figure 1E-G). Additionally, parallel analyses were also conducted for those associations remembered with low confidence. However, we did not observe any reliable effect across three studies (Figure 1—figure supplement 5 with statistics). Altogether, these results indicate that our observed emotion-induced retroactive benefit for related memory only occurs on relatively strong-encoded associations with high-vividness rating.”

We have also discussed above results in the Discussion section (Page 14), and it now reads as follows: “Third, our observed retroactive benefit was selective for strongly encoded associations with high-vividness ratings (presumably strong memories), which contradicts with the rescued effect for weak memories expected by the behavioral tagging model (Dunsmoor et al., 2015; Holmes et al., 2021; Ritchey et al., 2016; Wong et al., 2019). Neutral associations with high-vividness ratings were, at least temporarily, encoded into memory during initial learning phase. Their memory traces could be subsequently reactivated by corresponding face cues, and further transformed into a more stable and long-lasting state by arousal during emotional learning phase. However, associations with low-vividness ratings might be encoded relatively weaker. Since no specific memory traces were formed for these low-vividness associations, they could not be reactivated nor enhanced later. It is thus conceivable that the emotion-charged trial-specific retroactive memory integration tends to occur on relatively strong associations.”

Since the behavioral tagging literature is less relevant to interpret our observed immediate and trial-specific effect on strongly encoded associations, we have now decided to avoid using the terms of “emotional tagging” throughout the revised manuscript. Instead, we used “emotional learning”, “emotional arousal” or “emotion-charged” when appropriate in the context. Given the observed selective effect on high-vividness encoded associations (i.e., relatively strong items), we have also avoided any claims on the transformation of initially weak memories into strong memories throughout the revised manuscript.

Thank the Reviewer again for prompting us to analyze our behavioral data regarding to vividness ratings. Outcomes from these new analyses further strengthen our conclusions of the emotion-charged retroactive memory benefit.

3. While I found the RSA results to be quite interesting, I had a few methodological concerns that made me question their validity. First, in general affective information can increase activation in many of the ROIs of interest in a non-specific way, including the FFA and LO. The authors should re-run their analyses controlling for univariate activation. Second, I think the RSA comparisons could be more specific and better controlled. Rather than collapsing across the entire condition, the RSA analysis would be more powerful run on an individual-trial level. Thus, the comparisons would be RSA between a pair of trials in phase 1 and 2, and then that trial in comparisons to all other trials within the same condition and all other trials in the other condition. This analysis would show whether a specific item is being reinstated or a more broad category representation.

The Reviewer raised very good points. According to the Reviewer’s suggestions, we have now conducted several new additional analyses as detailed below. First, we re-ran our RSA analyses by controlling the difference in univariate activation between aversive and neutral conditions. We implemented two steps to control the univariate activation differences for each ROI (i.e., hippocampus, LOC and FFA) and have now added several sentences to incorporate these steps into the Methods section of our revised manuscript. It reads as follows:

1) Page 21: “We eliminated the mean activation from each pattern by z-scoring across voxels of each ROI. Thus, the resulting mean-centered pattern has relative voxel amplitudes (i.e., voxel-level variability) preserved without any difference in the overall amplitudes, ensuring that the subsequent similarity results are fully attributable to the pattern itself (Coutanche, 2013; Tompary and Davachi, 2017).”

2) Page 22: “we conducted separate 2 (Emotion: aversive vs. neutral) by 2 (Measure: pair-specific vs. across-pair) repeated-measures ANCOVAs in each ROI, also with individual’s univariate activation differences (aversive vs. neutral) in both initial and emotional learning phases as covariates of no interest to further mitigate the potential interference of overall activation from pattern results.”

By these two steps, we believe that our analyses are valid and state-of-the-art to reveal the emotion-induced effects on pattern similarity while controlling univariate activation differences.

Second, we re-ran our RSA analyses on individual-trial level and computed the following three measures of trial-level pattern similarity. As shown in Figure 2—figure supplement 3*,* multi-voxel activity pattern of each trial (i.e., face-voice pair) during emotional learning phase was separately correlated with: (1) its corresponding face-object pair during initial learning (pair-specific similarity), (2) different face-object pairs within the same aversive/neutral condition during initial learning (across-pair within-condition similarity), and (3) different face-object pairs from the different condition during initial learning (across-pair between-condition similarity).

For above three pattern similarity measures, we conducted separate 2 (Emotion: aversive vs. neutral) by 3 (Measure: pair-specific vs. across-pair within-condition vs. across-pair between-condition) repeated-measures ANCOVAs in each ROI (see Figure 2—figure supplement 4), with individual’s univariate activation differences (aversive vs. neutral) in both initial and emotional learning phases as covariates of no interest. This analysis for hippocampal pattern similarity revealed a significant main effect of Emotion (F_(1, 25)_ = 11.10, p = 0.003, partial η^2^ = 0.31) and an Emotion-by-Measure interaction (F_(2, 50)_ = 3.72, p = 0.031, partial η^2^ = 0.13), but no main effect of Measure (F_(2, 50)_ = 0.62, p = 0.541, partial η^2^ = 0.02). Post-hoc comparisons (by controlling univariate activation differences between aversive and neutral conditions) revealed a significantly higher pair-specific similarity in the aversive than neutral condition (F_(1, 25)_ = 7.24, p = 0.013, partial η^2^ = 0.22), but no such Emotion effect in across-pair within-condition similarity or across-pair between-condition similarity (both F_(1, 25)_ < 1.16, p > 0.290, partial η^2^ < 0.05). It also revealed that pair-specific similarity was significant higher than across-pair within-condition similarity (F_(1, 25)_ = 4.43, p = 0.046, partial η^2^ = 0.15), and marginally significant higher than across-pair between-condition similarity (F_(1, 25)_ = 3.96, p = 0.058, partial η^2^ = 0.14) in the aversive condition. No difference among these three similarity measures was observed in the neutral condition (all F_(1, 25)_ < 1.00, p > 0.330, partial η^2^ < 0.04). Parallel analysis for LOC pattern similarity also revealed a significant main effect of Emotion (F_(1, 25)_ = 8.45, p = 0.008, partial η^2^ = 0.25), but neither a main effect of Measure (F_(2,50)_ = 0.16, p = 0.849, partial η^2^ = 0.01) nor an Emotion-by-Measure interaction (F_(2, 50)_ = 1.87, p = 0.165, partial η^2^ = 0.07). Consistent with hippocampal similarity, post-hoc comparisons with covariates controlled revealed a significantly higher pair-specific LOC similarity in the aversive than neutral condition (F_(1, 25)_ = 8.28, p = 0.008, partial η^2^ = 0.25), but no such Emotion effect in across-pair within- and between-condition similarity measures (both F_(1, 25)_ < 2.91, p > 0.100, partial η^2^ < 0.11). The pair-specific LOC similarity was also marginally significant higher than across-pair between-condition similarity in the aversive condition (F_(1, 25)_ = 3.42, p = 0.076, partial η^2^ = 0.12). However, parallel analysis for FFA pattern similarity revealed neither main effects of Emotion (F_(1, 25)_ = 2.30, p = 0.142, partial η^2^ = 0.08) and Measure (F_(2,50)_ = 0.53, p = 0.591, partial η^2^ = 0.02) nor their interaction effect (F_(2, 50)_ = 0.06, p = 0.938, partial η^2^ = 0.003). These results indicated that emotional arousal increased pair-specific pattern similarity, rather than across-pair within- nor between-condition similarities, in the hippocampus and LOC. No effect was found in the FFA, because the visual input of face cues during emotional learning might cover potential reactivation pattern of initial face information.

Since there was no significant difference between across-pair within- and between-condition similarity measures (F_(1, 25)_ < 1.30, *p* > 0.268, partial η^2^ < 0.05) and no Emotion effect (i.e., aversive vs. neutral) in each across-pair similarity measure (F_(1, 25)_ < 2.91, *p* > 0.100, partial η^2^ < 0.11) for each ROI, we combined all across-pair similarities (i.e., averaging all correlations in both across-pair within- and between-condition) to quantify a generally category-level representation pattern. To better characterize the relationships between trial-specific reinstatement (pair-specific similarity) and category-level representation (across-pair similarity), we conducted separate 2 (Emotion: aversive vs. neutral) by 2 (Measure: pair-specific vs. across-pair) repeated-measures ANCOVAs in each ROI, also with individual’s univariate activation differences (aversive vs. neutral) in both initial and emotional learning phases as covariates of no interest (see Figure 2B). Consistent with above results, these analyses revealed significant main effects of Emotion in the hippocampus (F_(1, 25)_ = 9.48, *p* = 0.005, partial η^2^ = 0.28) and LOC (F_(1, 25)_ = 9.34, *p* = 0.005, partial η^2^ = 0.27) as well as Emotion-by-Measure interaction effects in the hippocampus (F_(1, 25)_ = 4.89, *p* = 0.036, partial η^2^ = 0.16) and LOC (F_(1, 25)_ = 3.54, *p* = 0.072, partial η^2^ = 0.12), but no main effects of Measure (F_(1, 25)_ < 0.60, *p* > 0.440, partial η^2^ < 0.03 in both ROIs). Post-hoc comparisons with covariates controlled revealed significant higher pair-specific similarity (both statistics were the same as above), and marginally significant higher across-pair similarity in the hippocampus (F_(1, 25)_ = 4.16, *p* = 0.052, partial η^2^ = 0.14) and LOC (F_(1, 25)_ = 3.37, *p* = 0.078, partial η^2^ = 0.12) in the aversive than neutral condition. It also revealed the (marginally) significant higher pair-specific than across-pair similarity in both the hippocampus (F_(1, 25)_ = 4.42, *p* = 0.046, partial η^2^ = 0.15) and LOC (F_(1, 25)_ = 2.96, *p* = 0.097, partial η^2^ = 0.11) in the aversive condition, but not in the neutral condition (F_(1, 25)_ < 2.20, *p* > 0.150, partial η^2^ < 0.08 in both ROIs). Parallel analysis for FFA pattern similarity revealed neither main effect nor interaction effect (all F_(1, 25)_ < 1.59, *p* > 0.219, partial η^2^ < 0.06). These results more directly indicated that emotional arousal promoted greater trial-specific reinstatement relative to category-level representation in the hippocampus and LOC, supporting our observed trial-specific retroactive memory enhancement.

**Author response image 1. sa2fig1:** Post-encoding hippocampal connectivity changes in relation to memory difference (i.e., aversive vs. neutral). (A) Significant cluster in the lateral occipital cortex (LOC), showing its greater connectivity with the hippocampus at Rest 2 (vs. Rest 1) in positive relation to the memory difference. (B) Significant clusters in the right anterior prefrontal cortex (aPFC), the bilateral inferior parietal lobule (IPL) extending into angular gyrus and the right posterior cingulate cortex (PCC), showing their greater connectivity with the hippocampus at Rest 3 (vs. Rest 2) in positive relation to the memory difference. Notes: Color bars represent T values; two-tailed tests.

Altogether, we hope the Reviewer is now convinced that results from above new analyses suggest an emotion-induced enhancement in trial-specific reinstatement of initially encoded memories, rather than in broad category representation, in the hippocampus and LOC. However, given the overlap in faces shown during both initial and emotional learning phases, FFA pattern similarity did not reflect a pure reinstatement (or reactivation) of face information, and thus did not show the trial-specific emotional effect. The marginally significant emotional effects for across-pair similarity (i.e., category-level representation) in three ROIs might be due to that the stimuli associated with screams generally attracted more attention to the aversive compared to neutral condition, rather than pattern reinstatement (or reactivation). Hence, our observed retroactive memory benefit is most likely resulted from an emotion-enhanced trial-specific reinstatement of the hippocampal and LOC activity patterns, rather than a broad category representation due to arousal-based attentional priority in the aversive (vs. neutral) condition. We have now added these results into the Results (Page 7; Figure 2A and B) and the Supplemental Materials (Figure 2—figure supplement 3 and Figure 2—figure supplement 4). We also updated the Discussion (Page 14-15) and Methods (Page 22) sections accordingly of our revised manuscript.

We would like to thank the Reviewer again for raising above methodological concerns and suggestions, which prompt us to conduct a set of additional trial-level pattern similarity analyses. Outcomes from these analyses are very interesting, and provide novel trial-specific evidence to complement our original findings at the condition level.

4. There is a growing body of work characterizing post-encoding mechanisms contributing to memory benefits, in particular emotional memory benefits, that should be included in the discussion (see DeVoogd, Tambini, Murty, Hermans).

We thank the Reviewer for this specific suggestion. We have now added several sentences in the Discussion section to better discuss possible post-encoding mechanisms that contribute to the emotional memory benefits in our study. The references suggested by the Reviewer have been also cited appropriately. It now reads as follows (Page 16): “Besides, neural circuits activated by emotional learning exhibit persistent activity and alter a series of systems-level interactions during post-emotional-learning rest (i.e., hippocampal connectivity with aPFC, mPFC, PCC and IPL) (de Voogd et al., 2016; Hermans et al., 2017; Murty et al., 2017). Thus, our observed offline hippocampal-neocortical connectivity changes may contribute to emotion-charged retroactive memory benefit, likely through functional reorganization mechanisms of memory-related brain systems at both pre- and post-emotional-learning phases.” and “In line with emotion-charged retroactive memory benefit and increased trial-specific reactivation for associated events, our observed hippocampal connectivity changes during emotional learning and post-learning rests most likely reflect relational and/or contextual processes underlying memory integration, but not due to an indiscriminate generalization or persistence of emotional arousal during learning and post-learning rest (Hermans et al., 2017; Tambini and Davachi, 2013; Tambini et al., 2010). ”

5. The post-encoding analyses need to have additional controls. Ideally, the authors would run an interaction analysis to characterize regions in which post-encoding coupling predicts Emotional but not Neutral memory. Further, control analyses in sensory regions that are not directly related to the stimuli (such as PPA) would be beneficial.

We appreciate the Reviewer for this good point. We have now undertaken the following steps to investigate the potential interaction effects on how post-encoding coupling changes predicted aversive but not neutral memory. We first computed the difference maps of hippocampal connectivity during post-initial-learning rest (relative to before: Rest 2 minus Rest 1), as well as during post-emotional-learning rest (relative to before: Rest 3 minus Rest 2). Thereafter, we submitted these difference connectivity maps (i.e., Rest 2 vs. 1, and Rest 3 vs. 2) into the second-level multiple regression models, with memory accuracies in aversive and neutral conditions as two separate covariates of interest. We then conducted a contrast to test the difference in regression coefficients between aversive and neutral conditions – that is the interaction effects of post-encoding hippocampal connectivity in relation to memory in the aversive versus neutral condition. This interaction analysis identified significant clusters in the LOC (Rest 2 vs. Rest 1), and the inferior parietal lobule (IPL), the posterior cingulate cortex (PCC), the anterior prefrontal cortex (aPFC) and the medial prefrontal cortex (mPFC) (Rest 3 vs. Rest 2) (see Figure 4B and C, Figure 4—figure supplement 1 and Table supplement 4). Specifically, we found that post-encoding hippocampal connectivity with stimulus-relevant LOC at post-initial-learning rest positively predicted associative memory in the aversive (controlling for neutral memory, partial r = 0.52, *p* = 0.007) but not neutral (controlling for aversive memory, partial r = -0.50, *p* = 0.010) condition (see Figure 4—figure supplement 1A). We also found that post-encoding hippocampal connectivity with transmodal prefrontal-parietal regions (i.e., IPL, PCC, aPFC and mPFC) at post-emotional-learning rest positively predicted memory in the aversive (controlling for neutral memory, IPL: partial r = 0.61, *p* = 0.001; PCC: partial r = 0.55, *p* = 0.003; aPFC: partial r = 0.56, *p* = 0.003; mPFC: partial r = 0.54, *p* = 0.004) but not neutral (controlling for aversive memory, IPL: partial r = -0.61, *p* = 0.001; PCC: partial r = -0.54, *p* = 0.004; aPFC: partial r = -0.56, *p* = 0.003; mPFC: partial r = -0.57, *p* = 0.002) condition (see Figure 4—figure supplement 1B*).* Besides the regions reported in our original manuscript with memory difference (i.e., aversive vs. neutral) as a single covariate of interest (see Author response image 2), this interaction analysis revealed another significant cluster in mPFC, a core default network structure. Together, these new interaction results replicated and complemented our original findings. It suggests a shift of post-encoding hippocampal connectivity from stimulus-relevant neocortex to transmodal prefrontal-parietal areas (i.e., the default network) at rest states, which could contribute to our observed emotion-charged retroactive memory enhancement.

**Author response image 2. sa2fig2:** Hippocampal connectivity changes from baseline rest (Rest 1) to post-initial-learning rest (Rest 2) in relation with general/non-arousal memory performance. (A) Significant clusters show their greater connectivity with the hippocampus at Rest 2 relative to Rest 1, with average memory across aversive and neutral conditions as the covariate of interest. (B) Significant clusters show their greater connectivity with the hippocampus at Rest 2 relative to Rest 1, with memory in neutral condition as the covariate of interest. Notes: Color bars represent T values; L, left; R, right.

**Author response table 1. sa2table1:** Post-encoding PPA connectivity changes in negative relation to memory in the aversive but not neutral condition.

Brain Regions	Hemisphere	T values	MNI Coordinates		
			X	Y	Z
Rest 2 vs. Rest 1					
Angular	R	-4.74	44	-58	40
Precuneus	R	-7.12	10	-52	28
					
Rest 3 vs. Rest 2					
Superior frontal gyrus	L	-4.93	-20	-4	74
Inferior frontal gyrus	L	-5.38	-36	30	14
Inferior Parietal Lobule	L	-4.61	-40	-50	58
Supramarginal gyrus	L	-4.83	-52	-30	28
Precentral gyrus	L	-8.48	-50	0	6
	L	-4.62	-34	-2	62
Lingual gyrus	L	-5.17	-12	-90	-8

With respect to additional control analyses, it is worth noting that we did not find any significant cluster in other sensory regions including the parahippocampal place area (PPA) in our post-encoding hippocampal connectivity analyses. Thus, we assume that the Reviewer suggested conducting parallel post-encoding connectivity analyses with PPA as a seed of interest. We therefore defined the posterior parahippocampal gyrus with the WFU PickAtlas toolbox as a PPA mask (Epstein et al., 1999; Epstein and Ward, 2010). However, the posterior parahippocampal area is not only a key sensory region processing visual scenes, but also involved in the encoding of face-object associations into an integrated representation (Qin et al., 2007). To avoid potential interference, we further restricted the PPA seed by removing its overlapping area with two activation contrast maps of encoding during initial and emotional learning as relative to fixation (i.e., all encoding trials vs. fixation during each learning phase), by a less stringent height threshold of *p* < 0.01 and an extent threshold of *p* < 0.05 corrected for multiple comparisons (see Figure 4D). Thereafter, we computed PPA-seeded functional connectivity maps for Rest 1, 2, and 3. These analyses revealed no any reliable clusters with positive effects survived for post-initial-learning (i.e., Rest 2 vs. Rest 1) and post-emotional-learning (i.e., Rest 3 vs. Rest 2) PPA connectivity, in relation to the above-mentioned interaction effect between aversive and neutral conditions. But there were several clusters with significant negative effects, which is out of scope of our study (see Author response table 1). Thus, we did not observe any reliable effect for PPA connectivity with other regions, indicating that our observed offline reorganization of hippocampal connectivity appears only for regions or connectivity pathways relevant to stimuli and tasks used in this study.

Notes: Regions were derived from the multiple regression analyses on post-encoding PPA-seeded functional connectivity with memory accuracies in aversive and neutral conditions as two separate covariates of interest. Clusters, significant at a height threshold of p < 0.001 and an extent threshold of p < 0.05 corrected for multiple comparisons, are reported with local maximum T statistic in Montreal Neurological Institute (MNI) space. L, left; R, right.

In sum, outcomes from our additional multiple regression analyses provide further evidence to strengthen our claim on a shift of post-encoding hippocampal connectivity from stimulus-relevant neocortex to transmodal prefrontal-parietal areas during offline rests before and after emotional learning. We have now updated the Results ( Page 13; Figure 4) and Methods (Page 23-24) sections with these new analyses in our revised manuscript. The other corresponding tables and graphs were included in the Supplemental Materials (Table supplement 4 and Figure 4—figure supplement 1). Additionally, the control PPA region was not involved in this rapid hippocampal-neocortical functional reorganization. We have now added several sentences to incorporate these additional control results into the Results section of our revised manuscript. It now reads as follows ( Page 13): “Additionally, we conducted parallel control analyses with the parahippocampal place area (PPA) as a seed that is a sensory region but not related to the stimuli used in our study. These control analyses revealed no reliable clusters survived for PPA connectivity during post-learning rests in relation with emotion-charged retroactive memory benefit (Figure 4D). It indicates that our observed offline hippocampal-neocortical functional reorganization is a special process involving with stimuli- and memory-related regions (i.e., LOC, IPL, PCC, aPFC and mPFC), but not with irrelevant brain regions (i.e., PPA).”

We hope that the Reviewer is now convinced by outcomes from our additional analyses for interaction effects on post-encoding hippocampal connectivity between aversive and neutral conditions.

Reviewer #2:The manuscript by Zhu and colleagues examines the retroactive influence of emotional 'tags' on associative memory for tagged items, providing new behavioral evidence for a retroactive memory benefit associated with emotional arousal. This is replicated across a few behavioral cohorts and is importantly distinct from prior demonstrations that emotion can enhance item memory for material that is conceptually related to 'tagged' information. fMRI is used to examine possible neural mechanisms for the retroactive benefit: greater reactivation of initial learning patterns is present for the emotionally tagged condition, and hippocampal interactions are enhanced in this condition as well. There is also evidence for post-tagging resting hippocampal interactions that predict the emotional-tagging memory benefit, suggesting that both mechanisms at the time of 'tagging', and subsequent mechanisms may mediate this behavioral effect. I think the main results in the manuscript are sound and advance the literature.

We thank the Reviewer for the enthusiasm and positive evaluation of our manuscript. We are encouraged by the Reviewer’s commendation that “our main results in the manuscript are sound and advance the literature”. We also appreciate the thoughtful and constructive comments that improved our manuscript.

I think there a few points that need to be considered, however.Main points:1. The reinstatement effect (greater for emotional vs. neutral trials) is a compelling demonstration that is consistent with the retroactive memory benefit. A couple of points to consider regarding this analysis:a. The current approach creates an average pattern for 'aversive' and 'neutral' stimulus pairs, both during tagging and initial learning. Thus, greater reinstatement for the aversive condition likely reflects category-level reinstatement of face/object information present during learning, rather than the specific reinstatement of a particular pair (which is implied in the discussion, e.g. "stimulating reactivation of overlapping memory traces" on line 414). Did the authors also examine evidence for pair-specific reinstatement (i.e. greater similarity between the tagging of Face A and encoding trial of Face A, vs. the tagging of Face A and encoding of Face B)? Evidence for trial-specific reinstatement would provide a tighter mechanistic link between reactivation and enhanced associative memory associated with emotional tagging.b. At face value, the relationship between reinstatement evidence and associative memory (in Figure 2D) helps to link reinstatement with subsequent memory. However, given that the relationship is not specific (present for both emotional and neutral conditions) it makes me wonder whether there are alternative explanations beyond reactivation explaining better memory. Since reinstatement is a simple relationship between across-trial encoding and tagging phases, it is possible that participants with less reliable or consistent activity patterns during either the encoding or tagging phase alone (due to fluctuations in attention or other factors) may have worse memory, which may drive the correlation. This can be addressed by (1): showing that the similarity within encoding, i.e. across trials or conditions is not predictive of memory, (2): the same for the tagging phase, and/or (3): by examining evidence for pair-specific reinstatement as described above, which is a more specific measure.

We appreciate the Reviewer for these thoughtful comments and suggestions. To examine (a) evidence for pair-specific reinstatement, we conducted additional analyses for trial-by-trial neural activation pattern similarity. This point is similar to the Reviewer #1’s point 3 (please also see our response to point 3 above). Briefly, we first computed the pair-specific multi-voxel pattern similarity between initial learning and emotional learning phases for each specific pair (i.e., correlation between the arousal trial pattern of Face A and encoding trial pattern of Face A), and across-pair similarity among different pairs (i.e., averaging all correlations between the arousal trial pattern of Face A and encoding trial patterns of Face B, C, D…) (see Figure 2—figure supplement 3). To examine whether there is reliable evidence for emotion-charged pair-specific reinstatement, we then conducted separate 2 (Emotion: aversive vs. neutral) by 2 (Measure: pair-specific vs. across-pair) repeated-measures ANCOVAs in each ROI, with individual’s univariate activation differences (aversive vs. neutral) in both initial and emotional learning phases as covariates of no interest (see Figure 2B). These analyses revealed (marginally) significant Emotion-by-Measure interaction effects in the hippocampus and LOC (see our response to point 3 for statistics). Post-hoc comparisons (with covariates controlled) revealed marginally significant greater pair-specific than across-pair similarity in both the hippocampus and LOC in the aversive condition, but not in the neutral condition (see our response to Q1-3 for statistics). It also revealed the significant greater pair-specific similarity in the aversive (vs. neutral) condition in the hippocampus and LOC (see our response to point 3 for statistics). No significant pair-specific effect showed in the FFA which was interfered with the visual input of face and did not reflect pure reactivation during emotional learning. Altogether, these results provided prominent trial-specific evidence to strengthen our original claim that the observed retroactive memory benefit was most likely resulted from emotional arousal stimulating reactivation of overlapping memory trace for each specific pair in the hippocampus and LOC.

Moreover, we conducted machine-learning based prediction analyses to examine the relationships between pair-specific/across-pair similarity and associative memory performance (see Figure 2C and Figure 2—figure supplement 5). These analyses revealed that hippocampal pair-specific similarity positively predicted memory in the aversive (r_(predicted, observed)_ = 0.46, *p* = 0.008) but not neutral condition (r_(predicted, observed)_ = -0.09, *p* = 0.480). Further Steiger’s test (Steiger, 1980) revealed a significant difference in correlation coefficients of hippocampal pair-specific similarity between aversive and neutral conditions (z = 2.54, *p* = 0.011). Hippocampal across-pair similarity positively predicted memory in both aversive (r_(predicted, observed)_ = 0.43, *p* = 0.014) and neutral (r_(predicted, observed)_ = 0.50, *p* = 0.001) conditions, but with no significant difference in correlation coefficients between two conditions (z = -1.09, *p* = 0.275). The prediction effects of hippocampal pair-specific relative to across-pair similarity on memory were non-significant in both conditions (both r_(predicted, observed)_ < 0.10, *p* > 0.201). These results indicated that both trial-level reinstatement (i.e., pair-specific similarity) and category-level representation (i.e., across-pair similarity) could explain better memory. The category-level representation had a general predication effect for memory regardless of emotion arousal. However, the trial-specific reinstatement showed an emotion-bias predictive effect for memory in the aversive condition.

To further address (b) whether there are alternative explanations beyond reactivation explaining better memory, we conducted a set of new trial-by-trial pattern similarity analyses. Besides pair-specific and across-pair similarity between two phases outlined above, we also computed two other similarity measures across trials: (1) within-encoding similarity (i.e., averaging all correlations among patterns of face-object pairs within initial learning phase), and (2) within-arousal similarity (i.e., averaging all correlations among patterns of face-voice pairs within emotional learning phase) (see Figure 2—figure supplement 3). Thereafter, we conducted machine-learning prediction analyses of these two similarity measures with associative memory performance. As shown in Figure 2—figure supplement 6, we did not find any reliable relation of within-encoding similarity (aversive: r_(predicted, observed)_ = -0.20, *p* = 0.676; neutral: r_(predicted, observed)_ = -0.11, *p* = 0.543) and within-arousal similarity (aversive: r_(predicted, observed)_ = -0.28, *p* = 0.960; neutral: r_(predicted, observed)_ = -0.28, *p* = 0.960) with memory in both conditions. These results indicated that the reliability or consistency of activity pattern within each phase did not account for our observed emotion-charged retroactive memory benefit.

To summarize, three major findings could be drawn from our above new analyses. First, the emotional learning promoted greater trial-specific reactivation/reinstatement in the aversive than neutral condition in the hippocampus and LOC. Second, such pair-specific reinstatement was predictive of better associative memory in the aversive condition, but not evident in the neutral condition, indicating the specificity of emotional modulation effect. The category-level representation, however, generally predicted better memory regardless of aversive or neutral condition. Third, the pattern reliability or consistency during either initial learning or emotional leaning phase could not explain memory benefit in our study. Altogether, the emotion-charged retroactive memory benefit found in our study is more likely resulted from an emotion-induced increase in trial-specific reinstatement of initial encoding activity, but neither broad category-level representation nor reliable/consistent activity pattern within each phase. We have now added these new results into the Results (Page 7-9; Figure 2C), and have also updated the Discussion (Page 14-15) and Methods (Page 22) sections accordingly of our revised manuscript. The other corresponding graphs have been included in the Supplemental Materials (Figure 2—figure supplement 5 and Figure 2—figure supplement 6).

2. The differences in hippocampal interactions based on emotional tagging (Figure 3D,E) reveal greater coupling with the amygdala and FFA/LOC during emotional vs. neutral 'tagging', consistent with the retroactive emotional memory enhancement. The further analyses in Figure 3F,G indicate, however, that these differences are not specifically linked with the emotion-related memory enhancement and instead show main effects of memory and emotion (not an interaction). Despite this lack of differences due to the emotional memory enhancement (across trials) the authors examine whether individual differences in these measures are related to emotionally-tagged memory.a. Please provide logic linking these analyses – why was it expected that these measures would predict emotionally-tagged memory across subjects when trial-level effects relating to emotional memory (specifically) were not found?

Indeed, we only observed the main Memory and Emotion effects for hippocampal connectivity depicted in Figure 3F and G in our original manuscript (now Figure 3E-G in the revised manuscript), but not specifically linked with the emotion-related memory enhancement (no interaction effect). We further conducted machine-learning prediction analysis to investigate the relationships between hippocampal connectivity and emotion-specific memory enhancement across subjects based on the following three aspects. First, evidence from many previous studies introduced in our original manuscript has converged onto linking emotion-modulated hippocampal coupling to subsequent memory benefits for emotional (compared to neutral) condition, suggesting the potential specificity of emotional modulation (de Voogd et al., 2017; Richardson et al., 2004). Second, we found the emotional specificity in reactivation/reinstatement analyses through significant interaction effect and correlation difference between aversive and neutral conditions (see Figure 2B and C). The hippocampal connectivity has been well proposed to coordinate with reactivation together to promote memory integration (Schlichting and Preston, 2014; Sutherland and McNaughton, 2000). Thus, we assume that hippocampal connectivity might also show pontential emotional specificity in some aspect. Third, it is not surprising that we found significant correlation difference but not interaction effect in hippocampal connectivity analyses, due to the different principles of these two statistical approaches. The interaction effect in ANOVA, based on group mean and variance, emphasizes different effects on connectivity itself between conditions (Nieuwenhuis et al., 2011). No significant interaction effect found in our study indicated that hippocampal connectivity for high-confidence remembered associations compared with forgotten ones was not significantly greater in the aversive than neutral condition. However, the prediction analysis catches up the linear covariation of connectivity and memory performance (Aggarwal and Ranganathan, 2016). Our results of significant correlation difference indicate that hippocampal connectivity (i.e., remembered with high confidence relative to forgotten) could better predict memory performance in the aversive than neutral condition (see our response to 2b below). In other words, emotion-charged hippocampal connectivity showed more direct and tight relationship with memory performance than that in neutral situation.

Based on above theoretical motivation, empirical observations as well as statistical principles, we have clarified the logic linking the ANOVAs and further prediction analyses on hippocampal connectivity more clearly in the revised manuscript. It now reads as follows:

Page 9: “Given the modulatory effects of emotion for hippocampal connectivity reported in previous studies (de Voogd et al., 2017; Richardson et al., 2004) and emotion-charged neural reactivation in relation to associative memory obeserved in our study, we assumed that hippocampal connectivity, which works in concert with reactivation to promote memory integration (Schlichting and Preston, 2014; Sutherland and McNaughton, 2000), might also show emotion-specific relationship with associative memory. Although we did not find reliable interaction effect in above ANOVA analyses, we employed machine learning-based prediction analysis approach to investigate whether greater hippocampal coupling during emotional learning could predict better memory and whether this relationship would show emotional specificity.”

We would like to thank the Reviewer again for prompting us to clarify this point.

b. Please also show the correlations that are described in the text (lines 327-335). It is a bit surprising to see the come up as boxes in the SEM analysis (Figure 3H) although they are not shown individually first. Do the correlations between memory and hippocampal connectivity (w/ amygdala and FFA/LOC) significantly differ between the emotionally-tagged and neutral conditions? The correlations are significant for the emotional condition, but not the neutral condition, implying specificity although this is not shown/tested. Or, the analysis can be done with the difference in memory for emotional vs. neutral conditions to gain this specificity, which was used for the rest analyses in Figure 4.

We thank the Reviewer for this suggestion. We have now conducted connectivity analyses in the FFA and LOC separately, instead of combining them as an overall neocortical region in our original manuscript, to elaborate our observed mechanism with regional specificity. This also was suggested by the Reviewer 3’s point 4. The correlations are shown in Figure 3H-J and Figure 3—figure supplement 2 . We then conducted Steiger’s tests to examine the difference between correlations in aversive and neutral conditions in each ROI.

**Author response table 2. sa2table2:** Hippocampal connectivity changes at post-initial-learning rest (Rest 2 vs.1).

Brain Regions	Hemisphere	T values	MNI Coordinates		
			X	Y	Z
Related to average memory					
Superior temporal gyrus	R	3.69	54	0	-6
Superior temporal pole	R	3.99	54	16	-20
Insula	L	-3.62	-38	14	10
Cuneus	R	-3.84	20	-66	24
Supramarginal gyrus	R	-3.46	58	-32	36
Postcentral gyrus	L	-3.65	-54	-8	22
	R	-3.76	64	0	20
					
Related to memory in the neutral condition					
Superior temporal gyrus	R	3.87	54	0	-6
Thalamus	R	3.56	20	-16	6
Postcentral gyrus	R	-3.57	64	0	20
Cuneus	R	-3.57	20	-66	24
Supramarginal gyrus	R	-3.36	58	-32	36

To emphasize our main goal of the prominent emotional effects on hippocampal connectivity in the aversive condition, we have now added the aversive correlation graphs in the Results section of our revised manuscript (Figure 3H-J) and included the corresponding neutral graphs in the Supplemental Materials (Figure 3—figure supplement 2). If the Reviewer still prefers to add all correlation graphs including neutral ones in our revised manuscript, we are also happy to do that.

We have also updated the Results section of our revised manuscript (Page 9-11), and it now reads as follows: “Interestingly, we found that greater hippocampal connectivity with the amygdala, FFA and LOC (i.e., remembered with high confidence relative to forgotten) were predictive of better associative memory (i.e., remembered with high confidence) in the aversive condition (amygdala: r_(predicted, observed)_ = 0.24, p = 0.076; FFA: r_(predicted, observed)_ = 0.57, p < 0.001; LOC: r_(predicted, observed)_ = 0.65, p < 0.001; Figure 3H-J), but not in the neutral condition (all r_(predicted, observed)_ < 0.15, p > 0.160; Figure 3—figure supplement 2). Further Steiger’s tests revealed significant differences in correlation coefficients between two conditions for hippocampal coupling with FFA (z = 2.13, p = 0.033) and LOC (z = 2.15, p = 0.032), and marginally significant difference for hippocampal-amygdala coupling (z = 1.79, p = 0.073). These results indicate that emotion-charged hippocampal connectivity with the amygdala and stimulus-sensitive neocortical regions positively predicts associative memory for initial neutral events in the aversive but not neutral condition, also implying an emotional specificity effect.”

c. Lastly, logic should also be provided for constructing the SEM in the specific manner it was setup (hippocampal-cortical connectivity serving as the mediating variable between hipp-amygdala connectivity and memory).

Consistent with above correlation analyses, we have now constructed the SEM with hippocampal-FFA/LOC connectivity as separate mediators to account for the positive relationship between hippocampal-amygdala connectivity and associative memory (see Figure 3K).

This SEM was constructed based on the following considerations. First, as we introduced in our original manuscript, emotion-induced memory enhancement is most likely based on the modulatory role of the amygdala on hippocampal communication with the neocortex (Hamann, 2001; LaBar and Cabeza, 2006). This modulatory role may support more efficient information transmission and communication between the hippocampus and related neocortical regions (Alvarez et al., 2008; Hermans et al., 2017; Phelps and LeDoux, 2005), and thereby further lead to emotion-charged memory enhancement. Thus, we theoretically assumed that hippocampal-amygdala connectivity predicted memory most likely through hippocampal-neocortical dialogue. Second, the correlation analyses revealed that memory in the aversive condition was not only marginally correlated with hippocampal-amygdala connectivity (see Figure 3H), but also highly correlated with hippocampal-FFA/LOC connectivity (see Figure 3I and J). Therefore, we assumed the potential mediatory pathways among these variables, that hippocampal-amygdala connectivity might indirectly account for emotion-charged memory performance through the mediation of hippocampal-FFA/LOC connectivity. Third, besides above theoretical motivation as well as empirical observations, we conducted additional control analyses to examine alternative mediation and modulation models (see Figure 3—figure supplement 3) using the procedures similar to previous studies (Burghy et al., 2012; Jiang et al., 2020; Zhu et al., 2019). We observed no reliable mediating or modulating effect in these alternative models, practically indicating that our data did not support other models.

We have now clarified the above logic for the SEM construction and updated the Results section of our revised manuscript. We have also included the alternative models in the Supplemental Materials (Figure 3—figure supplement 3). It now reads as follows:

Page 11: “The modulatory role of the amygdala on hippocampal dialogue with the neocortex is recognized to promote more efficient information transmission and communication between the hippocampus and related neocortical regions (Alvarez et al., 2008; Hermans et al., 2017; Phelps & LeDoux, 2005), which could further lead to emotion-charged memory enhancement. In addition, according to our above empirical observations, memory performance in the aversive condition was not only marginally correlated with hippocampal-amygdala connectivity, but also highly correlated with hippocampal-FFA/LOC connectivity. Therefore, we assumed the potential mediatory pathway among these variables, that is, hippocampal-amygdala connectivity would indirectly account for emotion-charged memory performance through the mediation of hippocampal-FFA/LOC connectivity. Based on above theoretical motivation as well as empirical observations, we further implemented structural equation modeling (SEM) to investigate how the hippocampus, amygdala and neocortical systems during emotional learning work in concert with each other to support emotion-charged retroactive memory enhancement.

Specifically, we constructed a mediation model with hippocampal-amygdala connectivity (i.e., remembered with high confidence relative to forgotten) as input variable, hippocampal-FFA and -LOC connectivity pathways as two separate mediators and individual’s associative memory performance (i.e., remembered with high confidence) as outcome variable in the aversive condition (Figure 3K). The SEM analysis revealed a good model fit for our data (Chi^2^ = 1.88, p > 0.050; RMSEA = 0.00; SRMR = 0.035; CFI = 1.00). The two mediating effects on the positive relationship between hippocampal-amygdala connectivity and memory outcome were both significant (hippocampal couplings with FFA and LOC: Indirect Est = 0.19, p = 0.010, 95% CI = [0.044, 0.328], and hippocampal–LOC coupling: Indirect Est = 0.20, p = 0.016, 95% CI = [0.036, 0.360]). That is, hippocampal-amygdala connectivity affected hippocampal-LOC connectivity through a partial mediating effect of hippocampal-FFA connectivity, which could ultimately account for emotion-charged associative memory performance in the aversive condition. Notably, we also conducted additional control analyses for a set of alternative mediation and modulation models in the aversive condition (Figure 3—figure supplement 3). These analyses, however, did not reveal any reliable mediating or modulating effect, indicating that our data did not support such alternative models. Since there was no significant relationship between hippocampal connectivity and associative memory in the neutral condition, we did not conduct SEMs in this condition. Altogether, these results indicate that increased hippocampal-amygdala coupling indirectly accounts for emotion-charged associative memory, mediating by hippocampal-neocortical couplings during emotional learning.”

We have further added several sentences to discuss above results in the Discussion section ( Page 15-16), and it now reads as follows: “This mediation effect suggests that emotional arousal directly paired with face cues could induce greater hippocampal-amygdala connectivity acting on hippocampal-FFA connectivity during emotional learning, which then stimulated the cued trial-specific reactivation in the hippocampus and LOC through increasing hippocampal-LOC interaction, and ultimately contributed to retroactive memory benefit for related events.”

3. The primary rest analysis – which examines changes in hippocampal connectivity (rest-3 minus rest-2) that are related to individual differences in emotionally-tagged memory enhancement – is clear and well-motivated. However, the logic underlying the first analysis, which relates changes in connectivity from rest-1 to rest-2 (baseline to post-initial-learning rest) to the emotionally-tagged memory enhancement, is not clear. Given that emotional tagging occurs after rest-2, there should be no meaningful signature associated with biased consolidation of this material prior to this time point. I understand that the authors are trying to show how emotional tagging, per se, alters post-encoding consolidation signatures during rest above and beyond those that are present after initial learning alone. I infer that the goal is to compare the post-tagging rest to an analysis that illustrates simple relationships between immediate post-encoding connectivity and later memory (not related to tagging). To achieve this goal with the current design, I would suggest a more straightforward analysis of examining changes in hippocampal connectivity from rest-1 to rest-2 that are related to individual differences in associative memory that are not related to tagging, such as the average across neutral and emotionally-tagged stims (or perhaps neutral stimuli that are not emotionally tagged). Average associative memory would likely be the best measure, considering that emotional tagging may enhance memory for related material at the cost of impaired memory for non-tagged material. There is clear logic in this analysis, in that it would reveal initial post-encoding resting connectivity that is related to memory and NOT the emotional tagging procedure. The contrast of this kind of analysis with the rest-3 minus rest-2 analysis would then be clear. Note that the main analysis (examining changes from rest-2 to rest-3) on its own somewhat controls for initial post-encoding activity (captured in rest-2), so perhaps this other analysis is not needed (although the claims made in the Discussion would have to be altered).

We appreciate the Reviewer for raising this thoughtful comment and suggestion on how to characterize post-encoding hippocampal connectivity changes from rest-1 to rest-2 linked to subsequent memory performance. First of all, we would like to clarify the logic for our analyses of post-encoding hippocampal connectivity changes. As the Reviewer pointed out, our goal is indeed to compare hippocampal connectivity changes during post-emotional-learning rest with that during post-initial-learning rest in order to predict the same subsequent memory outcome – that is, the emotion-charged retroactive memory enhancement (i.e., interaction effects for post-encoding hippocampal connectivity changes between aversive and neutral conditions). This strategy allows us to test whether neural engagement prior to emotional learning contributes to the later emotion-charged memory benefits, implicated in memory allocation and integration models introduced in our original manuscript. These models propose that a preparatory state with relevant neuronal excitability may actively affect the way of how newly acquired memories are allocated and integrated into existing neural networks (Park and Rugg, 2010; Schlichting and Frankland, 2017; Yoo et al., 2012). In addition, from a perspective of statistical comparisons, this analytic strategy allows us to directly compare post-encoding connectivity changes as a function of before and after emotional learning, while holding the dependent variable same. We have clarified the logic of our above analytic strategy and also adjusted the discussion in our revised manuscript. It now reads as follows:

Page 12 in Results: “The offline neurocognitive processes along with hippocampal-neocortical functional reorganization are thought to support episodic memory (Liu et al., 2022; Tambini et al., 2010). Considering the potential influences of memory-related brain network configurations and representations both prior to and after emotional learning on subsequent memory outcomes proposed in memory allocation and integration models (Schlichting and Frankland, 2017; Schlichting and Preston, 2014), we thus investigated whether and how hippocampal connectivity changes at resting states after initial learning and emotional learning contribute to emotion-charged retroactive memory benefit.”

Page 16 in Discussion: “Neural engagement and circuit connectivity before emotional learning (i.e., hippocampal-LOC connectivity during post-initial-learning rest) serve as a preparatory state that could modulate allocation of new information into overlapping neural populations to support later memory integration (Park and Rugg, 2010; Schlichting and Frankland, 2017; Yoo et al., 2012).”

For the Reviewer’s suggestion on additional new analyses using the average memory performance (or neutral memory), we are afraid that this measure might be contaminated to some extent by emotion-induced retroactive reorganization of previously encoded memories. Given our experimental procedure (i.e., an initial learning phase, followed by an emotional learning phase, and finally a surprise memory testing) and the prominent effects of emotional learning, we thus think subsequent emotional learning and post-emotional-learning rest might overwrite the effect of post-initial-learning hippocampal connectivity changes on final memory performance. Ideally, future studies with memory test before emotional learning are required to directly address how hippocampal connectivity changes link to non-arousal associative memory.

Following the Reviewer’s suggestion, nevertheless, we have also conducted several additional analyses to examine hippocampal connectivity changes from rest-1 to rest-2 that are related to individual differences in non-arousal associative memory. We first conducted a second-level multiple regression analysis for post-encoding hippocampal connectivity changes from rest-1 to rest-2, with the average associative memory (collapsing across aversive and neutral conditions) as a covariate of interest. As expected, this analysis revealed no significant clusters at the same thresholding criteria used for other analyses in our present study (i.e., a height threshold of p < 0.005 and an extent threshold of p < 0.05 corrected for multiple comparisons). Likewise, parallel analysis for memory in the neutral condition also revealed no significant cluster on the whole brain level.

To further explore potential regions related to the average memory at post-initial-learning rest 2 (relative to baseline rest 1), we applied a relatively less stringent height threshold of *p* < 0.01 and a spatial extent cluster size of more than 30 voxels. We observed clusters in the superior temporal gyrus (STG) and other regions, with greater post-encoding hippocampal connectivity linked to the average associative memory (or memory in the neutral condition) (see Author response image 2 and Author response table 2). We feel that such pattern of weak results might be due to the fact we outlined above – that is, emotional learning and its relevant rests prominently modulated the subsequent memory performance, which covered the effects of initial learning.

**Author response table 3. sa2table3:** A set of fMRI studies with moderate sample size (N = 16 to 35).

	Author	Journal	Year	Sample Size (effective)	Main conclusion	Correlations(including mediating effects)
1	Günseli,Aly	*eLife*	2020	N=29	Hippocampus and vmPFC support memory-guided attention.	Correlation between vmPFC activity and hippocampal activity by skipped_pearson_correlation.m function (*Figure 3B*).
2	Keogh, Bergmann, Pearson	*eLife*	2020	N=32	Cortical excitability is linked to individual differences in the strength of mental imagery.	Spearman rank correlations between cortical excitability and imagery strength (Figure 3 and also see Study design in Materials and methods).
3	Liu et al.	Nature Communication	2016	N=18	Consolidation reconfigures neural pathways underlying the suppression of emotional memories.	Correlations between hippocampal functional connectivity/ pattern dissimilarity and behavioral suppression score by prediction analyses (*Figure 4 and 6*).
4	Gruber et al.	Neuron	2016	N=19	Post-learning hippocampal dynamics predict reward-related memory advantages.	Pearson’s correlations between hippocampal functional connectivity/ reactivation and memory benefits (*Figure 2B and 3C*).
5	Schlichting, Preston	PNAs	2014	N=35	Memory reactivation and hippocampal–neocortical functional connectivity during rest support subsequent learning.	Partial correlations between FFA reactivation/FFA-HPC connectivity and memory performance (*Figure 2 and 3*).
6	Wimmer, Shohamy	Science	2012	N=28	Hippocampal activation, reactivation, and coupling predict decision bias.	Mediating effects of visual reactivation on the relationship between hippocampal activity and striatum activity.
7	Tambini,Ketz, Davachi	Neuron	2010	N=16	Offline hippocampal-cortical interactions relate to subsequent associative memory.	Pearson’s correlations between offline hippocampal−LO correlations and associative memory (*Figure 3D*).

**Author response table 4. sa2table4:** Statistical power for each significant correlation in main results (two-tailed tests).

	Correlations	Sample Size(n)	Significance Criterion(α)	Effect size(r)	Power(1-β)
Pattern similarity	Correlation between pair-specific pattern similarity and associative memory in aversive condition *(Figure 3C)*	28	0.05	0.46	0.72
Task-dependent functional connectivity	Correlation between hippocampal-FFA connectivity and associative memory in aversive condition *(Figure 5E)*			0.57	0.91
	Correlation between hippocampal-LOC connectivity and associative memory in aversive condition *(Figure 5F)*			0.65	0.98

**Author response image 3. sa2fig3:** Additional condition-level pattern similarity results. Bar graphs depict the average pattern similarities in aversive and neutral conditions during the presentations of face-voice associations (i.e., Face + Voice) separately for the bilateral hippocampus, bilateral ventral LOC (vLOC) and bilateral FFA. Error bars represent standard error of mean. Notes: *p < 0.05; **p < 0.01; two-tailed t-tests.

Notes: Regions were derived from the multiple regression analyses on post-encoding hippocampal functional connectivity with average/neutral memory as the covariate of interest. Clusters, significant at a height threshold of p < 0.01 and a spatial extent cluster size of more than 30 voxels, are reported with local maximum T statistic in Montreal Neurological Institute (MNI) space. L, left; R, right.

Given the central goal we clarified above and the additional results above which cannot survive at the general thresholding criteria used for other analyses in our study, we therefore decide not to include them into our revised manuscript. Nevertheless, we have added one sentence to incorporate this point into the Discussion section ( Page 17), and it now reads as follows: “However, we did not find any reliable results of post-initial-learning hippocampal connectivity changes in relation to neutral or averaged memory performance, as subsequent emotional learning and post-emotional-learning rest might overwrite the effect of initial learning on final memory performance. Future studies are required to test memory before emotional learning as well to disentangle post-learning signatures linked to neutral and emotion-charged memory performances separately.”

We hope that the Reviewer now agrees with us on the logic and the central goal of our analyses for post-encoding hippocampal connectivity changes in relation to emotion-charged memory benefit. We would like to thank the Reviewer again for these thoughtful comments and suggestion, which also inspire us to think about the goal and logic of our analyses as well as data interpretation via a more in-depth manner.

Reviewer #3:In this paper, Zhu and colleagues report a series of studies (2 behavioral and 1 fMRI) designed to test the retroactive effects of aversive associations-- which they call "emotional tagging"-- on memory for previously-learned, overlapping neutral associations. Across the 3 studies, they found that neutral face-object associations were remembered better if the face was later paired with an aversive sound. In the fMRI study, the authors additionally showed that the tagging phase was associated with greater pattern reactivation when new associations were aversive, compared to neutral. They also found changes in functional connectivity with the hippocampus that were associated with the retroactive effects of tagging, both during the tagging phase itself and during a post-tagging resting-state scan.This paper had several strengths. The behavioral findings were compelling, showing the same general pattern across the 3 studies. The topic is timely, as there is accumulating evidence for the retroactive effects of emotion on memory, but little neural data explaining such effects in humans. The analysis approach included multiple sophisticated methods.

We appreciate the Reviewer’s positive assessment of our work by commending that “this paper has several strengths” and “the topic is timely”.

However, there were also significant weaknesses that limit the impact of the paper.1. The first major weakness is the sample size of the fMRI study (effective N=28), coupled with the fact that many of the conclusions were based on correlations with individual differences in memory-- including the findings that hippocampal reactivation and changes in functional connectivity predicted retroactive memory benefits. This is a concern because correlations across small sample sizes are less likely to replicate in future work. The mediation model is subject to these same concerns.

We thank the Reviewer for this comment on the sample size of our fMRI study. We have now undertaken several strategies to ensure the reproducibility and robustness of our results. First of all, we would like to take this opportunity to clarify that there is several aspects of results supporting our major conclusion on how neural reactivation and connectivity reconfiguration contribute to the emotion-charged retroactive memory benefit. Specifically, our major findings are derived from the prominent retroactive effects of emotional learning on episodic memory across three separate studies (Figure 1B-G), as well as condition-level/trial-specific reactivation/reinstatement (Figure 2B ) and connectivity changes during both learning (Figure 3B-D) and post-encoding states (Figure 4) in the aversive (vs. neutral) condition. Our major conclusions are thus based on the main and/or interaction effects from ANOVAs and stringent whole-brain analyses. The brain-behavior correlations of memory with reactivation (Figure 2C) and functional connectivity (Figure 3H-K) serve as complementary and exploratory results to support our observed specificity of emotion-charged retroactive memory effects. In a word, our main conclusions still hold, even if we toning down our original claims on brain-behavior correlations.

Secondly, we feel that the brain-behavior correlations identified in our study are robust, even with a moderate sample size of 28 participants. The sample sizes (i.e., from 26 to 30 participants) of our three studies were pre-determined by power analysis using G*Power 3.1, which would give us power of around 85-90% for a moderate repeated-measures ANOVA, consistent with many studies with similar analyses on neural reactivation and connectivity (Gruber et al., 2016; Liu et al., 2016; Schlichting and Preston, 2014; Tambini et al., 2010; Wimmer and Shohamy, 2012). These studies with a similar sample size have demonstrated robust correlations of brain activation (or connectivity) patterns with individual differences in memory performance (please see Author response table 3). Besides, we agree that a larger sample size would improve the reproducibility of brain-behavior correlations in general. However, large sample sizes are not a substitute for good hypothesis testing according to the fallacy of classical inference outlined by Karl Friston (Friston, 2012, 2013). It is because large sample sizes increase the probability of rejecting the null hypothesis under trivial treatment effects. A significant result in a proper small sample indicates that the treatment effect is larger than the equivalent result with a large sample.

Thirdly, we have now undertaken three steps to ensure the robustness of our correlation analyses and to improve the integrity of our logic in the revised manuscript. (1) We have implemented a machine learning approach based on linear regression models in combination with balanced four-fold cross-validation algorithms to confirm the robustness of our observed correlations (please see Figure 2C and Figure 3H-J). This approach has been widely used in small samples to confirm the robustness of the predictive relationship between an independent (input) variable and a dependent (outcome) variable. This approach can nicely complement the conventional regression models which are sensitive to outliers and have no predictive value (Cohen, 2010; Geisser, 1993; Qin et al., 2014; Supekar et al., 2013). We have now reported robust r _(predicted, observed)_, estimated based on the average of four repetitions of the four-fold cross-validation procedure, for each correlation and also conducted further analyses (i.e., Steiger’s difference test; please also see our response to points 1a and 2b above) based on it. (2) Referring to some recent studies published on *eLife* (Günseli and Aly, 2020; Keogh et al., 2020), we have now calculated post-hoc power for each significant correlation in our main results. As shown in Author response table 4, the connectivity correlations with large effect sizes (r > 0.5) give high power more than 85% (Cohen, 1992, 2013). The similarity correlation with a moderate effect size (r > 0.45) also gives an acceptable power of 72%. These results statistically prove the power of our sample to detect individual differences. (3) We have implemented a bootstrapping procedure for our mediation analyses to gain the sensitivity and robustness of statistical estimates in small-to-moderate samples (Preacher and Hayes, 2008; Shrout and Bolger, 2002). Such bootstrap approach has been widely used in the field, because it does not impose the assumption regarding to the normal distribution of the sampling data which is usually required by the parametric approaches. In addition, it is helpful for gaining statistical power with small sample sizes and could yield more accurate estimates of the indirect effect standard errors than other conventional approaches (MacKinnon et al., 2004; Shrout and Bolger, 2002).

In sum, although we agree that a larger sample size would strengthen the reproducibility pertaining to our brain-behavior correlation results, it is very challenging to scan sufficient participants during the COVID-19 pandemic. Therefore, we adopted the above-mentioned steps to address this concern on sample sizes for our brain-behavior correlation analyses. We have clarified our sample size and statistical power in the Methods section of the revised manuscript (Page 17), and it now reads as follows: “The sample sizes across three studies were estimated by a power analysis using G*Power 3.1, which yielded the power of around 85-90% (i.e., from 26 to 30 participants) for a moderate repeated-measures ANOVA, consistent with many memory studies (Gruber et al., 2016; Liu et al., 2016; Schlichting and Preston, 2014; Tambini et al., 2010; Wimmer and Shohamy, 2012). In Study 3, the sample size of 28 valid participants could give us power more than 70% for a moderate correlation (i.e., r > 0.45) (Cohen, 1992, 2013).” We have also reported the effect size (i.e., r _(predicted, observed)_) for each correlation in the revised manuscript (Figure 2C and Figure 3H-J). We hope that these steps would strengthen the Reviewer’s confidence on the robustness of our observed brain-behavior correlations. Besides, we have also explicitly acknowledged in the Discussion section of our revised manuscript (Page 17), that: “In addition, larger sample size would be helpful to reinforce the reproducibility of our observed brain-behavior association effects in future studies.”

2. The second major weakness is a lack of theoretical clarity regarding the mechanisms supporting the retroactive memory benefit. Two main ideas are introduced: the idea of behavioral tagging (i.e., as related to synaptic tag-and-capture models) and the idea that reactivation plays an important role in memory allocation and integration. I see these as two separate theoretical perspectives that could make different predictions here, which may have been the authors' intention. However, they are not clearly set up as competing hypotheses, nor are they clearly integrated into a unified account. This made it difficult to interpret the results in light of either account.

We appreciate the Reviewer for the opportunity to clarify the theoretical mechanisms supporting our observed retroactive memory benefit. We have now rewritten the Introduction section of our revised manuscript to clarify our theoretical perspectives and hypotheses. As our response to Q1-1 above, we actually intended to combine both behavioral tagging and memory allocation/integration models (including sensory preconditioning suggested by Reviewer 1) into a unified account from multiple neurobiological levels. Considering new results from additional analyses (also see our response to point 1), however, our observed immediate trial-specific retroactive benefit for associations with high-vividness ratings (i.e., presumably strong memories) is more consistent with sensory preconditioning literature and memory integration models. This clearly differs from the delayed retroactive memory benefit for initial weak memories according to the conventional synaptic tag-and-capture (STC) and behavioral tagging models. Our interpretation from a perspective of sensory preconditioning and memory integration is further supported by evidence from our fMRI data with emotion-induced increases in trial-specific pattern reinstatement/reactivation and emotion-charged online/offline hippocampal-neocortical reorganization. Thus, our central goal now is to put forward a new alternative mechanism (i.e., sensory preconditioning as well as memory integration) underlying trial-specific associative learning circumstance in our revised manuscript, which cannot be readily explained by the behavioral tagging models.

To better clarify our theoretical perspectives and hypotheses, we have now rewritten the Introduction section of our revised manuscript (Page 1-3). In brief, we first introduced a set of well-known phenomena of how emotion reshapes episodic memory, especially emotion-charged retroactive memory benefits (Holmes et al., 2018; Shohamy and Daw, 2015; Wong et al., 2019), and raised an open question on its underlying neurocognitive mechanisms in our integrated memory system. We then focused on the emotion-charged event/trial-specific retroactive effect on episodic memory, which always occurs in our daily life but could not readily accounted by conventional STC and behavioral tagging models (Ballarini et al., 2009; Clewett et al., 2022; Dunsmoor et al., 2015; Takeuchi et al., 2016). Thus we introduced sensory preconditioning literatures, which allows us to investigate the trial-specific effect on associated events at behavioral level and provide more compelling evidence with this effect (Brogden, 1939; Kurth-Nelson et al., 2015; Li et al., 2008; Sadacca et al., 2018; Sharpe et al., 2017; Wimmer and Shohamy, 2012; Wong and Pittig, 2021). Finally, we brought up memory allocation and integration models, which have been well proposed to accommodate sensory preconditioning (Holmes et al., 2021; Schlichting and Frankland, 2017; Schlichting and Preston, 2015; Shohamy and Daw, 2015; Shohamy and Wagner, 2008; Wong et al., 2019), to understand the potential mechanisms underlying the emotion-charged trial-specific retroactive memory benefit at neural level.

We have also adapted the Discussion section of our revised manuscript ( Page 14) to interpret our behavioral and fMRI findings mainly through the lens of sensory preconditioning and memory integration models, and discuss the differences between our findings and prior results reported in behavioral tagging studies.

We feel that our revisions have now improved the theoretical clarity regarding the mechanisms supporting the retroactive memory benefit substantially. We would like to thank the Reviewer again for raising this concern which helped us improve our manuscript.

3. The terms "reorganization" and "reconfiguration" are used to describe the functional connectivity results. I have seen these terms used to refer to network-level changes, but they overstate the functional connectivity differences here, which are simply changes in the maps resulting from a seed-based functional connectivity analysis.

We thank the Reviewer for raising this concern. In our current study, we intend to opt for the terms “reorganization” and "reconfiguration" for two reasons. First, we intend to refer it to “memory reorganization” – that is, episodic memories can be reorganized according to their future significance in support of flexibility and adaptivity (Ritchey et al., 2016; Tambini and Davachi, 2019). Specifically, memory reorganization in our study represents an emotion-charged neurocognitive mechanism, by which neutral episodic memories can be strengthened by subsequent emotional learning, and reorganized into an integrated network to protect from forgetting. Second, we feel that “reorganization” can, at least in part, reflect changes in hippocampal-neocortical functional connectivity in our study, as characterized by (a) an increase in online hippocampal-neocortical functional coupling modulated by the amygdala during emotional learning phase, and (b) a shift of hippocampal-neocortical intrinsic functional connectivity from local stimulus-sensitive neocortex to more distributed prefrontal and posterior parietal areas during post-learning rest. A similar idea of brain functional reorganization has been demonstrated by several previous studies on learning, memory, and among other domains (Bassett et al., 2011; Liu et al., 2016; Mutso et al., 2014; Qin et al., 2014; Sevinc et al., 2019; Zhuang et al., 2022). As such, we agree that the term “reorganization” in our current study appears to reflect changes in hippocampal functional connectivity with other brain regions.

We have now adapted the entire manuscript accordingly to better clarify the use of these two terms when appropriate. We have also adjusted our original statements to avoid potential overstatements or claims on brain network reorganization and reconfiguration in our revised manuscript, given the fact that we only assessed changes in hippocampal-seeded connectivity during initial/emotional learning phases and post-learning rests. We are also happy to replace “reorganization” with another more appropriate term in the title if the Reviewer has any better suggestion. We thank the Reviewer again for bringing this point into our attention, as this could help us improve the scrutiny of using “reorganization” and “reconfiguration” in our revised manuscript.

4. It would be more informative to examine pattern reactivation in face-selective and object-selective areas separately (here, they appear to be lumped together), since only reactivation of object-selective areas should be taken as "pure" evidence of learning-phase reactivation, due to the overlap in faces shown during the learning and tagging phases. If there's an increase in pattern similarity in face-selective areas, this could be explained by changes in attention to the visual stimuli associated with the aversive sound rather than reactivation per se.

This is indeed a good point. Following the Reviewer’s suggestion as well as similar comments by two other Reviewers, we have now conducted several trial-specific multivoxel pattern similarity analyses for face-selective FFA and object-selective LOC separately (please also see our responses to point 3 and point 1*)*. As shown in Figure 2 and Figure 2—figure supplement 4*,* these analyses revealed that emotional learning promoted trial-specific reactivation of initially encoded memories rather than broad category representation in the hippocampus and LOC but not in FFA, contributing to the emotion-charged retroactive memory benefit. We have incorporated these new results into the Results section (Page 7). We have also added several sentences to clarify this point in the Discussion section of our revised manuscript, and it now reads as follows (Page 15): “The object-sensitive LOC showed an increase in trial-specific reactivation of object stimuli, which could reflect reinstatement of corresponding initial face-object associations during emotional learning (Hofstetter et al., 2012; Tambini et al., 2010). But the face-sensitive FFA could not provide pure evidence of reactivation since face information was presented during both phases. These results suggest task-related regional specificity of the observed emotion-charged trial-specific reactivation. In addition, the marginally significant emotional effects for category representation (i.e., across-pair similarity) found in the hippocampus, LOC and FFA might be due to more attention in the aversive (vs. neutral) condition generally attracted by the screaming stimuli, rather than trial-specific reactivation. This explanation is also supported by our condition-level similarity results, showing strengthened reactivation only occurred after the onset of aversive (vs. neutral) voices during emotional learning.”

We feel that our claims on regional specificity of trial-specific reactivation/reinstatement are now more clearly supported. To keep consistency with these reactivation analyses, we have also examined hippocampal connectivity in the FFA and LOC areas separately in the Results section of our revised manuscript (Figure 3). These analyses yielded even more robust and interesting results as compared to our original analyses by lumping these ROIs together. We would like to thank the Reviewer again for this excellent suggestion!

5. The prediction approach was a nice addition here, in that it can help to determine which effects are robust across subjects. However, it appears that they were based on clusters that were already deemed significant through conventional statistical analyses (e.g., see lines 794-795), which suggests that they are biased by the group effects.

We thank the Reviewer for this specific comment. For post-encoding hippocampal connectivity data, significant clusters were indeed pre-selected from whole-brain analyses (i.e., interaction effects in relation to memory in the aversive than neutral condition). We would like to clarify that the predication analyses were implemented only for the confirmatory and visualization purposes. To avoid the potential bias or double dipping, we have now decided to only report the partial correlations with no further statistical inferences, as suggested by the Reviewer #1’s minor comment Q1-7. The correlation plots have now moved into the Supplemental Materials (Figure 4—figure supplement 1).

It is worth noting that we have also conducted the prediction analyses to examine whether neural reactivation and online functional connectivity measures (see Figure 2C and Figure 3H-J) could predict subsequent memory performance. Neuroimaging data in these analyses were derived from significant clusters through the unbiased main effects of encoding (i.e., all encoding trials vs. fixation during each learning phase) and emotional learning (i.e., trials in the aversive vs. neutral condition). This approach is thus independent from the correlation variables of interest. We have now added several sentences to clarify the prediction analyses in the Methods section of our revised manuscript (Page 24). It now reads as follows: “We used a machine learning-based prediction algorithm with balanced fourfold cross-validation (S. Qin, Cho, et al., 2014) to confirm the robustness of relationships of reactivation and online functional connectivity with memory performance. This prediction analysis complements conventional correlation models which are sensitive to outliers and have no predictive value (Cohen, 2010; Geisser, 1993; Supekar et al., 2013).”

We thank the Reviewer again for prompting us to clarify the purposes of our prediction analyses.

References:

Aggarwal R, Ranganathan P (2016) Common pitfalls in statistical analysis: The use of correlation techniques *Perspectives in clinical research* 7: 187.

Alvarez R P, Biggs A, Chen G, Pine D S, Grillon C (2008) Contextual fear conditioning in humans: cortical-hippocampal and amygdala contributions *Journal of Neuroscience* 28: 6211-6219.

Anderson J S, Druzgal T J, Lopez‐Larson M, Jeong E K, Desai K, Yurgelun‐Todd D (2011) Network anticorrelations, global regression, and phase‐shifted soft tissue correction *Human brain mapping* 32: 919-934.

Arcediano F, Escobar M, Miller R R (2003) Temporal integration and temporal backward associations in human and nonhuman subjects *Animal Learning & Behavior* 31: 242-256.

Ballarini F, Moncada D, Martinez M C, Alen N, Viola H (2009) Behavioral tagging is a general mechanism of long-term memory formation *Proceedings of the National Academy of Sciences* 106: 14599-14604.

Bassett D S, Wymbs N F, Porter M A, Mucha P J, Carlson J M, Grafton S T (2011) Dynamic reconfiguration of human brain networks during learning *Proceedings of the National Academy of Sciences* 108: 7641-7646.

Braun E K, Wimmer G E, Shohamy D (2018) Retroactive and graded prioritization of memory by reward *Nat Commun* 9: 4886.

Brogden W J (1939) Sensory pre-conditioning *Journal of Experimental Psychology* 25: 323.

Burghy C A, Stodola D E, Ruttle P L, Molloy E K, Armstrong J M, Oler J A,... Essex M J (2012) Developmental pathways to amygdala-prefrontal function and internalizing symptoms in adolescence *Nature neuroscience* 15: 1736-1741.

Clewett D, Dunsmoor J, Bachman S L, Phelps E A, Davachi L (2022) Survival of the salient: Aversive learning rescues otherwise forgettable memories via neural reactivation and post-encoding hippocampal connectivity *Neurobiol Learn Mem* 187: 107572.

Cohen J (1992) A power primer *Psychological bulletin* 112: 155.

Cohen J. (2013). *Statistical power analysis for the behavioral sciences*: Academic press.

Cohen J R (2010) Decoding developmental differences and individual variability in response inhibition through predictive analyses across individuals *Frontiers in human neuroscience* 4: 47.

Coutanche M N (2013) Distinguishing multi-voxel patterns and mean activation: why, how, and what does it tell us? *Cogn Affect Behav Neurosci* 13: 667-673.

de Voogd, Fernández G, Hermans E J (2016) Awake reactivation of emotional memory traces through hippocampal–neocortical interactions *Neuroimage* 134: 563-572.

de Voogd, Klumpers F, Fernández G, Hermans E J (2017) Intrinsic functional connectivity between amygdala and hippocampus during rest predicts enhanced memory under stress *Psychoneuroendocrinology* 75: 192-202.

Dunsmoor J E, Murty V P, Davachi L, Phelps E A (2015) Emotional learning selectively and retroactively strengthens memories for related events *Nature* 520: 345-348.

Epstein R, Harris A, Stanley D, Kanwisher N (1999) The parahippocampal place area: recognition, navigation, or encoding? *Neuron* 23: 115-125.

Epstein R A, Ward E J (2010) How reliable are visual context effects in the parahippocampal place area? *Cereb Cortex* 20: 294-303.

Friston K (2012) Ten ironic rules for non-statistical reviewers *Neuroimage* 61: 1300-1310.

Friston K (2013) Sample size and the fallacies of classical inference *Neuroimage* 81: 503-504.

Geisser S. (1993). An Introduction to Predictive Inference. In: Chapman and Hall, New York.

Green S R, Kragel P A, Fecteau M E, LaBar K S (2014) Development and validation of an unsupervised scoring system (Autonomate) for skin conductance response analysis *International journal of psychophysiology* 91: 186-193.

Gruber M J, Ritchey M, Wang S F, Doss M K, Ranganath C (2016) Post-learning Hippocampal Dynamics Promote Preferential Retention of Rewarding Events *Neuron* 89: 1110-1120.

Günseli E, Aly M (2020) Preparation for upcoming attentional states in the hippocampus and medial prefrontal cortex *ELife* 9: e53191.

Hamann S (2001) Cognitive and neural mechanisms of emotional memory *TRENDS in Cognitive Sciences* 5: 394-400.

Hermans E J, Kanen J W, Tambini A, Fernández G, Davachi L, Phelps E A (2017) Persistence of amygdala–hippocampal connectivity and multi-voxel correlation structures during awake rest after fear learning predicts long-term expression of fear *Cerebral Cortex* 27: 3028-3041.

Hofstetter C, Achaibou A, Vuilleumier P (2012) Reactivation of visual cortex during memory retrieval: content specificity and emotional modulation *Neuroimage* 60: 1734-1745.

Holmes N M, Raipuria M, Qureshi O A, Killcross S, Westbrook F (2018) Danger Changes the Way the Mammalian Brain Stores Information About Innocuous Events: A Study of Sensory Preconditioning in Rats *eNeuro* 5.

Holmes N M, Wong F S, Bouchekioua Y, Westbrook R F (2021) Not "either-or" but "which-when": A review of the evidence for integration in sensory preconditioning *Neurosci Biobehav Rev*.

Jiang N, Xu J, Li X, Wang Y, Zhuang L, Qin S (2020) Negative parenting affects adolescent internalizing symptoms through alterations in amygdala-prefrontal circuitry: A longitudinal twin study *Biological psychiatry*.

Keogh R, Bergmann J, Pearson J (2020) Cortical excitability controls the strength of mental imagery *ELife* 9: e50232.

Kurth-Nelson Z, Barnes G, Sejdinovic D, Dolan R, Dayan P (2015) Temporal structure in associative retrieval *ELife* 4: e04919.

LaBar K S, Cabeza R (2006) Cognitive neuroscience of emotional memory *Nat Rev Neurosci* 7: 54-64.

Li W, Howard J D, Parrish T B, Gottfried J A (2008) Aversive learning enhances perceptual and cortical discrimination of indiscriminable odor cues *Science* 319: 1842-1845.

Liu Y, Lin W, Liu C, Luo Y, Wu J, Bayley P J, Qin S (2016) Memory consolidation reconfigures neural pathways involved in the suppression of emotional memories *Nat Commun* 7: 13375.

Liu Y, Nour M M, Schuck N W, Behrens T E J, Dolan R J (2022) Decoding cognition from spontaneous neural activity *Nat Rev Neurosci*.

MacKinnon D P, Lockwood C M, Williams J (2004) Confidence limits for the indirect effect: Distribution of the product and resampling methods *Multivariate behavioral research* 39: 99-128.

Murphy K, Birn R M, Handwerker D A, Jones T B, Bandettini P A (2009) The impact of global signal regression on resting state correlations: are anti-correlated networks introduced? *Neuroimage* 44: 893-905.

Murty V P, Tompary A, Adcock R A, Davachi L (2017) Selectivity in postencoding connectivity with high-level visual cortex is associated with reward-motivated memory *Journal of Neuroscience* 37: 537-545.

Mutso A A, Petre B, Huang L, Baliki M N, Torbey S, Herrmann K M,... Apkarian A V (2014) Reorganization of hippocampal functional connectivity with transition to chronic back pain *Journal of neurophysiology* 111: 1065-1076.

Nieuwenhuis S, Forstmann B U, Wagenmakers E-J (2011) Erroneous analyses of interactions in neuroscience: a problem of significance *Nature neuroscience* 14: 1105-1107.

Park H, Rugg M D (2010) Prestimulus hippocampal activity predicts later recollection *Hippocampus* 20: 24-28.

Phelps E A, LeDoux J E (2005) Contributions of the amygdala to emotion processing: from animal models to human behavior *Neuron* 48: 175-187.

Preacher K J, Hayes A F (2008) Asymptotic and resampling strategies for assessing and comparing indirect effects in multiple mediator models *Behavior Research Methods* 40: 879-891.

Qin S, Cho S, Chen T, Rosenberg-Lee M, Geary D C, Menon V (2014) Hippocampal-neocortical functional reorganization underlies children's cognitive development *Nat Neurosci* 17: 1263-1269.

Qin S, Piekema C, Petersson K M, Han B, Luo J, Fernandez G (2007) Probing the transformation of discontinuous associations into episodic memory: an event-related fMRI study *Neuroimage* 38: 212-222.

Richardson M P, Strange B A, Dolan R J (2004) Encoding of emotional memories depends on amygdala and hippocampus and their interactions *Nat Neurosci* 7: 278-285.

Ritchey M, Murty V P, Dunsmoor J E (2016) Adaptive memory systems for remembering the salient and the seemingly mundane *Behavioral and Brain Sciences* 39.

Sadacca B F, Wied H M, Lopatina N, Saini G K, Nemirovsky D, Schoenbaum G (2018) Orbitofrontal neurons signal sensory associations underlying model-based inference in a sensory preconditioning task *ELife* 7: e30373.

Schlichting M L, Frankland P W (2017) Memory allocation and integration in rodents and humans *Current Opinion in Behavioral Sciences* 17: 90-98.

Schlichting M L, Preston A R (2014) Memory reactivation during rest supports upcoming learning of related content *Proc Natl Acad Sci U S A* 111: 15845-15850.

Schlichting M L, Preston A R (2015) Memory integration: neural mechanisms and implications for behavior *Curr Opin Behav Sci* 1: 1-8.

Sevinc G, Hölzel B K, Greenberg J, Gard T, Brunsch V, Hashmi J A,... Lazar S W (2019) Strengthened hippocampal circuits underlie enhanced retrieval of extinguished fear memories following mindfulness training *Biological psychiatry* 86: 693-702.

Sharpe M J, Batchelor H M, Schoenbaum G (2017) Preconditioned cues have no value *ELife* 6: e28362.

Shohamy D, Daw N D (2015) Integrating memories to guide decisions *Current Opinion in Behavioral Sciences* 5: 85-90.

Shohamy D, Wagner A D (2008) Integrating memories in the human brain: hippocampal-midbrain encoding of overlapping events *Neuron* 60: 378-389.

Shrout P E, Bolger N (2002) Mediation in experimental and nonexperimental studies: new procedures and recommendations *Psychological methods* 7: 422.

Steiger J H (1980) Tests for comparing elements of a correlation matrix *Psychological bulletin* 87: 245.

Supekar K, Swigart A G, Tenison C, Jolles D D, Rosenberg-Lee M, Fuchs L, Menon V (2013) Neural predictors of individual differences in response to math tutoring in primary-grade school children *Proceedings of the National Academy of Sciences* 110: 8230-8235.

Sutherland G R, McNaughton B (2000) Memory trace reactivation in hippocampal and neocortical neuronal ensembles *Current opinion in neurobiology* 10: 180-186.

Takeuchi T, Duszkiewicz A J, Sonneborn A, Spooner P A, Yamasaki M, Watanabe M,... Morris R G (2016) Locus coeruleus and dopaminergic consolidation of everyday memory *Nature* 537: 357-362.

Tambini A, Davachi L (2013) Persistence of hippocampal multivoxel patterns into postencoding rest is related to memory *Proceedings of the National Academy of Sciences* 110: 19591-19596.

Tambini A, Davachi L (2019) Awake Reactivation of Prior Experiences Consolidates Memories and Biases Cognition *Trends Cogn Sci* 23: 876-890.

Tambini A, Ketz N, Davachi L (2010) Enhanced brain correlations during rest are related to memory for recent experiences *Neuron* 65: 280-290.

Tambini A, Rimmele U, Phelps E A, Davachi L (2017) Emotional brain states carry over and enhance future memory formation *Nat Neurosci* 20: 271-278.

Tompary A, Davachi L (2017) Consolidation Promotes the Emergence of Representational Overlap in the Hippocampus and Medial Prefrontal Cortex *Neuron* 96: 228-241 e225.

Wimmer G E, Shohamy D (2012) Preference by association: how memory mechanisms in the hippocampus bias decisions *Science* 338: 270-273.

Wong A H, Pittig A (2021) Avoiding a feared stimulus: Modelling costly avoidance of learnt fear in a sensory preconditioning paradigm *Biological Psychology*108249.

Wong F S, Westbrook R F, Holmes N M (2019) 'Online' integration of sensory and fear memories in the rat medial temporal lobe *ELife* 8.

Yoo J J, Hinds O, Ofen N, Thompson T W, Whitfield-Gabrieli S, Triantafyllou C, Gabrieli J D (2012) When the brain is prepared to learn: enhancing human learning using real-time fMRI *Neuroimage* 59: 846-852.

Zeng Y, Tao F, Cui Z, Wu L, Xu J, Dong W,... Qin S (2021) Dynamic integration and segregation of amygdala subregional functional circuits linking to physiological arousal *Neuroimage*118224.

Zhu Y, Chen X, Zhao H, Chen M, Tian Y, Liu C,... Qin S (2019) Socioeconomic status disparities affect children's anxiety and stress-sensitive cortisol awakening response through parental anxiety *Psychoneuroendocrinology* 103: 96-103.

Zhuang L, Wang J, Xiong B, Bian C, Hao L, Bayley P J, Qin S (2022) Rapid neural reorganization during retrieval practice predicts subsequent long-term retention and false memory *Nat Hum Behav* 6: 134-145.

[Editors' note: further revisions were suggested prior to acceptance, as described below.]

Reviewer #1 (Recommendations for the authors):The authors did a truly outstanding job responding to the concerns that I raised in the first review. I sincerely believe that the new introduction and discussion provide a much more emperically-motivated depiction of the prior literature, and does a stand-out job delineating these findings from other related concepts such as behavioral tagging.I was also excited to see such clear results from the additional analyses run. The trial-specific reactivation results give a much more clear picture of the types of reactivation supporting their retroactive memory effects, and the extent of the specificity is quite compelling.Similarly, the additional controls of the post-encoding analyses were quite strong. One point did emerge that was interesting, which was that there were negative correlations with neutral memory. These findings, albeit with some speculation, suggest that there is a prioritization of related emotional information rather than just an enhancement, meaning there may be a trade-off with neutral information. This would be very aligned with the circuits of interest and the role of noreadrenaline (see Clewett & Murty, 2019). While not completely necessary for this manuscript, additional discussion of this feature of the data that emerged from the new analyses could be included.

We thank the Reviewer for the enthusiasm and positive evaluation of our revised manuscript. We are encouraged by the Reviewer’s commendation that “the authors did a truly outstanding job.” We also appreciate the Reviewer for this thoughtful suggestion on interpreting new results from our post-encoding analyses. We have added several sentences in the Discussion section to acknowledge the possible trade-off between emotional and neutral information during post-encoding periods. The reference suggested by the Reviewer has also been cited appropriately. It now reads as follows:

Page 16: “Interestingly, hippocampal-neocortical connectivity changes involved in this reorganization positively correlated with memory in the aversive condition, but negatively correlated in the neutral condition (i.e., partial correlations controlling for memory in the other condition). It is possible to speculate that the emotion-charged retroactive memory benefit might not only reflect an enhancement of emotion-related information but also a suppression of other neutral information. These findings point toward a potential trade-off between prioritization of emotion-related memory and neglection of mundane neutral memory, aligning with the activity of locus coeruleus-norepinephrine system during post-encoding periods (Clewett and Murty, 2019).”

Reviewer #2 (Recommendations for the authors):The manuscript by Zhu et al. has improved with revision; I appreciate the addition of the more specific trial-level reactivation analysis which more directly tests the notion that reactivation is a mechanism supporting retroactive emotional memory enhancements. The authors have responded to the prior concerns thoroughly. New details were added regarding ROI definition, in response to prior comments. These new details currently limit my enthusiasm for the manuscript due to the lack of straightforward presentation of some of the analyses/results in the paper: how regions in Figures 4-6 were identified (all other results/analyses are clearly described; see below). Otherwise, the paper makes an important contribution to the field.

We appreciate the Reviewer for the positive evaluation of our additional analyses and revised manuscript. We are pleased that the Reviewer values “this paper making an important contribution to the field”.

A primary concern is the way regions were isolated/identified in the analyses performed after the pattern similarity (those in Figures 4-6). I am not sure whether the FFA/LOC was defined once (i.e. those used in Figures 2-3) and then voxels within those regions were isolated showing connectivity effects for analyses in Figures 4-6, or whether separate regions were isolated across each analysis and are referred to with a common label (implying they are the same). This stems from the second paragraph of the ROI definition section (Methods) which I found to be unclear. As written, it is *implied* that FFA and LOC are re-defined in subsequent analyses "FFA and LOC were derived from a group contrast map of the aversive relative to neutral condition" and "significant clusters (i.e. LOC…) … were derived from the group-level multiple regression analysis on connectivity…". If the ROIs are re-defined in different analysis, it is misleading to use the same label to refer to them across analyses (which implies that they are the same regions or would at least be restricted to a common definition). Please both (1) clarify the approach because it is still not clear to me how the ROIs were defined for connectivity analysis (i.e. did you search for specific contrast within the original FFA/LOC definitions, or were these ROIs re-defined for each analysis within the broader anatomical mask from WFU pickatlas) and (2) the description of the ROIs/results in the main manuscript text should not be misleading as to how the analysis is conducted. If separate regions are isolated for different analyses, then this should be reflected in the labels used for these regions (they should not all have the same label which imply face and object processing regions if they are isolated by other means).The ROI definition issue raises possible problems of non-independence and multiple comparisons corrections which are currently obfuscated given that the analysis steps are not spelled out. If the 'FFA' and 'LOC' regions shown in Figure 4 were defined from showing greater functional connectivity w/ the hippocampus during aversive vs. neutral trials (not how these regions are typically defined), then the results reported in Figure 4C are non-independent and inferential tests (ANOVAs) should not be performed on this data as it is circular. It is also stated that the gPPI was performed separately for initial encoding and emotional learning phases and clusters were isolated from each – presumably the clusters shown in Figure 4 were isolated from the emotional learning analysis since they show a strong effect in that phase? Please state which analysis they were defined from. The problem of non-independence seems to be remedied in Figure 6 since no statistics are reported and just brain regions are shown. Another possible issue is multiple comparisons corrections, especially *if the FFA/LOC definitions are not carried forward into subsequent analyses*. The original ROI definition does not include an explicit correction but uses a stringent threshold of P<.0001, 30 voxels (fine for ROI definition such that ROIs are interrogated in later analyses). But this thresholding procedure is referenced in subsequent analyses, so presumably all regions isolated in Figures 4-6 are using this criterion. Thus it needs to be specified how this criterion satisfies multiple comparison/family wise error correction / was chosen. Moreover, please clarify if the FFA/LOC regions in Figures 4-6 were also constrained anatomically by the wfu pickatlas regions and neurovault contrasts.

We would like to thank the Reviewer for this opportunity to clarify the definition of FFA/LOC ROIs in our connectivity analyses, as well as possible problems of non-independence and multiple comparisons correction. We have undertaken the following steps to address these concerns.

First of all, we apologize for our unclear descriptions about the functional definition of separate FFA/LOC ROIs in connectivity analyses. Specifically, for task-dependent connectivity analysis in Figures 4-5 (i.e., now Figure 3 in the revised manuscript), we identified ROIs in the left middle portion of FFA (mFFA) (Visconti di Oleggio Castello et al., 2021) and the left superior portion of LOC (sLOC) (Barbieri et al., 2019; Olivo et al., 2019) using a group connectivity contrast during the emotional learning phase, which showed greater connectivity with the hippocampus in the aversive relative to neutral condition, by a height threshold of *p* < 0.005 and an extent threshold of *p* < 0.05 with family-wise error correction for multiple comparisons based on nonstationary suprathreshold cluster-size distributions computed by Monte Carlo simulations (Nichols and Hayasaka, 2003). Likewise, for post-encoding connectivity analysis in Figure 6 (i.e., now Figure 4 in the revised manuscript), we identified the ROI in the left inferior portion of LOC (iLOC) (Barbieri et al., 2019; Olivo et al., 2019) using a group connectivity contrast of Rest 2 relative to Rest 1 in the multiple regression analysis in relation to associative memory performance, by the same threshold criterion mentioned above. Same with pattern similarity analysis, the mFFA, sLOC and iLOC ROIs identified in the task-dependent and post-encoding connectivity analyses in Figures 4-6 (i.e., now Figure 3-4 in the revised manuscript) were further constrained by an anatomically defined mask from the WFU PickAtlas in combination with the mask derived from the Neurosynth platform with “face recognition” or “object recognition” as a searching term (*p* < 0.01 with FDR correction). We have described the definition of FFA/LOC ROIs for our connectivity analyses with more details in the Methods section, and it now reads as follows:

Page 21: “For task-dependent hippocampal connectivity analysis during initial and emotional learning phases, the right amygdala, left middle portion of FFA (mFFA) (Visconti di Oleggio Castello et al., 2021) and left superior portion of LOC (sLOC) (Barbieri et al., 2019; Olivo et al., 2019) ROIs were defined using a group-level connectivity contrast of the aversive relative to neutral condition during the emotional learning phase, by a height threshold of *p* < 0.005 and an extent threshold of *p* < 0.05 with family-wise error correction for multiple comparisons based on nonstationary suprathreshold cluster-size distributions computed by Monte Carlo simulations (Nichols and Hayasaka, 2003). For post-encoding hippocampal connectivity analysis during three resting phases, significant clusters in the left inferior portion of LOC (iLOC) (Barbieri et al., 2019; Olivo et al., 2019) as well as the bilateral IPL, right PCC, right aPFC and right mPFC were derived from the group-level multiple regression analyses on connectivity contrast maps (i.e., Rest 2 vs. 1, and Rest 3 vs. 2) with interaction effects between aversive and neutral conditions, by the same threshold criterion as task-dependent connectivity analysis above. Same with pattern similarity analysis, the mFFA, sLOC and iLOC ROIs were further constrained separately by the above-mentioned FFA or LOC mask.”

In addition, we have also re-labeled these FFA/LOC ROIs with their specific anatomical portions in the corresponding figures and legends (Figure 2 : ventral LOC and FFA for pattern similarity analysis; Figure 3 : mFFA and sLOC for task-dependent connectivity analysis; Figure 4 : iLOC for post-encoding connectivity analysis), as well as when appropriate throughout the entire revised manuscript.

Second, for the Reviewer’s concern on non-independence problem of task-dependent connectivity analysis in Figure 4 (i.e., now Figure 3 in the revised manuscript). Indeed, our mFFA and sLOC ROIs were derived from the connectivity contrast during emotional learning phase, which identified clusters with greater hippocampal functional coupling in the aversive (vs. neutral) condition. To avoid double dipping and circular analysis, we decided to omit the ANOVAs in original Figure 4E-G and move corresponding bar graphs without any statistics into the Supplemental Materials only for visualization purpose. Our main conclusion of emotion-charged enhancement on hippocampal connectivity with the amygdala and neocortex during emotional learning still holds based on results from the whole-brain contrast without those ANOVAs. Thus, this revision by omitting our original ANOVAs results does not change our main conclusion. In addition, we also conducted a parallel control PPI analysis for the initial learning phase. As expected, this analysis revealed no any emotion-charged effect (i.e., aversive vs. neutral) during initial learning in the mFFA and sLOC.

It is worth to note that our functionally defined mFFA and sLOC ROIs, according to the aversive (vs. neutral) condition during emotional learning, were further used to investigate emotion-induced connectivity changes in relation to associative memory (i.e., later remembered vs. forgotten) in the original Figure 5 (i.e., now Figure 3E-G in the revised manuscript). Since the definition of these ROIs does not bias toward either remembered or forgotten condition, we thus feel that our analyses in Figure 3E-G differ from double dipping according to Kriegeskorte and colleagues’ view on “the use of the same dataset (contrast here) for selection and selective analysis will give distorted descriptive statistics and invalid statistical inference” (Kriegeskorte et al., 2009). Nevertheless, we agree that our functionally defined ROIs in the present study could somehow bring potential selection bias toward the emotion learning phase, and thus are not fully independent compared to the ideal case (i.e., anatomical ROIs). Given that such non-independence with potential selection bias differs from double dipping and could be acceptable under certain circumstances as discussed by previous studies (Kriegeskorte et al., 2010; Kriegeskorte et al., 2009), we feel that our current approach is state-of-the-art and valid due to three considerations: (1) our central goal is to examine emotion-charged connectivity changes in aversive (vs. neutral) condition on our hypothesized ROIs; (2) these ROIs could not be functionally defined during initial learning without any emotional effect; (3) anatomically defined ROIs might include many irrelevant voxels and reduce statistical sensitivity.

In sum, we have now better clarified our ROI definition and omitted ANOVAs results (original Figure 4E-G) in the Results section, as well as moved corresponding plots into the Supplemental Materials (Figure 3—figure supplement 1). We have also incorporated results from the original Figure 5 into updated Figure 3. In addition, we have explicitly acknowledged potential bias of our functionally defined ROIs in the Limitations section of our revised manuscript. The amended texts read as follows:

*Page 9-11 in Results*: “Emotional learning enhances hippocampal coupling with the amygdala and neocortex which predicts associative memory for initial neutral events.

To investigate how emotional learning modulates functional interactions of the hippocampus and its related neural circuits involved in memory integration, we conducted a task-dependent psychophysiological interaction (PPI) analysis for the emotional learning phase to assess functional connectivity of the hippocampal seed with every other voxel of the brain (Figure 3A). In line with our priori hypothesis, we identified significant clusters in the right amygdala (Figure 3B), left middle portion of FFA (mFFA; Figure 3C) and left superior portion of LOC (sLOC; Figure 3D), which showed greater functional coupling with the hippocampus in the aversive (vs. neutral) condition during emotional learning (Figure 3—figure supplement 1 for visualization). We further conducted a parallel control PPI analysis for the initial learning phase. This analysis revealed no any reliable emotional effect (i.e., aversive vs. neutral) during initial learning in the three ROIs identified above (Figure 3—figure supplement 1 for visualization). Altogether, these results indicate that emotional learning induces functional connectivity changes with prominent increases in hippocampal-amygdala and hippocampal-neocortical coupling.”

Page 17 in Limitations: “And our functionally defined ROIs may be susceptible to potential selection bias. Independent ROI definition (i.e., anatomically defined ROIs) would reinforce the reproducibility of brain-behavior correlations in future studies.”

Figure 3—figure supplement 1 in Supplemental Materials.

Third, regarding the Reviewer’s concern on correction for multiple comparisons, we would like to clarify that only the bilateral ventral portion of LOC (vLOC) and the bilateral FFA ROIs for pattern similarity analysis were defined using this uncorrected threshold (i.e., a threshold of *p* < 0.0001 with more than 30 voxels) (Miller et al., 2022). All of other ROIs (i.e., mFFA, sLOC and iLOC) for connectivity analyses were defined using a widely-used corrected threshold. Actually, a threshold of *p* < 0.0001 with more than 30 voxels is satisfied with a height threshold of *p <* 0.0001 and an extent threshold of *p* < 0.05 with family-wise error correction for multiple comparisons, by using nonstationary suprathreshold cluster-size approach based on Monte Carlo simulations for the whole brain with an unbiased, anatomically defined, gray matter mask (Nichols and Hayasaka, 2003). We used this extremely stringent threshold criterion to restrict vLOC and FFA ROIs for pattern similarity analysis by the two following reasons: (1) these ROIs were defined by the activation contrasts of all encoding trials relative to fixation, which resulted in a large area of significant clusters; (2) we aim to specify the most engaged regions activated during initial learning and potentially reactivated during emotional learning. Therefore, this stricter correction for multiple comparisons could help us avoid overlaps and identify more specific ROIs. We have now updated the correction criterion in the Methods section. We have also added several sentences to describe this univariate activation analysis and its results in the Supplemental Materials (Figure 2—figure supplement 1). The amended texts read as follows:

Page 20 in Methods: “To investigate the emotion-charged reactivation in both condition- and trial-level pattern similarity analyses, we identified the bilateral hippocampal, bilateral ventral LOC (vLOC) and bilateral FFA ROIs by the overlapping area of two group-level univariate activation contrasts of face-object association encoding (i.e., initial learning) and face-voice association encoding (i.e., emotional learning) separately relative to fixation (i.e., all encoding trials vs. fixation during each learning phase), using a stringent height threshold of p < 0.0001 and an extent threshold of p < 0.05 with family-wise error correction for multiple comparisons based on nonstationary suprathreshold cluster-size distributions computed by Monte Carlo simulations (Miller et al., 2022; Nichols and Hayasaka, 2003) (Figure 2—figure supplement 1).”

Figure 2—figure supplement 1 in Supplemental Materials.

Altogether, we would like to thank the Reviewer again for prompting us to clarify our ROI definition and potential problems of non-independence and multiple comparisons correction. These comments are helpful to improve the integrity of our data analyses and results especially for Figures 4-6 (i.e., now Figure 3-4 in the revised manuscript). We hope that our above responses and revisions have increased your confidence on the clarity of our ROI definition and the presentation of corresponding analyses and results.

Reviewer #3 (Recommendations for the authors):In this version of the manuscript, Zhu and colleagues have revised their introduction and discussion to situate their experiment within the framework of sensory preconditioning, rather than behavioral tagging. I found the theoretical framework to be much better developed in this revision, and the results link more clearly to the existing conditioning and memory integration literature. Another major revision was the inclusion of a trial-specific reactivation analysis, in which they showed that trial-specific hippocampal patterns were reinstated during the emotional learning phase. This could provide a mechanism by which initially-neutral associations are strengthened through emotional learning. This is an interesting and potentially important result.In reading the manuscript again, I was impressed with the robust behavioral findings, and I thought the pattern similarity analyses were largely convincing (see one comment below).

We appreciate the Reviewer’s positive assessment of our work by commending that “I was impressed with the robust behavioral findings” and “the pattern similarity analyses were largely convincing”. We also appreciate thoughtful comments that improved our manuscript.

I remain concerned that there is too much emphasis placed on the results based on across-subject correlations, given the relatively small sample size (N=28). This includes the structural equation model as well as the results of the resting-state functional connectivity analyses. I am more convinced by the prediction analyses that follow up on results from the pattern similarity and task-related connectivity analyses. The resting-state analyses are particularly susceptible to the problems of underpowered correlations because they were computed across all voxels in the brain (see Marek et al. 2022 Nature for further discussion). In their response to the previous reviews, the authors described the across-subject correlation analyses as "exploratory" and complementary to the main lines of evidence. Yet this is not how they are presented in the paper itself, where the SEM and resting-state analyses are highlighted as key findings (e.g., see lines 558-563 in the results summary on p. 29). At minimum, more caution should be expressed throughout, with analyses clearly marked as exploratory.

We appreciate the Reviewer for constructive comments and helpful suggestions on across-subject correlations and the sample size. We apologize for our inappropriate highlights of the across-subject correlation results from structural equation modeling and resting-state functional connectivity analyses. Indeed, these results should be described as “exploratory” and “complementary” to our main findings. We have now undertaken four following steps to address this concern. First, considering our hypothesis and conclusion on the mediating effect, we have now decided to focus on results from the mediation analysis rather than “structural equation modeling” (please see our response to point 2 below for details). Second, we conducted additional prediction analyses for resting-state functional connectivity data to confirm the robustness of originally reported across-subject correlation results. We have added several sentences to describe these new prediction analyses and results in the Results section. Given that our manuscript is dense with results and to avoid redundancy, we decided not to include the prediction plots in the revised manuscript, which show very similar patterns as correlation plots (Figure 4—figure supplement 1). It now reads as follows:

Page 13: “As a complement, the machine learning-based prediction analyses for post-learning hippocampal connectivity changes revealed very similar patterns of positive predictions for associative memory in the aversive condition (iLOC: r_(predicted, observed)_ = 0.47; IPL: r_(predicted, observed)_ = 0.54; PCC: r_(predicted, observed)_ = 0.51; aPFC: r_(predicted, observed)_ = 0.52; mPFC: r_(predicted, observed)_ = 0.48; all *p* < 0.005 while controlling for memory in the neutral condition), but negative predictions in the neutral condition (iLOC: r_(predicted, observed)_ = -0.45; IPL: r_(predicted, observed)_ = -0.55; PCC: r_(predicted, observed)_ = -0.50; aPFC: r_(predicted, observed)_ = -0.53; mPFC: r_(predicted, observed)_ = -0.51; all *p* < 0.006 while controlling for memory in the aversive condition).”

Third, we fully agree with the Reviewer that reproducible brain-wide association studies (BWAS) require a relatively large sample size, especially for detecting associations between inter-individual differences in brain structure or function and complex cognitive/mental health phenotypes (Marek et al. 2022). In our current study, however, we actually conducted hypothesis-driven brain-behavior association analyses with a particular focus on functional brain metrics (i.e., RSA, connectivity) and emotion-charged memory performance – those variables were directly derived from our dedicated fMRI experimental manipulation, rather than from general cognitive/mental health phenotypes that are unrelated to the fMRI design. Nevertheless, we acknowledged that our limited sample size for the mediation analysis and the resting-state connectivity analysis in relation to memory performance is susceptible to the problem of underpowered correlations. As suggested, we have now explicitly acknowledged our mediation and resting-state analyses as “exploratory” and “complementary” throughout the entire revised manuscript. The amended texts read as follows:

Page 3 in Introduction: “A set of multi-voxel pattern similarity, task-dependent functional connectivity and machine learning-based prediction analyses, in conjunction with exploratory task-free functional connectivity and mediation analyses, …”

Page 11-12 in Results: “Therefore, we assumed a potential mediatory pathway among these variables, … Based on above theoretical motivation as well as empirical observations, we further implemented an exploratory mediation analysis ….

… Altogether, these exploratory results indicate that increased hippocampal-amygdala coupling might indirectly account for emotion-charged associative memory, likely mediating by hippocampal-neocortical couplings during emotional learning.”

Page 12-13 in Results: “Furthermore, to explore how offline hippocampal connectivity changes at resting states prior to and after emotional learning contribute to emotion-charged retroactive memory benefit, … Specifically, the exploratory whole-brain analyses revealed that… These exploratory results indicate a potential shift of post-learning hippocampal connectivity from the object-sensitive lateral occipital cortex to more distributed transmodal prefrontal and posterior parietal areas, which predicts emotion-charged retroactive memory benefit.”

Page 13 in Discussion: “Complementally, hippocampal-amygdala coupling positively predicted the emotion-charged retroactive memory benefit, mediating by increased hippocampal-neocortical interactions. Moreover, we explored a potential shift of hippocampal-neocortical connectivity contributing to the emotion-charged retroactive memory enhancement during post-learning rests, …”

Page 15-16 in Discussion: “Third, we explored that increased hippocampal-neocortical coupling could mediate the positive relationship between hippocampal-amygdala coupling and emotion-charged retroactive memory enhancement. This exploratory mediation effect suggests …”

Last but not least, we explored a potential hippocampal-neocortical functional reorganization during post-learning rests predictive of emotion-charged retroactive memory benefit, …”

Fourth, we have also noted the relatively small sample in the Limitations section of our revised manuscript. It now reads as follows:

Page 17: “Second, our relatively small sample size in fMRI Study 3 would be underpowered to detect individual differences in across-subject correlations and mediation analyses. A larger sample size would reinforce the reproducibility of brain-behavior correlations in future studies.”

For the structural equation model, the authors used a bootstrapping technique that may improve the ability to estimate direct and indirect effects even in small sample sizes. The model fit indices, however, seem to have been computed in the standard way. Given that the analysis may be underpowered to detect model misspecification, the authors should tone down their description of the model fit (both for the primary model as well as the alternative tested models).

We agree with the Reviewer that the model fit indices computed in a standard way may be underpowered to detect model misspecification. We have now undertaken three steps to mitigate this concern. **First**, we decided to focus on the mediation effect rather than “structural equation modeling”. In other words, our main goal is to investigate whether hippocampal-mFFA/sLOC connectivity could account for the indirect relationship between hippocampal-amygdala connectivity and emotion-charged memory performance. Second, as the Reviewer suggested, we have now toned down our descriptions of the model fit, by removing the model fit indices for both primary model and alternative tested models in the Results, Methods sections and the Supplementary Materials (Figure 3—figure supplement 3). **Third**, although previous fMRI studies also conducted mediation analyses on similar sample sizes (Hare et al., 2010; Jimura et al., 2010) as our current study, we agree that small sample size limits the power to detect individual differences in across-subject correlations. Therefore, we have now clearly marked the mediation analysis as an “exploratory analysis” throughout the revised manuscript (please also see our response to point 1 above). The amended texts read as follows:

Page 11 in Results: “This exploratory mediation analysis revealed two significant mediating effects on the positive relationship between hippocampal-amygdala connectivity and memory outcome …”

Page 24 in Methods: “We conducted a mediation analysis to further explore how emotional learning affects initial associative memory through functional amygdala-hippocampal-neocortical pathways during emotional learning. … The mediating effect of hippocampal-neocortical connectivity was tested by a bias-corrected bootstrap with 1,000 samples, which could improve the sensitivity and robustness of statistical estimates in small-to-moderate samples (Preacher and Hayes, 2008; Shrout and Bolger, 2002).”

In the pattern similarity analyses, reinstatement was observed at the onset of the voice but not in response to the face cue alone. Yet it appears that the voice onset occurred only 2s after the onset of the face cue, which is the equivalent of one TR. With that temporal resolution, it shouldn't be possible to reliably separate these two signals in time, and including them in the same model may lead to unstable parameter estimates. How have the authors addressed this?

Indeed, it is challenging to reliably separate the two BOLD-fMRI signals associated with face cues and face-voice pairs with only 2s intervals. First of all, we would like to clarify that we did not actually intend to separate these two signals in the same model. Instead, we included the presentations of face cues in aversive and neutral conditions into the model as covariates of no interest, in order to control for brain activity associated with the process of face cues. This approach has been used in previous studies to account for brain activity patterns associated with tasks or events of no interest (Ballard et al., 2011; Costumero et al., 2015; Crockett et al., 2017). Our main goal of this analysis is to investigate the emotional effect induced by aversive voices during the presentation of face-voice associations, while controlling for brain activity linked to the presentation of face cues. Second, as suggested by the Reviewer #2’s minor comment Q2-13: “this is essentially captured by showing a main effect of emotion in the Figure 3 results” (i.e., now Figure 2 with trial-level reactivation results in the revised manuscript), we have now moved these condition-level reactivation results into the Supplementary materials (Figure 2—figure supplement 2 ). Thus, this potentially unreliable signal separation would not affect our main conclusions derived from trial-level pattern similarity analysis. Third, we have also conducted an additional GLM analysis of emotional learning phase which only included the presentations of face-voice associations in aversive and neutral conditions as two regressors of interest, without considering the presentation of face cues. This additional analysis revealed almost identical pattern of emotional effect for each ROI as our current analysis (hippocampus: t_(27)_ = 2.53, p = 0.017, d_av_ = 0.24; vLOC: t_(27)_ = 2.14, p = 0.042, d_av_ = 0.14; FFA: t_(27)_ = 2.98, *p* = 0.006, d_av_ = 0.19; see Author response image 3). To avoid redundancy, we decided not to include these results in the revised manuscript.

Fourth, in response to the Reviewer’s comment, nevertheless, we have now noted this issue in the Limitations section of our revised manuscript. It now reads as follows:

Page 17: “Third, it is challenging to reliably separate neural signals associated with "face cues" and "face-voice associations" due to only an interval of 2 seconds. Future design with longer and jittered intervals may resolve this issue.”

There are a couple of places in the Results where non-significant results are described as significant, in cases when one ROI shows a significant effect but the other doesn't: the LOC interaction in line 295, the amygdala correlation in line 433. Conclusions should also be updated to refer only to those findings that are significant.

We thank the Reviewer for pointing this out. We have gone through the entire manuscript to carefully check and change the descriptions of marginally but not significant results (i.e., 0.05 < *p* < 0.08). And we have also updated the corresponding conclusions accordingly. The amended texts read as follows:

Page 17 in Results: “These analyses revealed significant main effects of Emotion in the hippocampus (F_(1, 25)_ = 9.48, *p* = 0.005, partial η^2^ = 0.28) and vLOC (F_(1, 25)_ = 9.34, *p* = 0.005, partial η^2^ = 0.27), as well as a significant Emotion-by-Measure interaction effect in the hippocampus (F_(1, 25)_ = 4.89, *p* = 0.036, partial η^2^ = 0.16) and a similar trend but non-significant interaction in the vLOC (F_(1, 25)_ = 3.54, *p* = 0.072, partial η^2^ = 0.12), … Post-hoc comparisons (with covariates controlled) revealed significantly higher pair-specific similarity (hippocampus: F_(1, 25)_ = 7.24, *p* = 0.013, partial η^2^ = 0.22; vLOC: F_(1, 25)_ = 8.28, *p* = 0.008, partial η^2^ = 0.25), but only a non-significant trend of higher across-pair similarity (hippocampus: F_(1, 25)_ = 4.16, *p* = 0.052, partial η^2^ = 0.14; vLOC: F_(1, 25)_ = 3.37, *p* = 0.078, partial η^2^ = 0.12) in the aversive than neutral condition. We also observed significantly higher pair-specific than across-pair similarity in the hippocampus (F_(1, 25)_ = 4.42, *p* = 0.046, partial η^2^ = 0.15) and a similar but non-significant trend in the vLOC (F_(1, 25)_ = 2.96, *p* = 0.097, partial η^2^ = 0.11) in the aversive condition, but not in the neutral condition (F_(1, 25)_ < 2.20, *p* > 0.150, partial η^2^ < 0.08 in both ROIs). … These results indicate that emotional learning prompts greater trial-specific reinstatement relative to category-level representation in the hippocampus, and it also leads to a similar but non-significant trend in the vLOC.”

Page 9-11 in Results: “A similar trend (though not significant) was also shown in hippocampal-amygdala connectivity (i.e., remembered with high confidence relative to forgotten; aversive: r_(predicted, observed)_ = 0.24, *p* = 0.076; neutral: r_(predicted, observed)_ = -0.25, *p* = 0.839). Further Steiger’s tests revealed significant differences in correlation coefficients between two conditions for hippocampal coupling with mFFA (z = 2.13, *p* = 0.033) and sLOC (z = 2.15, *p* = 0.032), and a non-significant trend of difference for hippocampal-amygdala coupling (z = 1.79, *p* = 0.073). These results indicate that emotion-charged hippocampal connectivity with stimulus-sensitive neocortical regions positively predicts associative memory in the aversive but not neutral condition, implying an emotional specificity effect, though hippocampal-amygdala connectivity only shows a similar trend but non-significant effect.”

Page 11 in Results: “According to our empirical observations, memory performance in the aversive condition showed only a trending positive correlation with hippocampal-amygdala connectivity, but highly positive correlations with hippocampal-mFFA/sLOC connectivity.”

Page 15 in Discussion: “Critically, we found the prominent emotional enhancement for trial-specific reactivation (i.e., pair-specific similarity) in the hippocampus and vLOC, but not in the FFA. … In addition, the trending but not significant emotional effects for category representation (i.e., across-pair similarity) found in the hippocampus, vLOC and FFA might be due to more attention in the aversive (vs. neutral) condition generally attracted by the screaming stimuli, rather than trial-specific reactivation.”

Page 15 in Discussion: “Second, although such hippocampal connectivity patterns did not show reliable Emotion-by-Memory interaction effect, results from our prediction analyses revealed that hippocampal connectivity with the (delete: the amygdala and) neocortical regions during emotional learning positively predicted memory for face-object associations in the aversive rather than neutral condition.”